# Changes in neurotensin signalling drive hedonic devaluation in obesity

Neta Gazit Shimoni[1,7], Amanda J. Tose[1,7], Charlotte Seng[2], Yihan Jin[3,6], Tamás Lukacsovich[2], Hongbin Yang[4], Jeroen P. H. Verharen[1], Christine Liu[1], Michael Tanios[1], Eric Hu[1], Jonathan Read[1], Lilly W. Tang[1], Byung Kook Lim[5], Lin Tian[3,6], Csaba Földy[2] & Stephan Lammel[1✉]

Calorie-rich foods, particularly those that are high in fat and sugar, evoke pleasure in both humans and animals[1]. However, prolonged consumption of such foods may reduce their hedonic value, potentially contributing to obesity[2–4]. Here we investigated this phenomenon in mice on a chronic high-fat diet (HFD). Although these mice preferred high-fat food over regular chow in their home cages, they showed reduced interest in calorie-rich foods in a no-effort setting. This paradoxical decrease in hedonic feeding has been reported previously[3–7], but its neurobiological basis remains unclear. We found that in mice on regular diet, neurons in the lateral nucleus accumbens (NAcLat) projecting to the ventral tegmental area (VTA) encoded hedonic feeding behaviours. In HFD mice, this behaviour was reduced and uncoupled from neural activity. Optogenetic stimulation of the NAcLat→VTA pathway increased hedonic feeding in mice on regular diet but not in HFD mice, though this behaviour was restored when HFD mice returned to a regular diet. HFD mice exhibited reduced neurotensin expression and release in the NAcLat→VTA pathway. Furthermore, neurotensin knockout in the NAcLat and neurotensin receptor blockade in the VTA each abolished optogenetically induced hedonic feeding behaviour. Enhancing neurotensin signalling via overexpression normalized aspects of diet-induced obesity, including weight gain and hedonic feeding. Together, our findings identify a neural circuit mechanism that links the devaluation of hedonic foods with obesity.

Excessive consumption of high-calorie foods is a key contributor to the development and progression of obesity in humans and animals[8,9]. Chronic exposure to a HFD profoundly influences eating behaviours, particularly those driven by the pleasurable (that is, hedonic) properties of food[10]. To investigate how a chronic HFD affects these behaviours, we placed C57Bl/6 mice on a chronic HFD. Although these mice consistently preferred high-fat chow over standard chow in their home cages (Fig. 1a–c), they paradoxically exhibited a reduced drive to opportunistically consume high-calorie foods in an acute feeding assay, even when no effort was required to obtain the food (Fig. 1d). One possible explanation is that chronic HFD exposure leads to a reduction in the hedonic value of high-calorie foods, decreasing their pleasurable or rewarding aspects.

The mesolimbic dopamine system, consisting of dopamine-producing cells in the VTA projecting to the nucleus accumbens (NAc), has been implicated in the motivational aspects of feeding behaviour[11–13], although other dopaminergic projections are also likely to be involved[14]. Activation of VTA dopamine neurons projecting to the NAc is associated with the rewarding aspects of food consumption[15]. Anticipation of food or liquid rewards enhances dopamine neuron firing, promoting goal-directed behaviours[15–18]. Conversely, chronic HFD exposure has been shown to reduce dopamine activity in both mice and humans[2,19–24], potentially impairing reward-related processes and contributing to obesity.

Although much research has focused on the role of dopamine neurons in feeding and obesity, less is known about the effects of chronic HFD on inhibitory projections from the NAc to the VTA. The NAc provides substantial GABAergic input to the VTA, directly and indirectly influencing dopamine neurons[25,26]. We previously demonstrated that optogenetic stimulation of the NAcLat→VTA pathway induces robust reward-related behaviours, such as place preference and intracranial self-stimulation, possibly via disinhibition of dopamine neurons[26]. However, whether increased neural activity in this pathway is associated with hedonic feeding behaviours and how it is affected by diet-induced obesity remain unclear.

## Chronic HFD disrupts NAcLat→VTA activity

To study the effects of chronic HFD on NAcLat→VTA activity during feeding behaviours, we first injected C57Bl6 mice with a retrogradely

[1]Department of Neuroscience and Helen Wills Neuroscience Institute, University of California Berkeley, Berkeley, CA, USA. [2]Brain Research Institute, Faculties of Medicine and Science, University of Zurich, Zürich, Switzerland. [3]Department of Biochemistry and Molecular Medicine, School of Medicine, University of California Davis, Davis, CA, USA. [4]Department of Neurobiology and Department of Affiliated Mental Health Center of Hangzhou Seventh People's Hospital, Zhejiang University School of Medicine, NHC and CAMS Key Laboratory of Medical Neurobiology, MOE Frontier Science Center for Brain Science and Brain-Machine Integration, School of Brain Science and Brain Medicine, Zhejiang University, Hangzhou, China. [5]Division of Biological Sciences, University of California San Diego, San Diego, CA, USA. [6]Present address: Max Planck Florida Institute For Neuroscience, Jupiter, FL, USA. [7]These authors contributed equally: Neta Gazit Shimoni, Amanda J. Tose. ✉e-mail: lammel@berkeley.edu

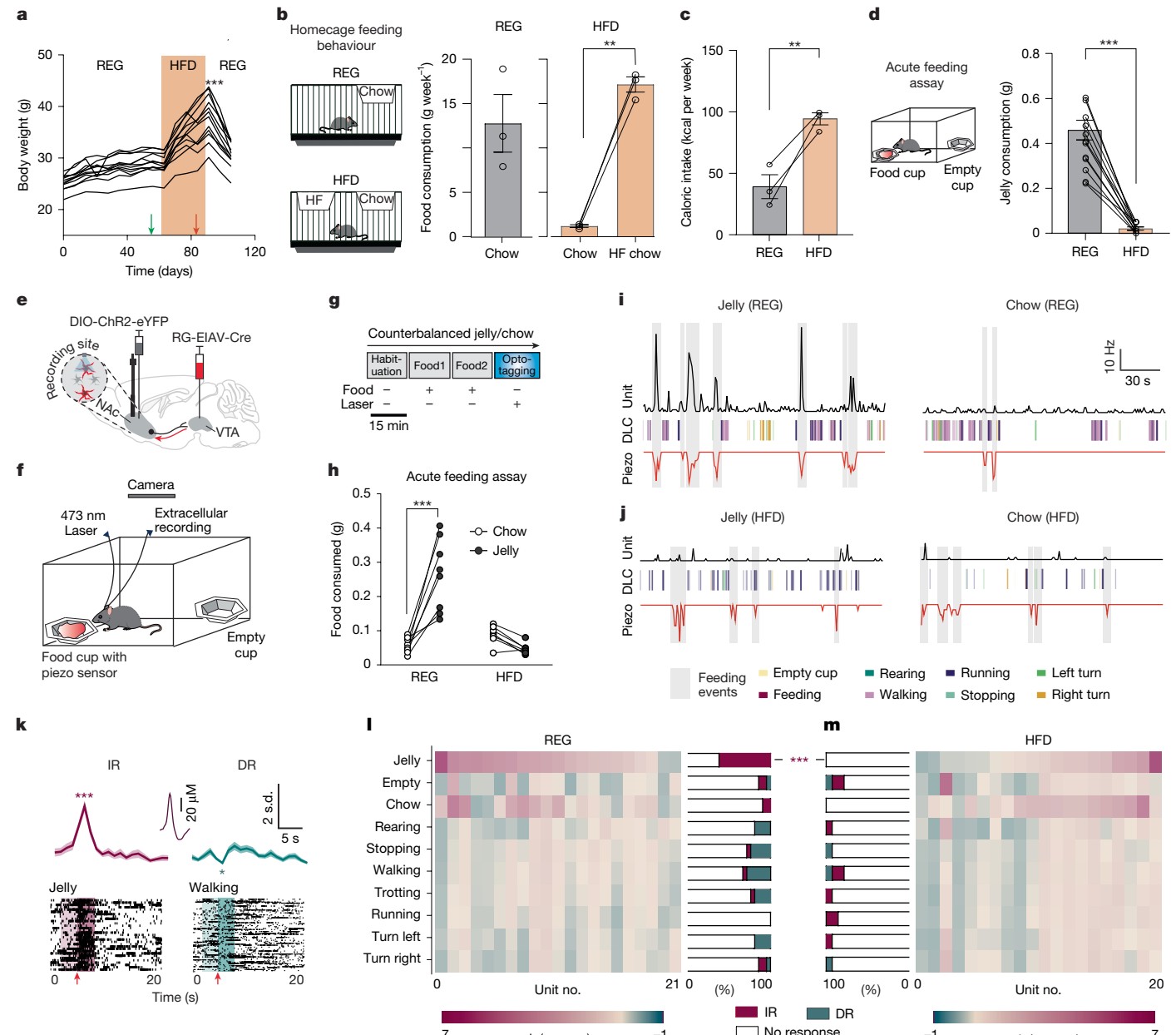

**Fig. 1 | Chronic HFD promotes uncoupling of NAcLat→VTA activity during hedonic feeding. a**, Body weight of REG mice that are switched to HFD and then returned to REG. Arrows indicate timing of acute feeding assays on REG (green) and HFD (red) (\*\*\*$P < 0.001$, 1-way repeated measures ANOVA with Holm–Šídák multiple comparisons test; $n = 12$ mice). **b,c**, Mean weekly consumption of regular chow and high-fat (HF) chow (\*\*$P = 0.0022$) (**b**) and caloric intake (\*\*$P = 0.0036$) (**c**) in home cages while on REG or HFD ($n = 3$ cages; normalized as grams per mouse per week; 2-sided paired Student's $t$-test). **d**, Mean jelly consumption during acute feeding assays for REG mice and after 4 weeks of HFD (\*\*\*$P < 0.001$, 2-sided paired Student's $t$-test; $n = 12$ mice). **e**, Experimental design. **f**, Acute feeding assay. **g**, Timeline: trial 1 (habituation, no food); trials 2 and 3 (food presentation, chow or jelly, counterbalanced); trial 4 (NAcLat→VTA opto-tagging). **h**, Food consumption for REG and HFD mice (\*\*\*$P < 0.001$, 2-way repeated measures ANOVA with Holm–Šídák test;

REG: $n = 8$ mice, HFD: $n = 7$ mice). **i,j**, DLC behavioural motifs for REG (**i**) and HFD (**j**) mice, with example unit firing rates and piezo activity. **k**, Top, $z$-scored average of all recorded action potentials across trials relative to events. Total unit events analysed for each motif to determine whether the unit shows significantly increased response (IR) or decreased response (DR) relative to baseline (average unit waveform in inset). Bottom, sample action potentials during feeding or walking (arrows show event onsets; \*$P = 0.024$, \*\*\*$P = 0.0002$; 2-sided Wilcoxon signed-rank test). **l,m**, Relative $z$-score average of individual NAcLat→VTA units for REG (**l**) and HFD (**m**) mice during different behavioural motifs. Bar graphs show percentage of IR, DR and non-responsive units in each behavioural motif (\*\*\*$P = 0.0002$, 2-sided Chi-squared test for proportions with Bonferroni correction for multiple comparisons; REG: $n = 21$ units from $n = 8$ mice, HFD: $n = 20$ units from $n = 7$ mice). Data are mean ± s.e.m. (error bars or shading).

transported virus carrying Cre recombinase (pseudotyped equine infectious anaemia virus, RG-EIAV-Cre) into the lateral VTA and a Cre-dependent adeno-associated virus (AAV) carrying ChR2 into the NAcLat. The mice were also implanted with a custom-made drivable optoelectrode (optrode) above the NAcLat (Fig. 1e). In these mice, ChR2 expression was largely restricted to the NAcLat and not observed in

the NAc core or NAc medial shell (Extended Data Fig. 1a–d). Two weeks after stereotaxic surgery, mice were randomly split into two cohorts, with one cohort remaining on a regular diet (REG mice; 4% fat, standard mouse chow) and the other being placed on a HFD where both regular (4% fat) and high-fat chow (60% fat) were freely available in the home cage. HFD mice rapidly gained weight when compared with REG mice

(Extended Data Fig. 1e). After 30 days of diet, we recorded the neural activity of opto-tagged NAcLat→VTA cells (examples of opto-tagging in units from REG and HFD mice are shown in Extended Data Fig. 1i–r) during free exploration of an open-field chamber containing calorie-rich (jelly) and low-calorie (chow) foods (Fig. 1f,g). The behaviour of REG and HFD mice was recorded on video and analysed using DeepLabCut (DLC) to identify discrete behavioural motifs (Extended Data Fig. 1f), which included actions such as feeding, touching an empty food cup, rearing, turning and various forms of locomotion at different velocities (Extended Data Fig. 1g,h). A piezo sensor placed under the food cup was used to detect precise feeding event timestamps, which showed a strong correlation with DLC-detected feeding events (Extended Data Fig. 2a–d). Food consumption was measured by weighing the food cups after each session. As expected, REG mice consumed significantly more jelly than chow, whereas HFD mice consumed less jelly overall (Fig. 1h). To analyse neural activity, we quantified unit firing rates before and during the onset of each behavioural motif and classified responses as unchanged (non-responsive), significantly increased (IR type) or significantly decreased (DR type) (Fig. 1i–k). No significant differences were observed in the average time spent in each DLC-detected motif between REG and HFD mice (Extended Data Fig. 2e). Of note, firing rates were negatively correlated with total time spent in each motif, with higher firing rates during motifs with less time spent (for example, jelly and chow feeding or touching the empty cup) and lower firing rates during longer-duration motifs such as locomotion (Extended Data Fig. 2f). Next, we assessed the proportions of classified response types in REG and HFD mice to determine whether they differed between diets. In REG mice, opto-tagged units showed high firing rates during jelly consumption, with the majority of units exhibiting significantly increased responses, whereas other behavioural motifs frequently showed decreased responses. By contrast, opto-tagged units in HFD mice displayed lower firing rates during jelly consumption, with none of the tagged units reaching statistical significance (Fig. 1l,m). Similar results were observed in piezo-based analyses, with increased firing rates during jelly consumption in REG mice and a marked reduction in HFD mice (Extended Data Fig. 2a,b). Non-tagged units also exhibited higher firing rates during jelly consumption compared to other motifs as well as reduced proportions of IR responses in HFD mice compared with REG mice for both DLC- and piezo-based analyses, although the effect size was smaller (Extended Data Fig. 2g–j). Together, these results suggest that increased activity of NAcLat→VTA cells is associated with hedonic feeding, but chronic HFD disrupts this relationship.

## Diet-dependent control of hedonic feeding

Next, we examined whether increased activity in the NAcLat→VTA pathway is sufficient to induce feeding behaviour. We injected AAV-hSyn-ChR2 or AAV-hSyn-eYFP into the NAcLat of C57Bl6 mice and implanted an optical fibre above the VTA (Fig. 2a and Extended Data Figs. 3a,b and 4a,b). Mice were subjected to REG or HFD for four weeks. We then analysed the consumption of different food types (one food type per day) without and with optogenetic stimulation of the NAcLat→VTA pathway. On each experimental day, mice were subjected to habituation and a primed-feeding trial (15 min each; laser off), which were followed by 3 additional trials (15 min each: laser off, laser on (473 nm, 20 Hz), laser off). Mice had free access to each food type during all trials, except the habituation trial, when no food was present (Fig. 2b,c). Although mice could consume each food type to satiating levels already during the primed-feeding trial, we noticed that in this trial, HFD mice showed significantly reduced consumption of high-calorie foods when compared to REG mice (Extended Data Fig. 3c). Even though feeding levels were low after the primed-feeding trial, high-frequency optogenetic stimulation of the NAcLat→VTA pathway in REG mice resulted in significantly increased consumption of

high-calorie foods (jelly, chocolate, peanut butter, butter and high-fat chow). By contrast, optogenetic stimulation of NAcLat→VTA did not change the consumption of low-calorie foods (regular chow) or water (Fig. 2d and Extended Data Fig. 3d), including when the same mice were tested after 24 h of food deprivation (Extended Data Fig. 3e,f). Additionally, stimulating NAcLat→VTA did not modify the feeding of quinine-adulterated butter that holds similar calorie value as butter but with reduced palatability (Extended Data Fig. 3g,h). The mean weights of ChR2 and eYFP REG mice did not differ before and after the series of feeding experiments, indicating that the feeding behaviour in this assay did not affect overall weight gain (Extended Data Fig. 3i). We performed several additional experiments in separate cohorts of REG mice to further examine the effects of manipulating the NAcLat→VTA pathway on hedonic feeding behaviour. First, we applied optogenetic stimulation in an acute feeding assay (equivalent to primed-feeding trial) and found that this was sufficient to increase jelly consumption in ChR2 but not eYFP mice (Extended Data Fig. 3j). Second, to examine whether optogenetic activation of NAcLat cell bodies affects hedonic food consumption, mice were injected with ChR2 into the NAcLat and implanted with two optical fibres (one each above the VTA and NAcLat). In these mice, light stimulation of the VTA, but not NAcLat, increased jelly consumption (Extended Data Fig. 3k–n), suggesting that projection specificity is necessary for inducing hedonic feeding behaviour. Third, we optogenetically silenced the NAcLat→VTA pathway and assessed jelly consumption during the primed-feeding trial on separate days with and without laser stimulation. Optogenetic inhibition significantly reduced hedonic feeding behaviour (Extended Data Fig. 3o–t).

In HFD mice, 20 Hz optogenetic stimulation of the NAcLat→VTA pathway did not promote hedonic or non-hedonic feeding behaviour (Fig. 2e), even when mice were tested after 24 h of food deprivation (Extended Data Fig. 4c,d). Despite the loss of opto-induced feeding behaviour, optogenetic stimulation still induced place preference in these HFD mice (Fig. 2f). However, the effect of HFD was reversible, as the levels of food consumed during both primed-feeding trial (Extended Data Fig. 4e,f) and optogenetic stimulation of the NAcLat→VTA pathway (Fig. 2g) were gradually restored when HFD mice were placed back on a regular diet for at least two weeks, suggesting normalization of hedonic feeding behaviour to similar levels as REG mice. Together, stimulation of the NAcLat→VTA pathway increased hedonic feeding in REG mice but not in HFD mice—the response was restored when HFD mice were returned to a regular diet.

## HFD reduces neurotensin signalling

To investigate the cellular and molecular adaptations triggered by chronic HFD, we used single-cell patch sequencing of NAcLat→VTA cells (Fig. 3a). REG and HFD mice were injected with fluorescent retrobeads into the VTA. One week later, we measured the intrinsic membrane properties and responses to current injections with whole-cell patch-clamp recordings of bead-labelled NAcLat cells before extracting cell cytosol for RNA sequencing. Although the firing rate in response to depolarizing current injections was slightly increased in HFD mice, the comparison did not reach statistical significance, and we also found no differences in the intrinsic membrane properties of NAcLat→VTA neurons in REG and HFD mice (Fig. 3b–f).

We next evaluated diet-induced transcriptomic differences of more than 8,000 genes, and found that 280 genes were downregulated ($P < 0.05$, log-transformed fold change ($\log_2 FC$) $< -1$) and 183 genes were upregulated ($P < 0.05$, $\log_2 FC > 1$) (Extended Data Fig. 5a–e). As expected, NAcLat→VTA cells expressed marker genes for GABAergic (γ-aminobutyric acid-expressing) neurons and *Drd1* mRNA, whereas glutamatergic marker genes and *Drd2* mRNA were detected less frequently. Expression of these genes did not change in response to HFD (Extended Data Fig. 5b,c). When focusing our analysis on genes that

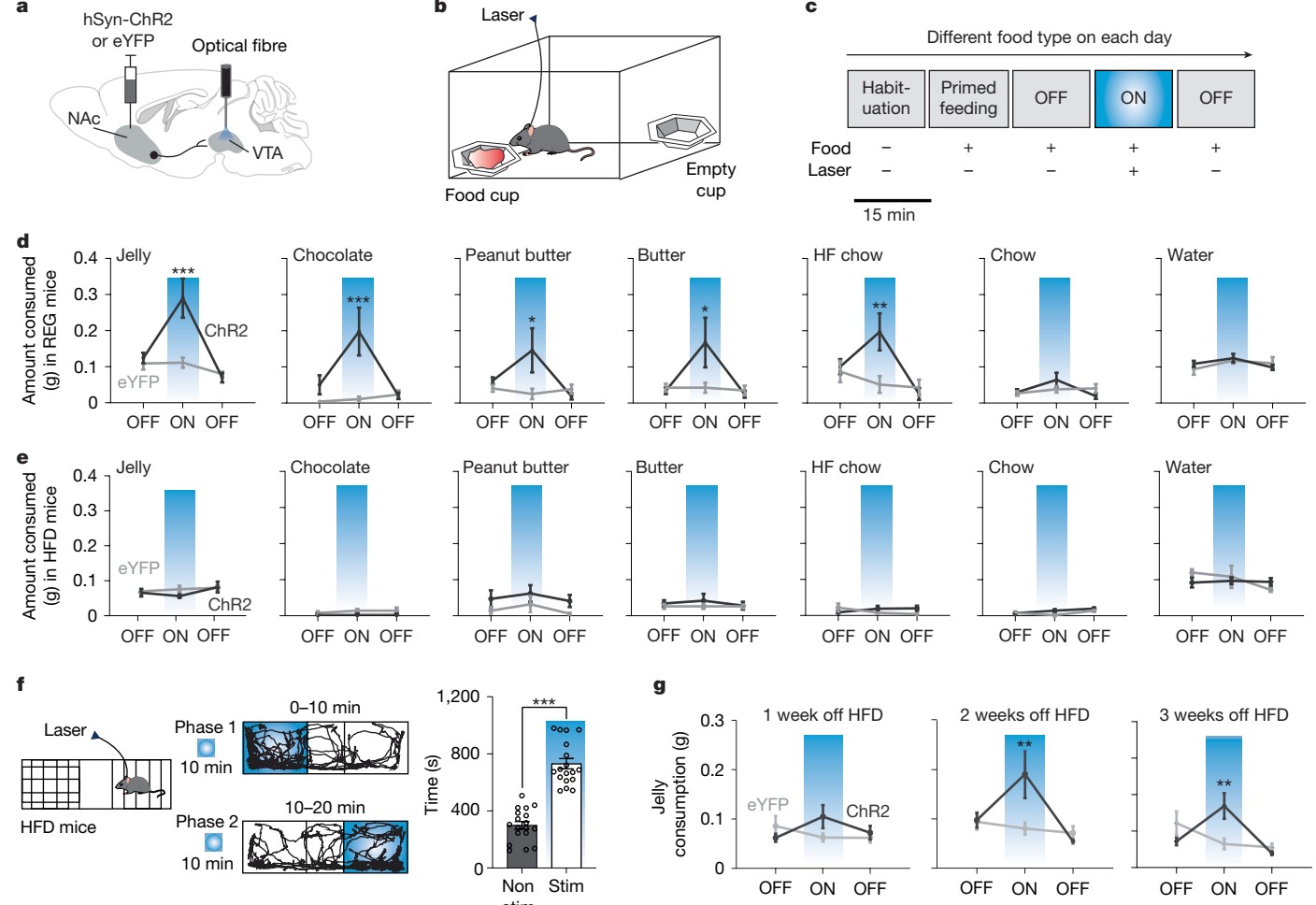

**Fig. 2 | Opto-stimulation of NAcLat→VTA promotes hedonic feeding in REG mice but not in HFD mice. a**, Experimental design. **b**, Acute feeding assay: mice were placed in an open-field chamber containing two cups (one with food and one empty). **c**, Timeline: mice were tested in 5×15-min trials. Food was presented during trials 2 to 5, with laser stimulation (473 nm, 20 Hz, 5 ms) during trial 4. After each trial, the food cup was replaced with a new cup of the same food type, and consumption was analysed. **d**, Consumption during the acute feeding assay for ChR2 and eYFP mice on a REG diet (jelly (ChR2: *n* = 10 mice, eYFP: *n* = 8 mice; ***P = 0.0001), chocolate (ChR2: *n* = 10 mice, eYFP: *n* = 9 mice; ***P = 0.0003), peanut butter (ChR2: *n* = 10 mice, eYFP: *n* = 9 mice; *P = 0.0112), butter (ChR2: *n* = 10 mice, eYFP: *n* = 9 mice; *P = 0.0201), high-fat chow (ChR2: *n* = 9 mice, eYFP: *n* = 9 mice; **P = 0.0032), chow (ChR2: *n* = 10 mice, eYFP: *n* = 9 mice; *P* > 0.05) or water (ChR2: *n* = 7 mice, eYFP: *n* = 9 mice; *P* > 0.05)). Blue indicates laser stimulation of the NAcLat→VTA pathway (2-way repeated

measures ANOVA with Holm–Šídák test). **e**, Food consumption during the acute feeding assay for HFD mice (*P* > 0.05; ChR2: *n* = 10 mice, eYFP: *n* = 5 mice, 2-way repeated measures ANOVA with Holm–Šídák test). **f**, Real-time place preference: HFD mice received NAcLat→VTA stimulation (473 nm, 20 Hz, 5 ms pulses) upon entry into one compartment of a 3-chamber apparatus. The paired side was switched after 10 min. Sample trajectories show movement during each phase. HFD mice spent significantly more time in the light-paired compartment (stim) compared to the unpaired compartment (non-stim) (***P < 0.001, 2-sided paired Student's *t*-test; *n* = 18 mice). **g**, Food consumption during the acute feeding assay for mice removed from HFD and returned to a regular diet, tested at different time points (2 weeks off HFD: **P = 0.0075, 3 weeks off HFD: **P = 0.0041, 2-way repeated measures ANOVA with Holm–Šídák test; ChR2: *n* = 8 mice, eYFP: *n* = 7 mice). Data are mean ± s.e.m.

have been associated with synaptic signalling as well as feeding behaviour, we found that *Nts* (encoding neurotensin, a neuropeptide involved in feeding behaviour[27–29]) was strongly expressed in NAcLat→VTA cells compared with other members of these gene families (the median expression of *Nts* was in the 98th percentile of observations from 587 genes). *Nts* was detected in 95% of NAcLat→VTA cells obtained from REG mice and in 90% of the cells obtained from HFD mice (Extended Data Fig. 5f). The overlap between *Nts* expression and projection target was also confirmed by additional retrograde tracing experiments in which cholera toxin subunit B (CTB) was injected into the VTA of NTS-Cre mice crossed to an Ai14 reporter mouse line. We found that approximately 75% of the CTB-labelled cells also express *Nts* (Extended Data Fig. 5g,h), further confirming that NTS is enriched in the NAcLat→VTA pathway. However, it is possible that NTS-expressing NAcLat neurons also project to other brain structures.

Transcriptomic difference analysis revealed that in HFD mice, *Nts* expression levels were reduced compared with REG mice (log₂FC = −1.52) (Fig. 3g–j). This reduction in *Nts* expression following HFD was further confirmed using in situ hybridization in a separate cohort of mice. Of note, *Nts* expression levels seemed to recover if HFD mice were returned to a regular diet for three weeks (Extended Data Fig. 5i–m).

We next sought to determine whether diet-induced reduction in *Nts* mRNA expression results in measurable changes in NTS release. To do this, we utilized a novel GPCR fluorescent sensor, ntsLight1.1, whose structure mimics the neurotensin receptor 1 (NTSR1) and increases fluorescence when NTS is bound. We validated the sensitivity and specificity of ntsLight1.1 for NTS in both cultured neurons and ex vivo brain-slice preparations (Extended Data Fig. 6a–f). To test whether NTS release is reduced in HFD mice, we injected ntsLight1.1 into the VTA and AAV-hSyn-ChrimsonR-tdTomato (Chrimson) into the NAcLat

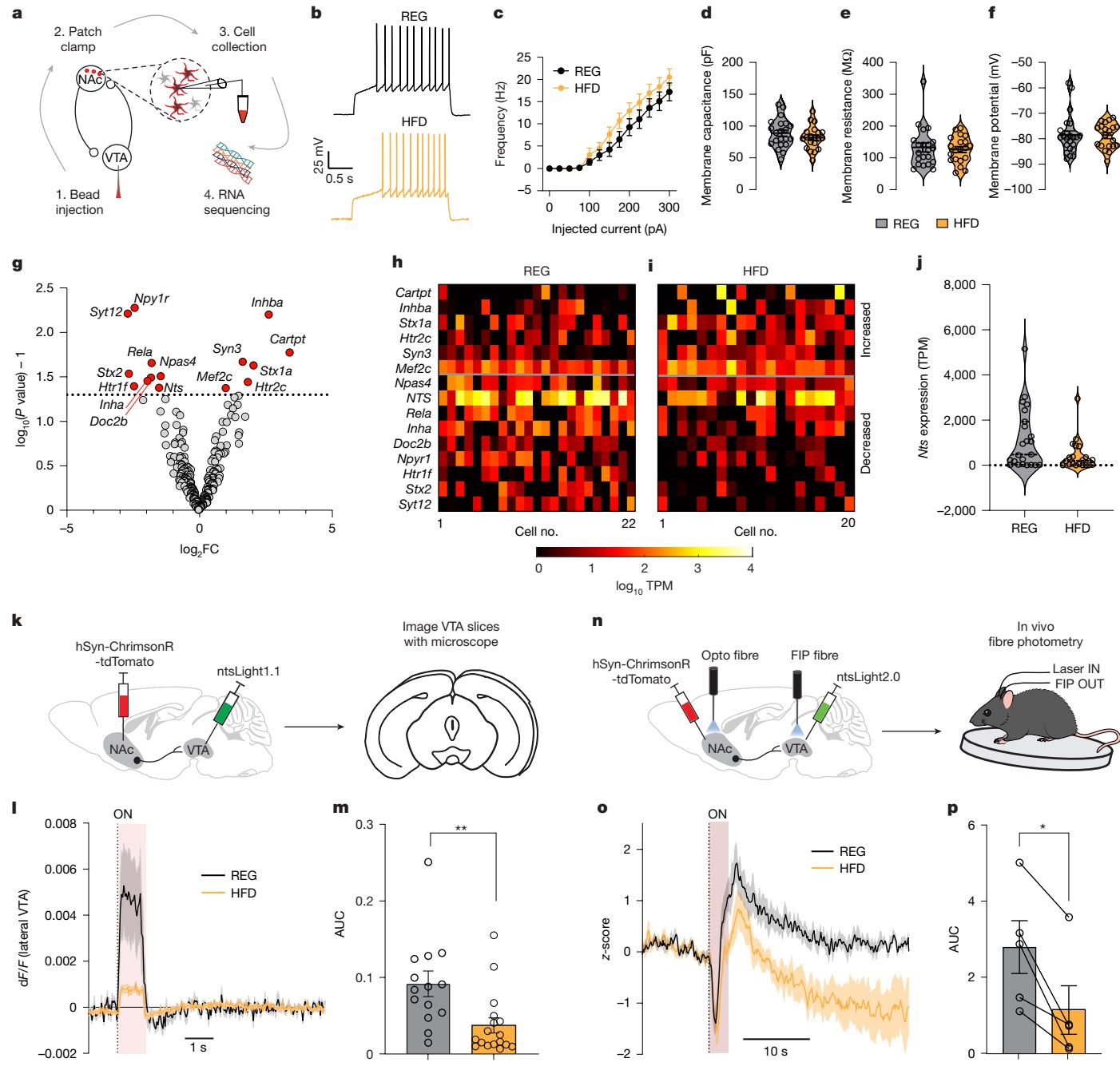

**Fig. 3 | Reduced NAcLat → VTA NTS expression and release in HFD mice.**
**a**, Patch-seq experimental design. **b**, Example current injection (150 pA) in bead-labelled NAcLat→VTA cells from REG (top) and HFD (bottom) mice. **c**–**f**, Electrophysiological properties of NAcLat→VTA cells from REG and HFD mice. **c**, Firing frequency ($P > 0.05$, 2-way repeated measures ANOVA; REG: $n = 23$ cells, $n = 10$ mice; HFD: $n = 19$ cells, $n = 11$ mice). **d**, Membrane capacitance ($P > 0.05$, unpaired Student's $t$-test; REG: $n = 21$ cells, $n = 10$ mice; HFD: $n = 20$ cells, $n = 11$ mice). **e**, Membrane resistance ($P > 0.05$, unpaired Student's $t$-test; REG: $n = 21$ cells, $n = 10$ mice; HFD: $n = 20$ cells, $n = 11$ mice). **f**, Resting membrane potential ($P > 0.05$, 2-sided unpaired Student's $t$-test; REG: $n = 23$ cells, $n = 10$ mice; HFD: $n = 20$ cells, $n = 11$ mice). **g**, Volcano plot of differential gene expression in NAcLat→VTA cells between REG and HFD mice. Red data points indicate significantly different genes (absolute value of $\log_2$FC > 1 and $P < 0.05$). Statistical significance was determined using a two-sided hypothesis. Values were not corrected for multiple comparisons. **h**,**i**, Heat maps showing relative

expression of synaptic and feeding-related genes in individual NAcLat→VTA cells from REG (**h**) and HFD (**i**) mice. TPM, transcripts per million. **j**, Violin plot of *Nts* gene expression in NAcLat→VTA cells from REG and HFD mice (REG: $n = 23$ cells, $n = 10$ mice; HFD: $n = 20$ cells, $n = 11$ mice). **k**–**m**, ntsLight1.1 experiment: AAV-hSyn-Chrimson-tdTomato and AAV9-hSyn-ntsLight1.1 were injected into the NAcLat and VTA, respectively. Optogenetic stimulation in acute brain slices revealed reduced ntsLight1.1 fluorescence in HFD compared with REG mice (\*\**P* = 0.0059, 2-sided unpaired Student's $t$-test; REG: $n = 14$ slices, $n = 5$ mice; HFD: $n = 17$ slices, $n = 4$ mice). **n**, In vivo opto-photometry experiment using ntsLight2.0. FIP, fibre photometry. **o**, ntsLight2.0 recorded in VTA 5 s before and 20 s after 3-s opto-stimulation of NAcLat cells from the same mice on REG or HFD ($n = 5$ mice, $z$-score average for $n = 30$ trials). **p**, Area under the curve (AUC) during the 3–5 s interval for mice on REG or HFD (\**P* = 0.0125, 2-sided paired Student's $t$-test; $n = 5$ mice). Data are mean ± s.e.m. (error bars or shading).

of mice placed on REG or HFD for 6 weeks. We recorded ntsLight1.1 in VTA slices and found that optogenetic stimulation of NAcLat terminals resulted in significantly less ntsLight1.1 fluorescence in HFD mice compared with REG mice (Fig. 3k–m and Extended Data Fig. 6g,h), indicating that NTS release is reduced in HFD mice ex vivo. Although ntsLight1.1 provided robust readouts in brain-slice recordings, its sensitivity was limited for reliably detecting signals in vivo. To address this, we developed ntsLight2.0, which exhibited enhanced sensitivity, making it more effective for in vivo applications (Extended Data Fig. 7a–x). We injected ntsLight2.0 into the VTA and Chrimson into the NAcLat, implanting optical fibres above the NAcLat and in the VTA. Four weeks later, we recorded ntsLight2.0 signals in the VTA using fibre photometry in head-fixed mice during optogenetic stimulation of NAcLat cells. Stimulation produced bidirectional ntsLight2.0 transients in the VTA composed of a fast negative peak that occurred during laser stimulation (0–2 s) followed by a slower positive peak (3–5 s). The slower positive, but not the fast negative peak, was blocked by intraperitoneal injection of an NTSR1 antagonist (SR48692, 5 mg kg$^{-1}$; Extended Data Fig. 7q–u), suggesting that laser stimulation produces a brief artefact that is followed by an increase of NTS release. Notably, after mice were on a HFD for 4 weeks, the increase in ntsLight2.0 signals triggered by NAcLat stimulation was significantly reduced only in the 3–5 s interval (Fig. 3n–p and Extended Data Fig. 7y–ab). Together, these experiments suggest that HFD reduces NTS expression and release in the NAcLat→VTA pathway.

## NTS is necessary for hedonic feeding

To test whether optogenetic stimulation of NTS-expressing NAcLat neurons is sufficient for inducing hedonic feeding behaviour, we expressed Cre-dependent ChR2 in NAcLat neurons of NTS-Cre mice and implanted an optical fibre above the VTA of mice kept on a regular diet. Optogenetic stimulation of NAcLat terminals in the VTA in an acute feeding assay (same as in Fig. 2) increased hedonic feeding without affecting general locomotor activity (Extended Data Fig. 8a–f).

Next, we sought to determine whether NTS expression in the NAcLat is necessary for inducing hedonic feeding behaviour. We performed a conditional knockout of NTS in NAcLat neurons by injecting AAV-hSyn-Cre into the NAcLat of *Nts*$^{flox}$ mice kept on a regular diet. These mice were also injected with AAV-hSyn-ChR2 into the NAcLat, and an optical fibre was implanted above the VTA. Control mice were treated identically but did not receive AAV-hSyn-Cre injection (Fig. 4a). In situ hybridization experiments, performed in a separate cohort of mice, confirmed a significant reduction of NTS expression in the NAcLat of *Nts*-knockout mice compared with control mice (Fig. 4b,c). We then optogenetically stimulated NAcLat terminals in the VTA in the acute feeding assay. As expected, control mice showed increased jelly consumption. However, in *Nts*-knockout mice, optogenetic stimulation of the NAcLat→VTA pathway did not increase hedonic feeding, indicating that NTS expression in the NAcLat is necessary for this behaviour (Fig. 4d,e and Extended Data Fig. 8g).

Finally, to test whether NTS receptor activation in the VTA is necessary for hedonic feeding behaviour, we implanted an infusion cannula with an optical fibre above the VTA of mice expressing ChR2 or eYFP in the NAcLat (Fig. 4f,g and Extended Data Fig. 8i,j). We then assessed jelly consumption in the acute feeding assay with the addition of infusions of an NTS receptor antagonist (SR142948A, 6 mM) or saline into the VTA before optogenetic stimulation of the NAcLat→VTA pathway (Fig. 4h). As expected, we found that following saline infusion, optogenetic stimulation of NAcLat→VTA increased jelly consumption in ChR2-expressing but not eYFP-expressing mice. By contrast, intra-VTA infusion of SR142948A prevented the increase in jelly consumption in response to NAcLat→VTA stimulation (Fig. 4i and Extended Data Fig. 8h), suggesting that functional NTS receptors are necessary for optogenetic-induced hedonic feeding behaviour.

## HFD reduces excitation of dopamine neurons

Previous studies have shown that NAcLat neurons synapse onto local VTA GABAergic neurons and potentially disinhibit dopamine neurons to promote reward-related behaviours[26]. However, in addition to disinhibition, VTA dopamine neurons may also be directly excited through NTS binding to NTSR1 following its release from NAcLat terminals, as demonstrated for NTS inputs from the lateral hypothalamus to VTA[30,31]— a circuit that is known to have a critical role in the regulation of feeding behaviour and obesity[32,33]. To test this possibility, we performed brain-slice patch-clamp recordings of NAcLat-projecting dopamine neurons and optogenetically stimulated ChR2-expressing NAcLat terminals in the VTA (Fig. 4j,k). Consistent with previous studies[26], we found that the firing rate of these cells increased during light stimulation. However, when VTA slices were bathed in solution containing an NTS receptor antagonist (SR142948A, 1 μM), the firing rate did not increase, suggesting that excitation of dopamine neurons following stimulation of NAcLat inputs involves binding of NTS to its receptor. Notably, in mice subjected to 4 weeks of HFD, optogenetic-induced excitation of dopamine neurons was already absent in the saline condition (Fig. 4l,m).

## No postsynaptic changes following HFD

It is possible that chronic HFD disrupts NTS signalling in the NAcLat→VTA pathway not only pre-synaptically, but also post-synaptically. However, additional experiments suggest that postsynaptic mechanisms are unlikely. First, bath application of NTS in brain-slice perforated-patch recordings increased firing of VTA dopamine neurons in both REG and HFD mice (Extended Data Fig. 9a–c). Conversely, the firing rate of VTA GABA neurons remained unaffected by NTS application (Extended Data Fig. 9d–f), which is consistent with previous studies suggesting that these cells lack NTSR1[30,34]. Second, we found no significant difference between REG mice and HFD mice in the expression of *Ntsr1* mRNA in VTA dopamine neurons (Extended Data Fig. 9g–i). Third, we found no significant differences in the intrinsic excitability and membrane properties of NAcLat-projecting dopamine neurons (Extended Data Fig. 9j–q). Thus, HFD-induced changes in NTS function are likely to occur pre-synaptically and involve downregulation of NTS expression and release, while NTS receptors remain functional.

## Enhancing NTS restores hedonic feeding

We argued that enhancing NTS release could alleviate the observed changes in hedonic feeding behaviour following chronic HFD and potentially affect other metrics that shape obesity progression such as home cage feeding[8], weight gain[35] and mobility[36]. To increase NTS expression in HFD mice, we developed a Cre-dependent virus to overexpress NTS (ssAAV-9/2-shortCAG-dlox-mNts(rev)-dlox-WPRE-SV40p(A) (AAV-NTS-OE)). We injected this virus bilaterally into the NAcLat together with RG-EIAV-Cre into the VTA to drive *Nts* mRNA overexpression only in NAcLat cells that project to the VTA. To assess NTS overexpression, we detected *Nts* mRNA using fluorescent in situ hybridization and observed increased *Nts* expression in the NAcLat of HFD mice injected with AAV-NTS-OE (NTS-OE mice) compared with mice injected with a control virus encoding mCherry (Extended Data Fig. 10a–c). To confirm whether AAV-NTS-OE also increased NTS release in HFD mice, we injected ntsLight1.1 into the VTA and Chrimson into the NAcLat of mice on HFD in combination with AAV-NTS-OE or AAV-mCherry. We observed greater NTS release in HFD mice injected with AAV-NTS-OE compared with those injected with AAV-mCherry in response to optogenetic stimulation of the NAcLat→VTA pathway (Fig. 5a–c and Extended Data Fig. 10d).

Next, we sought to determine whether NTS overexpression could rescue the HFD-induced loss of hedonic feeding behaviour in response

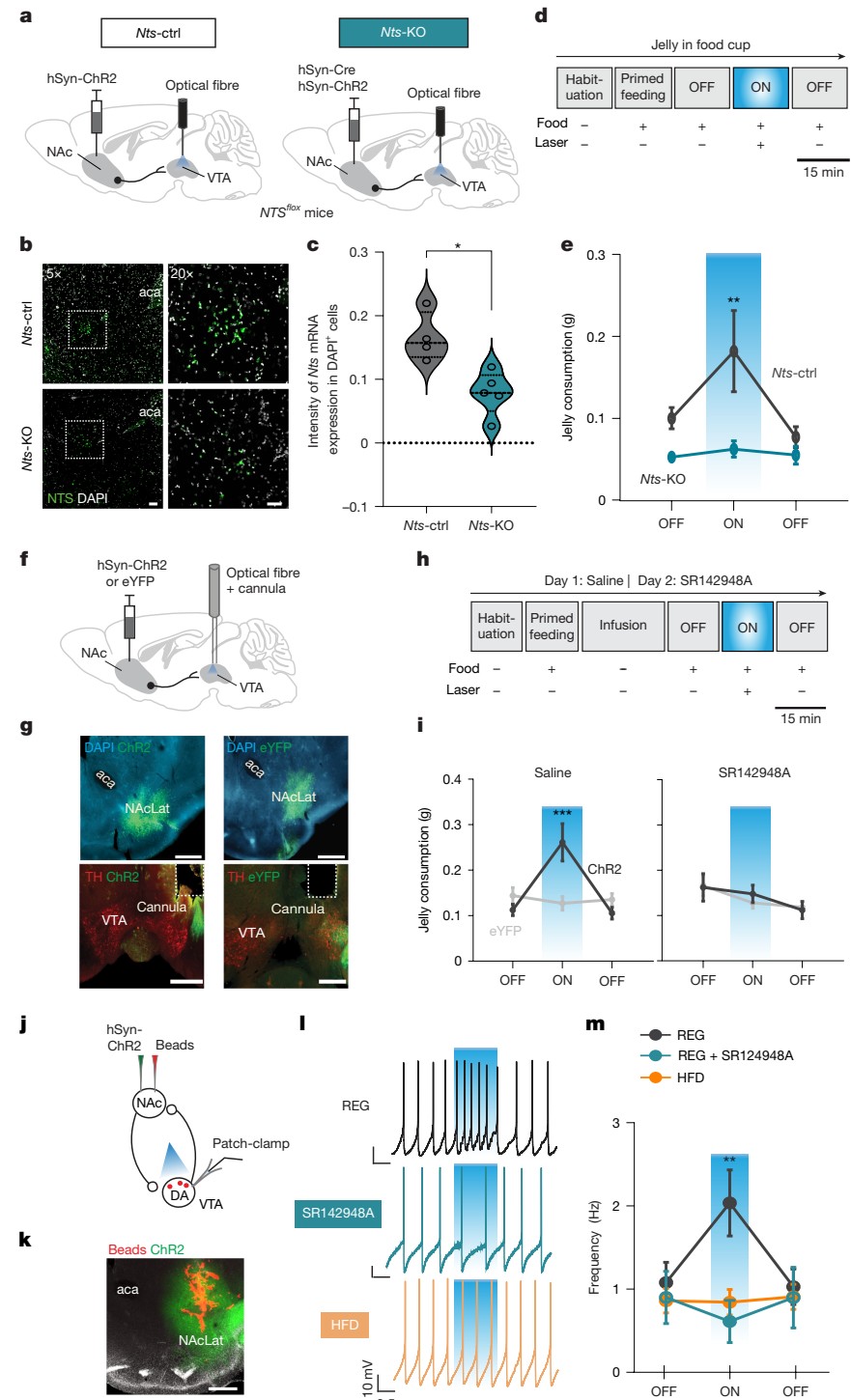

**Fig. 4 | NAcLat→VTA NTS is necessary for hedonic feeding and dopamine cell excitation. a**, AAV-hSyn-ChR2 was injected alone (*Nts*-ctrl) or with AAV-hSyn-Cre (*Nts*-KO) into NAcLat of *Nts*flox mice, with an optical fibre above VTA in REG mice. **b**, Fluorescent in situ hybridization images in NAcLat of *Nts*-ctrl and *Nts*-KO mice. Outlined regions are magnified on the left. aca, anterior commissure. Scale bars: 200 μm (5×), 50 μm (20×). **c**, *Nts* expression is significantly reduced in *Nts*-KO mice compared with *Nts*-ctrl mice (*$P$ = 0.0159, 2-sided Mann–Whitney test; *Nts*-ctrl: $n$ = 4 mice, *Nts*-KO: $n$ = 5 mice). **d**, Timeline of acute feeding assay. **e**, NAcLat→VTA opto-stimulation significantly increases jelly consumption in *Nts*-ctrl mice compared with *Nts*-KO mice (**$P$ = 0.0041, 2-way repeated measures ANOVA with Holm–Šídák test; *Nts*-ctrl: $n$ = 10 mice, *Nts*-KO: $n$ = 8 mice). **f**, Opto-pharmacology experiment in REG mice. **g**, Fluorescence images showing ChR2 or eYFP expression in the NAcLat (top) and VTA (bottom). Scale bars, 500 μm. **h**, On day 1, jelly was presented over 4 trials, with saline infused into VTA

before the first OFF trial. On day 2, SR142948A (6 mM, 500 nl) was infused instead. Laser stimulation (20 Hz, 5 ms) during ON trial. **i**, Jelly consumption by ChR2 and eYFP mice following saline (left) or SR142948A (right) infusion (***$P$ < 0.001, 2-way repeated measures ANOVA with Holm–Šídák multiple comparisons test; ChR2: $n$ = 11 mice, eYFP: $n$ = 15 mice). **j**, Whole-cell patch-clamp recordings of NAcLat-projecting VTA dopamine neurons (DA) during stimulation of NAcLat terminals in VTA. **k**, Fluorescent retrobeads and ChR2 expression in NAcLat. Scale bar, 500 μm. **l**, Firing in dopamine neurons from a REG mouse recorded in artificial cerebrospinal fluid (ACSF; top) or ACSF containing SR142948A (1 μM, middle) or from a HFD mouse recorded in ACSF (bottom). Blue shaded area indicates light stimulation (1 s, 20 Hz 5 ms pulses). **m**, Firing rates of dopamine neurons under conditions in **l** (**$P$ = 0.002, 2-way repeated measures ANOVA with Holm–Šídák test; REG: $n$ = 26 cells, $n$ = 12 mice; REG + SR142948A: $n$ = 12 cells, $n$ = 3 mice; HFD: $n$ = 21 cells, $n$ = 5 mice). Data are mean ± s.e.m.

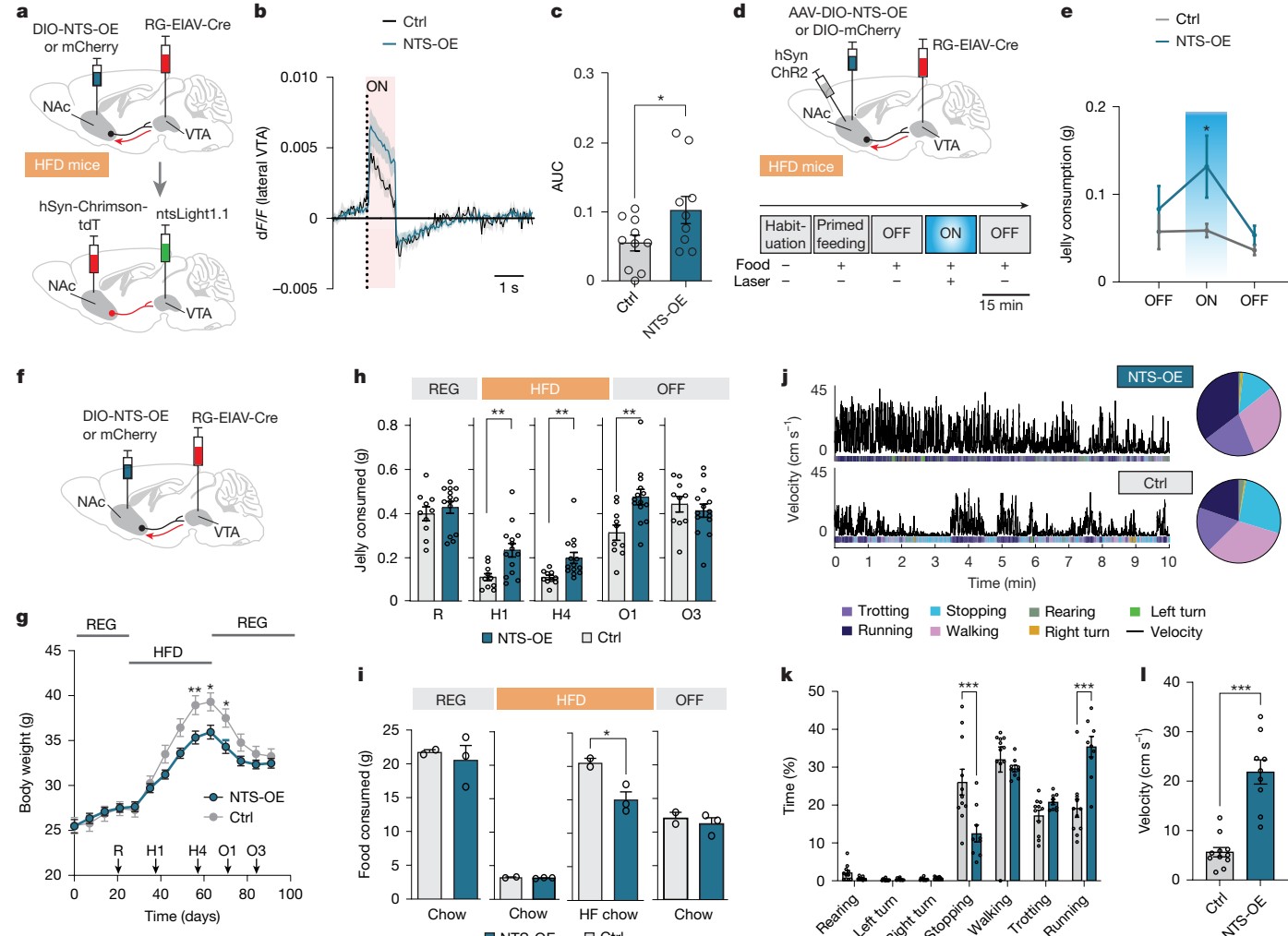

**Fig. 5 | NTS overexpression in NAcLat→VTA mitigates HFD-induced behavioural adaptations. a**, A retrograde virus (RG-EIAV-Cre) was injected into the VTA, and either Cre-dependent AAV for NTS overexpression (NTS-OE) or mCherry (ctrl) was injected into the NAcLat of HFD mice. AAV-hSyn-Chrimson-tdTomato and AAV9-hSyn-ntsLight1.1 were also injected, and acute VTA slices were prepared six weeks later to record ntsLight1.1. **b**, ntsLight1.1 activity during opto-stimulation of NAcLat inputs in NTS-OE and mCherry mice on HFD. **c**, AUC of ntsLight1.1 activity shows significantly higher NTS release in NTS-OE mice (*$P = 0.0397$, 2-sided unpaired Student's $t$-test; REG: $n = 10$ slices from $n = 2$ mice; HFD: $n = 9$ slices from $n = 2$ mice). **d**, Experimental design and timeline. **e**, Jelly consumption for NTS-OE and mCherry HFD mice with opto-stimulation of NAcLat→VTA (*$P = 0.0309$, 2-way repeated measures ANOVA with Holm–Šídák multiple comparisons test; NTS-OE: $n = 6$ mice, nCherry: $n = 8$ mice). **f**, Experimental design. **g**, Body weight in NTS-OE and mCherry mice on REG, HFD, and when returned to REG. Arrows indicate time points of the assays in **h**–**l**: REG (R), 1 week on HFD (H1), 4 weeks on HFD (H4), 1 week off HFD (O1) and 3 weeks off HFD (O3) (day 56: **$P = 0.0079$, day 63: *$P = 0.0164$, day 70: *$P = 0.0256$, 2-way repeated measures ANOVA with Holm–Šídák multiple comparisons test; NTS-OE: $n = 14$ mice, mCherry: $n = 10$ mice). **h**, Mean jelly consumption during acute feeding assays at different time points shown in **g** (H1: **$P = 0.0049$, H4: **$P = 0.0084$, O1: **$P = 0.0037$, 2-sided unpaired Student's $t$-test at each timepoint; mCherry: $n = 10$ mice, NTS-OE: $n = 14$ mice). **i**, Weekly home cage consumption of chow and high-fat chow during REG, 4 weeks of HFD and when returned to REG for 3 weeks (OFF) (*$P = 0.0397$, 2-sided unpaired Student's $t$-test at each timepoint; NTS-OE: $n = 3$ cages, mCherry: $n = 2$ cages, consumption was normalized to a single mouse). **j**–**l**, Open-field behaviour in HFD NTS-OE and mCherry mice. Behaviour was assessed at timepoint H4. **j**, Velocity (top) and individual motifs as indicated by colour code (bottom) and percentage of time spent in each motif (right). **k**, Time spent in motifs (***$P < 0.001$, 2-way repeated measures ANOVA with Holm–Šídák multiple comparisons test; mCherry: $n = 11$ mice, NTS-OE: $n = 9$ mice). **l**, Velocity (***$P < 0.001$, 2-sided unpaired Student's $t$-test). Data are mean ± s.e.m. (error bars or shading).

to optogenetic stimulation of the NAcLat→VTA pathway. To do this, we injected AAV-hSyn-ChR2 into the NAcLat and implanted an optical fibre above the VTA in a separate cohort of HFD mice injected with AAV-NTS-OE or AAV-mCherry (Fig. 5d). We found that HFD NTS-OE mice consumed significantly more jelly during the primed-feeding trial compared to HFD AAV-mCherry mice (Extended Data Fig. 10e). Furthermore, optogenetic stimulation of the NAcLat→VTA pathway increased jelly consumption in HFD NTS-OE mice but not in HFD AAV-mCherry mice (Fig. 5e).

To investigate whether NTS overexpression affects other aspects of diet-induced obesity, we subjected additional cohorts of NTS-overexpressing and mCherry control mice to a regular diet for 28 days.

During this period, the body weight of NTS-overexpressing and mCherry mice remained similar. However, when switched to an HFD, the control mice rapidly gained weight, whereas NTS-overexpressing mice gained significantly less weight—a difference that normalized when the mice were returned to a regular diet (Fig. 5f,g). The effects of NTS overexpression on weight gain were robust and reproducible, as confirmed in two independent cohorts of mice under varying experimental conditions (baseline body weight, housing scheme and virus expression time) (Extended Data Fig. 10f). Additionally, we assessed jelly consumption in the acute feeding assay at different time points: on a regular diet, during HFD exposure (after 1 and 4 weeks on HFD), and after returning to a regular diet (OFF HFD, after 1 and 3 weeks on

REG). NTS-overexpressing mice consumed significantly more jelly than mCherry controls starting from the first week of HFD, with this effect persisting for a week after returning to a regular diet (Fig. 5h). These findings suggest that NTS overexpression may attenuate the reduction in the hedonic value of high-calorie foods typically observed after HFD exposure. To further understand the behavioural changes contributing to reduced weight gain on HFD, we examined weekly consumption of chow and high-fat chow. We found that home cage consumption of high-fat chow was significantly reduced in NTS-overexpressing mice compared with mCherry mice after four weeks on HFD, whereas chow consumption remained unchanged at all time points (Fig. 5i).

We also analysed the body temperature and open-field behaviour of NTS-overexpressing and mCherry mice. Whereas body temperature did not differ between the two groups (Extended Data Fig. 10g), DLC-based behavioural motif analysis revealed that NTS-overexpressing mice spent more time in high-velocity motifs (such as fast walking and running) and exhibited increased locomotor activity (Fig. 5j–l and Extended Data Fig. 10i,j). Additionally, NTS-overexpressing mice spent more time in the centre of the open-field chamber, which may reflect reduced anxiety-related behaviours, as increased anxiety is often associated with diet-induced obesity[37] (Extended Data Fig. 10k). Together, our results demonstrate that selective overexpression of NTS in the NAcLat→VTA pathway can restore NTS signalling to normalize multiple aspects of diet-induced obesity, including weight gain, hedonic feeding behaviour and locomotor activity.

## Discussion

The increased availability of high-calorie foods has been closely linked to the rising prevalence of obesity[9], prompting the critical question of how continuous access to calorie-rich foods affects neural circuits involved in feeding and motivation. Here, we demonstrate that chronic HFD consumption alters hedonic feeding behaviours and disrupts signalling in the NAcLat→VTA pathway, specifically involving the neuropeptide NTS, influencing the progression of obesity.

### NTS signalling in hedonic feeding

Our finding that NTS release from NAcLat neurons decreases after chronic HFD provides a circuit-level explanation for the link between dopamine activity and weight gain, altered feeding behaviour and obesity progression[2,19–24]. NTS acts through NTSR1 to enhance VTA dopamine neuron activation and increase dopamine release in the NAc[30,31], where dopamine is a key regulator of reward learning and motivated behaviour[15,38]. Thus, HFD-induced reductions in NTS release from NAcLat terminals are likely to diminish dopamine excitation, reducing the desire to consume high-calorie foods.

In addition to NTS-induced excitation, dopaminergic cells in the VTA are activated via disinhibition mediated by VTA GABAergic neurons targeted by NAcLat inputs[26]. Our findings align with recent studies on lateral hypothalamus inputs to the VTA[31], suggesting that NTS-induced excitation and disinhibition are both essential components of the in vivo function of this circuit.

Stimulation of NTS-expressing neurons in the NAcLat→VTA pathway and in other regions, such as the central and extended amygdala, promotes hedonic food and liquid consumption[29,39]. However, in brain regions such as the lateral septum[40], hypothalamus[41] and nucleus of the solitary tract[42], NTS-expressing neurons suppress feeding behaviour. Moreover, direct infusion of NTS into the VTA has been shown to suppress feeding in food-deprived mice[43]. Several factors may explain these contrasting effects of NTS on feeding: (1) different sources of NTS innervation may target distinct subpopulations of VTA neurons, leading to variable behavioural outcomes. For example, NTS from the lateral hypothalamus has anorectic effects[30]. (2) NTS may bind to NTSR2, altering intracellular calcium levels in astrocytes[44] and

indirectly modulating dopamine signalling. (3) The activation of signalling cascades by NTSR1 exhibits variability depending on the specific molecules involved. This diversity in signalling cascades is based on the ability of NTSR1 to interact with distinct G proteins and recruit different intracellular effectors[45], which could produce varying effects on feeding. (4) Excessive NTS levels could desensitize NTSR1[46], reversing its typical effects.

While our results highlight the critical role of NTS in hedonic feeding, further work is needed to clarify how NTS release from different sources is integrated in the VTA to modulate feeding behaviours.

### Devaluation of high-calorie foods

We show that chronic HFD exposure leads to substantial alterations in hedonic feeding behaviour. Previous studies have reported reduced consumption of high-calorie liquids and foods in no-effort paradigms[3–7]. This pattern is likely to contribute to broader motivational impairments observed in rodents following chronic HFD, including reduced responses in operant self-administration of hedonic foods[47], a diminished drive to work for food, impaired learning and reduced conditioned place preference when food reward serves as an unconditioned stimulus[2,48]. Similarly, human studies suggest that obesity and chronic HFD consumption reduce sensitivity to the rewarding effects of palatable foods, as evidenced by blunted activity in reward-related brain regions, such as the striatum and prefrontal cortex[49,50].

The most parsimonious explanation that would account for this general reduction in hedonic feeding drive across different paradigms is that chronic HFD leads to a devaluation of calorie-rich foods. This explanation has been suggested previously[2–4,51] and is supported by other cases where the perceived hedonics of food is deliberately altered. Examples include reduced pleasantness after food aversion[52] or recent consumption (sensory-specific satiety[53]) and increased hedonic value of food during periods of food deprivation[54], where a single change in reward value alters motivational drive across different paradigms. Nonetheless, it is also conceivable that mice are in a general state of reduced motivation owing to the delay in obtaining the food in these tasks[55]. Nevertheless, this does not explain why mice show reduced sucrose preference and will not consume hedonic foods in behavioural paradigms that require no effort[3–7]. Alternatively, the high-calorie foods offered to the mice during the acute feeding assay might not be perceived as appetitive as their high-fat chow in their home cages[7,47]. However, this does not explain why the changes in hedonic feeding behaviour initially persist when mice are placed back on a regular diet as shown in Fig. 2 and by others[5].

Although we propose that chronic HFD results in a devaluation of food hedonics, it may be challenging to comprehend why mice will continue to prefer high-fat foods over regular chow in their home cages and reach an unhealthy body weight. One possible explanation is rooted in the dual-system theory, which suggests that action control can be predominantly habitual or goal-directed[56]. Obesity is often associated with reduced behavioural sensitivity to changes in the motivational value of hedonic food rewards, indicating habit-like behavioural control that encourages overconsumption of food[57]. Another explanation is that devaluation of hedonic foods modifies feeding habits and circadian rhythm[58,59], giving rise to obesogenic feeding patterns. Finally, a reduced drive to consume hedonic food may involve changes in locomotion or reduced exploration behaviour[60], which may indirectly promote weight gain.

Our results demonstrate that overexpression of NTS mitigates HFD-induced changes in hedonic feeding, anxiety, mobility, and home cage food consumption. While the primary driver of weight gain differences between NTS-OE and control mice remains uncertain, each of these factors has a critical role in obesity progression. Reduced anxiety, in particular, may not only improve food consumption behaviours but also enhance overall mental health, which is highly relevant for treating obesity and its comorbidities[37].

## Conclusion

In summary, we identify a circuit mechanism through which NTS regulates hedonic feeding and demonstrate how disruptions in NTS signalling contribute to disordered consumption of calorie-rich foods. Given the role of these foods in driving the obesity epidemic, targeting NTS signalling in the NAcLat→VTA pathway may offer a promising strategy to regulate food intake and support healthy weight maintenance without disrupting other essential NTS-mediated functions.

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

## Methods

### Subjects

The following mouse lines (6–8 weeks old, male and female) were used for experiments: C57BL/6 J mice (Jackson Laboratory, 000664), NTS-Cre (Jackson Laboratory, 017525, strain code: Nts[tm1(cre)Mgmj]), GAD2-Cre (Jackson Laboratory, 010802, strain code: Gad2[tm2(cre)Zjh]/J), Ai14 (Jackson Laboratory, 007914, strain code: B6.Cg-Gt(ROSA)26Sor[tm14(CAG-tdTomato)Hze]/J), Nts[flox] (Jackson Laboratory, 036262, strain code: B6;FVB-Nts[em1Evdr]/J). Mice were housed on a 12 h:12 h light cycle (lights on at 07:00) and a room temperature of 22–25 °C and 55% humidity. All procedures complied with the animal care standards set forth by the National Institutes of Health and were approved by University of California Berkeley's Administrative Panel on Laboratory Animal Care.

### Diet protocols

Mice subjected to a regular diet (REG) had ad libitum access to standard chow (5053 PicoLab Rodent Diet 20, Lab Diet) in their home cage. Mice subjected to a high-fat diet (HFD) mice had ad libitum access to both standard chow and 60% fat chow (Research Diets D12492) for a minimum of 4 weeks prior to experiments. The body weight of all mice was assessed at least once per week.

### Stereotaxic surgeries

Surgeries were performed under general ketamine–dexmedetomidine anaesthesia using a stereotaxic instrument (Kopf Instruments, Model 1900). For retrograde tracing, mice were injected unilaterally with fluorescent retrobeads (100 nl; LumaFluor) or cholera toxin b subunit (400 nl; Fisher Scientific) into the nucleus accumbens (NAc) lateral shell (NAcLat; bregma: 1.0 mm, lateral: 1.9 mm, ventral: −4.3 mm) or ventral tegmental area (VTA; bregma: −3.3 mm, lateral: 0.4 mm, ventral: −4.5 mm) using a 1 µl Hamilton syringe (Hamilton).

The AAVs used in this study were from the Wilson laboratory (pENN. AAV.hSyn.Cre.WPRE.hGH; ~10[13] infectious units per ml, prepared by AddGene), the Deisseroth laboratory (AAV5-EF1α–DIO-hChR2(H134R)-eYFP; AAV5-EF1α-DIO-mCherry; AAV5-hSyn-hChR2(H134R)-eYFP; AAV5-Ef1α-DIO-eYFP; AAV5-hSyn-eYFP; ~10[12] infectious units per ml, prepared by the University of North Carolina Vector Core Facility), from the Boyden laboratory (AAV5-Syn-ChrimsonR-tdTomato; AAV5-CaMKII-ArchT-GFP; ~10[12] infectious units per ml), from the Tian laboratory; ntsLight1.1: The NTS sensor gene was cloned into a hSyn promoter and WPRE enhancer containing SV40 poly (A) signal sequence. The vector was packaged into an adeno-associated virus serotype 9 (AAV9-ntsLight1.1-WPRE-SV40p(A), physical titre 6.45 × 10[14] vg ml[−1], produced by the Viral Core at the University of California Davis); ntsLight2.0: The NTS sensor gene was cloned into a hSyn promoter and WPRE enhancer containing SV40 poly (A) signal sequence. The vector was packaged into an AAV serotype 9 (AAV9-ntsLight2.0-WPRE-SV40p(A), physical titre 2.19 × 10[16] vg ml[−1], produced by the Viral Core at the University of California Davis; virus was diluted 5× with saline before injection), from the Földy laboratory (AAV-NTS-OE: the NTS precursor gene was cloned into a short CAG promoter and WPRE enhancer containing SV40 vector with two lox sequences on each side of the gene to make NTS expression Cre-dependent). The vector was packaged into an AAV serotype 9 (ssAAV-9/2-shortCAG-dlox-mNts(rev)-dlox-WPRE-SV40p(A), ~10[13] infectious units per ml; produced by the Viral Vector Facility of the University of Zurich) and from the Lim laboratory (University of California San Diego, RG-EIAV-Cre).

For AAV injections, 300–500 nl of concentrated AAV solution was injected into the NAcLat (same coordinates as above) or VTA (same coordinates as above) using a syringe pump (Harvard Apparatus) at 150 nl min[−1]. The injection needle was withdrawn 5 min after the end of the infusion. Experiments were performed 6–24 weeks (for AAVs), 7 days (for retrobeads) or 2 days (for CTB–Alexa Fluor 647) after stereotaxic injections.

For in vivo electrophysiology, animals were implanted unilaterally above the NAcLat (bregma: 1.0 mm, lateral: 1.9 mm, ventral: −3.5 mm) with a custom-built driveable optoelectrode (optrode), which consisted of eight tetrodes (12-µm polyimide-coated nickel-chrome wire protected by silica tubing) glued to a 200-µm optical fibre using epoxy. The tetrodes protruded from the tip of the optical fibre by ~0.5 mm. Wire tips were cut flat, and gold plated to reduce impedance to ~200 kΩ at 1 kHz. A small screw fixed to the skull served as a ground electrode. Data collection began <1 week after the optrode implantations.

For in vivo optogenetics, animals received unilateral (for ChR2) implantations of a chronically implanted optical fibre (200 µm, NA = 0.37; Newdoon) dorsal to the VTA (bregma: −3.3 mm, lateral: 0.4 mm, ventral: −3.9 mm) or dorsal to the NAcLat (bregma: 1.0 mm, lateral: 1.9 mm, ventral: −3.6 mm). For ArchT experiments, optical fibres were implanted bilaterally and were angled (10°) above the VTA (bregma: −3.3 mm, lateral: ±1.2 mm, ventral: −4.4 mm). One layer of adhesive cement (C&B Metabond; Parkell) was followed by acrylic cement (Dental cement) to secure the optical fibre to the skull. The incision was closed with a suture and tissue adhesive (Vetbond; 3 M). The animals were kept on a heating pad until they recovered from anaesthesia. Atipamezole was injected intraperitoneally to reverse the sedative effects of dexmedetomidine.

For in vivo opto-pharmacology, animals were chronically implanted with a cannula (PlasticsOne, 33 G 4.6 mm) above the VTA (bregma: −3.3 mm, lateral: 0.4 mm, ventral: −3.9 mm). Opto-infusion experiments were done >1 week after cannula implantations. Injection sites and optical fibre placements were confirmed in all animals by preparing coronal sections (50 or 100 µm) of injection and implantation sites. Although optical fibre placements varied slightly from mouse to mouse, behavioural data from all mice were included in the study.

For in vivo fibre photometry experiments using ntsLight2.0, mice were chronically implanted with an optical fibre (400 µm, NA = 0.37; Newdoon) in the VTA (bregma: −3.3 mm, lateral: 0.4 mm, ventral: −4.5 mm) and above the NAcLat (bregma: 1.0 mm, lateral: 1.9 mm, ventral: −3.6 mm).

### Anatomical nomenclature

**Nucleus accumbens.** The NAc, a key component of the ventral striatum, is traditionally divided into shell and core subregions, which are both anatomically and functionally distinct[61]. In this study and in previous studies[26,62–64], we describe an additional ventral striatal subregion, termed NAc lateral shell, which is located laterally to the NAc core. We realize that the terminology 'NAc lateral shell' may be misleading since it may suggest that the NAc lateral shell is part of the NAc shell, even though these regions are anatomically and functionally different. Nevertheless, we use 'NAc lateral shell' (NAcLat) as it refers to an anatomical region that is defined in the *The Mouse Brain in Stereotaxic Coordinates*[65] (bregma: 1.34 mm to 0.74 mm).

Our findings that optogenetic stimulation of the NAcLat→VTA pathway specifically increases the consumption of hedonic foods without affecting standard chow intake suggest that the role of NAc medial shell inputs to the VTA in hedonic feeding behaviour warrants further investigation. However, optogenetic stimulation of NAc medial shell terminals in the VTA predominantly inhibits dopamine neurons and induces a general state of behavioural suppression, which is not specific to either reward- or aversion-related behaviours[26]. This complexity would make interpreting the effects of optogenetic stimulation on feeding behaviour within the NAc medial shell pathway more challenging.

**Ventral tegmental area.** We defined the lateral VTA as the medio-dorsal and lateral parabrachial pigmented nucleus and the medial lemniscus region adjacent to the substantia nigra. Please note that the definition of medial versus lateral VTA is largely based on the anatomical location of projection-defined VTA dopamine neurons. It is not strictly based only on the medio-lateral axis, but also incorporates the dorso-ventral axis[63].

## Electrophysiology

**Ex vivo electrophysiology.** Mice were deeply anaesthetized with pentobarbital (200 mg kg$^{-1}$, intraperitoneal injection; Vortech). Coronal brain slices containing the NAcLat or VTA (200 µm) were prepared after intracardial perfusion with ice-cold ACSF containing (in mM) 50 sucrose, 125 NaCl, 25 NaHCO$_3$, 2.5 KCl, 1.25 NaH$_2$PO$_4$, 0.1 CaCl$_2$, 6.174 MgCl$_2$, 2.96 kynurenic acid (NAcLat slices only) and 2.5 glucose (oxygenated with 95% O$_2$/5% CO$_2$). After 60–90 min of recovery, slices were transferred to a recording chamber and perfused continuously at 2–4 ml min$^{-1}$ with oxygenated ACSF, containing (in mM) 125 NaCl, 25 NaHCO$_3$, 2.5 KCl, 1.25 NaH$_2$PO$_4$, 2.5 glucose, 22.5 sucrose, 2.058 MgCl$_2$ and 2 CaCl$_2$ at -30 °C. Cells were visualized with a 40× water-immersion objective on an upright fluorescent microscope (BX51WI; Olympus) equipped with infrared-differential interference contrast video microscopy and epifluorescence (Olympus). For whole-cell current clamp recordings, patch pipettes (3.8–4.4 MΩ) were pulled from borosilicate glass (G150TF-4; Warner Instruments) and filled with internal solution, which consisted of (in mM) 135 potassium gluconate, 5 KCl, 10 HEPES, 0.1 EGTA, 2 MgCl$_2$, 2 MgATP, 0.2 NaGTP, pH 7.35 (290–300 mOsm). For perforated-patch recordings, patch pipettes were first filled with internal solution, as described above, and then back filled with internal solution containing 100 µg ml$^{-1}$ gramicidin. Electrophysiological recordings were made at 32 °C using a MultiClamp700B amplifier and acquired using a Digidata 1440 A digitizer, sampled at 10 kHz, and filtered at 2 kHz. All data acquisition was performed using pCLAMP software (Molecular Devices, version 10.5). Channelrhodopsin-2 was stimulated by flashing 473 nm light through the light path of the microscope using an ultrahigh-powered light-emitting diode (LED) powered by an LED driver (Prizmatix) under computer control. A dual lamp house adapter (Olympus) was used to switch between fluorescence lamp and LED light source. The light intensity of the LED was not changed during the experiments and the whole slice was illuminated (5 mW mm$^{-2}$). Series resistance (15–25 MΩ) and input resistance were monitored online. Data were analysed offline using Clampfit (Molecular Devices, version 10.5). For recordings of spontaneous action potential firing, cells were held in current clamp mode and no current injections were made. Spontaneous firing was recorded for at least 3 s before and 5 s after light stimulation (20 Hz, 3 ms light pulses, 5 mW mm$^{-2}$). For pharmacological experiments, we recorded baseline responses and drugs were bath applied for 5–10 min (1 µM SR142948A (Tocris), 1 µM neurotensin (Sigma Aldrich)). To determine the neurochemical identity of retrobead-labelled neurons (that is, TH-immunopositive or TH-immunonegative), neurons were filled with 0.1% neurobiotin (Vector), which was included in the internal solution, during patch-clamp recordings, then fixed in 4% paraformaldehyde (PFA) and 24 h later immunostained for tyrosine hydroxylase. The dopaminergic phenotype was confirmed in experiments shown in Fig. 4j–m. Neurobiotin was not included in experiments shown in Fig. 3a–f and Extended Data Fig. 9a–f. A more detailed description on the neurochemical identity of retrogradely labelled neurons in the VTA can be found in Lammel et al.[64].

**In vivo electrophysiology.** Mice with optrode implants were attached to a fibreoptic cable that was connected to a 473 nm DPSS laser diode (Laserglow) through an FC/PC adapter. Laser output was controlled using a Master-8 pulse stimulator (A.M.P.I.). Power output was tested using a digital power meter (Thorlabs) and was checked before and after each experimental animal; output during light stimulation was estimated to be 3–5 mW mm$^{-2}$ at the targeted tissue 200 µm from the fibre tip. Neural signals were recorded using a Digital Lynx 4SX system with an HS-18-MM headstage pre-amplifier (Neuralynx) with a Millmax connector. Recorded signals were filtered between 0.6 and 6 kHz and sampled at 32 kHz. Spikes were sorted offline using SpikeSort3D 2.5.4 (Neuralynx) software. To identify opto-tagged cells, the optrode was advanced ~40 µm per day, and a brief light screening (30 light pulses, 5 ms width at 1 Hz) was performed to detect light-responsive cells. If activity was detected, the mice proceeded to the behavioural assessment. If no activity was found, the optrode was advanced another ~40 µm, and the mice were re-tested for light-induced cell activity the following day. After behavioural assessment, the optrode was moved ~80 µm to minimize the likelihood of recording the same cells on subsequent days, and new opto-tagged cells were screened. Food consumption experiments while recording the neural signals, consisted of 3 behavioural trials (habituation, jelly, chow, 15 min each, order of food counterbalanced, 45 min total), and an opto-tagging trial at the end of the behavioural session where light stimulation was given at 2 Hz for 2 min. At the end of the opto-tagging stage, the optrode was moved ventrally for ~40 µm until active units were detected. The final recording location was verified using histology after the electrolytic lesions (12 µA, 30 s). ChR2-tagged neurons were identified by delivering 473 nm (0.8 mW mm$^{-2}$, 1–5 ms pulses) of light at 2 Hz frequency for 2–3 min. A 2-ms bin with the highest number of spikes in the interval (0, +100 ms) around the laser pulse was identified. To test if the identified strongest response to light was higher than chance, we shuffled all the spike times in the same (0, +100 ms) interval 10,000 times and counted the highest number of spikes in a 2-ms bin for each iteration. If the number of spikes in the 2 ms bin from the real data exceeded the 99.9th percentile value of the distribution of number of spikes in the most active 2-ms bin for the shuffled data, we classified the cell as light-responsive. Response latency was defined as the average response time in the most active 2 ms bin, and only units with response latency <8 ms were classified as opto-tagged. Examples of the opto-tagging procedure in REG and HFD mice are shown in Extended Data Fig. 1i–r.

**Piezo-based analysis of feeding.** For detection of feeding events, a piezo-based sensor was placed under the food cup. Cells were included in the analysis only if piezo sensor activity was detected. To obtain time-locked events, activation of the sensor was transmitted as a TTL signal to the Neuralynx recording system via an Arduino Uno board. Time spent feeding included the sum of events in which the sensor was activated at least twice and lasted until there was an interval larger than 6 s between the sensor activations. Piezo-detected events were confirmed by randomly inspecting 5 frames where the behaviour was determined as feeding in the analysis.

**Video-based analysis of behavioural motifs.** A video-based offline tracking was performed via DeepLabCut[66]. Specifically, DeepLabCut.py (version 2.0.7) was used to track all points of interest. The network was trained using 20 frames from 6 randomly selected videos (containing mice of different diets and food types) for >1,000,000 iterations. Horizontal $xy$ coordinates of the nose, head, body centre and the tail base were extracted for each frame. Only DLC coordinates of 85% accuracy and higher were used in the analysis. Deduction of behavioural motifs from the DLC obtained coordinates was done using a custom-made MATLAB code. The video start time was aligned to the Neuralynx recording system using the LED readout of the piezo sensor, and the matching TTL signal that was recorded directly to the Neuralynx data acquisition system via an Arduino Uno board. Deduction of behavioural motifs: feeding/empty motifs were defined as high proximity of the head position with either cup (<5 cm). Rearing was defined by close proximity of the body and head positions (<2 cm). Body and head orientation were determined as vectors from tail to body, and from body to head, respectively. Turning behaviour was defined as turning the head more than 15° from the body centre, which was then counted across a session[67]. The rest of the frames were classified according to velocity. Velocity was calculated by the distance of body position between frames, normalized to the size of the open-field chamber (in cm) and the frame rate (15 frames s$^{-1}$). Velocity-based behavioural motifs were defined for in vivo electrophysiology as <1 stopping, 1–5 walking, 5–10 trotting, and >10 running (units of cm s$^{-1}$). Velocity motifs were not included if the

mouse was too close to the food plate (<7 cm from either plate) to avoid confusion with the food/empty plate motifs. Except for turning events, each motif occurrence was included only if it persisted for <7 frames (0.5 s). The behavioural readout and motif selection were visually verified for each experiment by randomly inspecting five frames of each motif to confirm the correct motif selection (Extended Data Fig. 1g,h).

**Classification of IR and DR response types.** Response types during different behavioural motifs or piezo-based feeding events were classified into non-responsive, increased response (IR), and decreased responses (DR) using the following criteria: For each unit, average time series responses were collected around event onset from each motif type. Pre-event unit firing rates (−3 to 0 s before event onset) and event rates (0 to 3 s from event onset) were analysed using the Wilcoxon signed-rank test to determine statistical significance and the direction of change. IR: pre-event rates <event rates and $P < 0.05$; DR: pre-event rates > event rates and $P < 0.05$; otherwise, a unit would be classified as non-responsive. The proportion of the classified units (for both opto-tagged and non-tagged units) was analysed in REG and HFD mice and compared with the Chi-squared test; $P$ value was corrected using the Bonferroni correction for multiple comparisons.

NeuralynxMatlabImportExport_v6.0.0 MATLAB package (available at https://neuralynx.fh-co.com/research-software/), custom MATLAB code and GraphPad Prism (versions 9.5.1 and 10.3.1) were used for analysis of in vivo electrophysiology data.

## Behavioural assays

All behavioural tests were performed during the light phase (unless otherwise specified) in a temperature (20–23.5 °C) and humidity (40–60%) controlled room that is illuminated by 8× 32 W fluorescent lights each producing 2,925 lumens. Behavioural equipment was cleaned with 70% ethanol between individual animals.

**Acute feeding assay.** Mice were placed in a chamber (25 cm length × 25 cm width × 25 cm height) with 2 small empty food cups fixed to the floor on opposite corners. Following a 15-min habituation period, one empty cup was replaced with a cup containing a pre-measured amount of a specific food type (that is, standard chow, high-fat chow, chocolate, peanut butter, strawberry jelly, butter, water, or butter with quinine; note on preparation of butter with quinine: 30 g of butter was microwaved for 30 s to melt. 1.4 g of quinine (Q1250), dissolved in 15 ml of distilled deionized water, was thoroughly mixed into the butter. The butter–quinine mixture was then refrigerated until solidified and subsequently brought back to room temperature for use in the behavioural assay) that was weighed while the other cup remained empty. After each 15-min trial, the food cup was weighed again and subtracted from the initial weight to determine the amount of food consumed. Mice received a sample of each food type in their home cage and were habituated to the behavioural chamber on three different days prior to behavioural testing. This procedure was performed to avoid stress, and neophobia to novel foods, a characteristic behaviour in mice[68].

For in vivo optogenetic activation, mice with optical fibre implants were attached to a fibreoptic cable that was connected to a 473 nm DPSS laser diode. Optogenetic experiments consisted of 5 trials in the following order: habituation, primed-feeding, OFF, ON, OFF (15 min each; 75 min total). During the habitation trial, mice could freely explore the chamber, but no food was present. We placed a new pre-measured food cup containing a specific food type into the chamber at the beginning of each subsequent trial. The primed-feeding trial allowed us to assess the effects of diet on feeding behaviour. Additionally, because feeding behaviour was reduced in the subsequent OFF trial, it allowed us to test the effects of optogenetic stimulation when feeding behaviour was low. During the ON trial, 20 Hz (or 1 Hz, 10 Hz for experiments in Extended Data Fig. 3d), 5 ms blue light was delivered through the fibreoptic cable. There was no optogenetic stimulation during the habituation, primed-feeding and OFF trials. In experiments shown in Extended Data Figs. 3e,f and 4c,d, mice were food-deprived (FD; that is, all home cage food was removed) 24 h before the start of the experiment. Food consumption during the primed-feeding trial under FD condition was compared to baseline levels of primed-feeding measured in the same, but not FD, mice (that is, baseline measurements were performed the day before initiation of FD and consisted of habituation and primed-feeding trials (15 min each)).

For in vivo opto-pharmacology (Fig. 4f–i), mice were placed in an open-field chamber and a custom-made optical fibre was inserted into the cannula for light stimulation. Same experimental design as above, except that between trial 1 and trial 2 (that is, laser OFF and ON trials), mice were placed into a separate box where they were infused with either saline or 6 mM SR142948A (500 nl via infusion pump at a rate of 300 nl min⁻¹). There was an additional 10 min waiting period after infusion before animals were placed back into the chamber to conduct the remaining trials.

For in vivo optogenetic inhibition (Extended Data Fig. 3o–t), mice with bilateral optical fibre implants were attached to a fibreoptic cable connected to a 593 nm DPSS laser diode. The experiment was conducted over three consecutive days. On each day, mice were placed in a behavioural chamber for habituation (15 min, no food present), and jelly feeding (15 min). On day 2, constant laser light (10 mW) was delivered through the fibreoptic cable when jelly was present. Jelly consumption was measured for each day.

**Home cage feeding.** Mice in home cage feeding experiments received a pre-measured amount of standard chow and 60% fat chow that was measured weekly for each cage. Food consumption was normalized to the number of mice in each cage.

**Open-field test.** Mice were placed in a custom open-field chamber (50 cm length × 50 cm width × 50 cm height) and their movement was recorded and analysed for 10 min using video-tracking. MouseActivity5.m (https://github.com/HanLab-OSU/MouseActivity/blob/master/Mouse Activity5.m) was used to analyse open-field behaviour. To assess anxiety-related behaviour, we determined the time the animals spent in the centre of the chamber (33 cm length × 33 cm width)[69]. For analysis of behavioural motifs using DeepLabCut (version 2.0.7), we used a similar approach as described in 'Video-based analysis of behavioural motifs' to identify rearing, turning, and different velocity modalities. The behavioural readout and motif selection were visually verified for each experiment by randomly inspecting five frames of each motif (Extended Data Fig. 10h).

**Real-time place preference.** Mice with optogenetic implants were connected to a fibreoptic cable and placed in a custom-made three-compartment chamber. For optogenetic stimulation, the cable was connected to a 473 nm DPSS laser diode (Laserglow) through an FC/PC adapter, and laser output was controlled using a Master-8 pulse stimulator (A.M.P.I.). Power output was tested using a digital power meter (Thorlabs) and was checked before and after each experimental animal; output during light stimulation was estimated to be 3–5 mW mm⁻² at the targeted tissue 200 μm from the fibre tip. One side of the chamber was designated as the initial stimulation side (phase 1) and after 10 min the stimulation side was switched to the other previously non-stimulated side of the chamber (phase 2). The middle of the chamber was a neutral area that was never paired with stimulation. At the start of each session, the mouse was placed in the middle of the chamber, and every time the mouse crossed to the stimulation side, constant laser stimulation (473 nm, 20 Hz, 5 ms pulses) was delivered until the mouse exited the stimulation area. There was no interruption between Phase 1 and Phase 2. The first stimulation side was counterbalanced across mice. The movement of the mice was recorded via a video-tracking system

(Biobserve, version 3.0.1.442) and the time the mice spent in each area (stimulated, non-stimulated, neutral) was calculated.

**Body temperature measurements.** To measure body temperature, mice were manually restrained to minimize stress during measurement, and a rectal thermometer was inserted into the rectum to a depth of about 1 cm.

**Nutritional values for different food types (all values per 1g).** Jelly (Smucker's Strawberry Jelly; calories: 2.5 kcal, total fat: 0 g, saturated fat: 0 g, *trans* fat: 0 g, cholesterol: 0 g, sodium: 0 g, total carbohydrate: 0.65 g, dietary fibre: 0 g, total sugars: 0.6 g, protein: 0 g), butter (Trader Joe's Unsalted Butter; calories: 7.14 kcal, total fat: 0.79 g, saturated fat: 0.5 g, *trans* fat: 0 g, cholesterol: 0 g, sodium: 0 g, total carbohydrate: 0 g, dietary fibre: 0 g, total sugars: 0 g, protein: 0 g), peanut butter (Skippy; calories: 6.39 kcal, total fat: 0.5 g, saturated fat: 0.11 g, *trans* fat: 0 g, cholesterol: 0 g, sodium: 0 g, total carbohydrate: 0.25 g, dietary fibre: 0.06 g, total sugars: 0.11 g, protein: 0.22 g), chocolate (Hershey Kisses; calories: 4.88 kcal, total fat: 0.29 g, saturated fat: 0.9 g, *trans* fat: 0 g, cholesterol: 0 g, sodium: 0 g, total carbohydrate: 0.61 g, dietary fibre: 0.02 g, total sugars: 0.56 g, protein: 0.07 g), regular chow (PicoLab Rodent Diet 20; calories: 3.02 kcal, total fat: 0.05 g, saturated fat: 0.01 g, *trans* fat: 0 g, cholesterol: 0.14 g, sodium: 0 g, total carbohydrate: 0.54 g, dietary fibre: 0.04 g, total sugars: 0.03 g, protein: 0.21 g), high-fat chow (Research Diets D12492; calories: 5.24 kcal, total fat: 0.35 g, saturated fat: 0.18 g, *trans* fat: 0 g, cholesterol: 0 g, sodium: 0 g, total carbohydrate: 0.26 g, dietary fibre: 0.06 g, total sugars: 0.25 g, protein: 0.26 g).

## Development of ntsLight1.1 and ntsLight2.0

All constructs were designed using circular polymerase extension cloning and restriction cloning. BamHI and HindIII sites were introduced via PCR for final subcloning onto pAAV-hSynapsin1 vector (Addgene). To enhance coupling between conformational changes and chromophore fluorescence, we used a cpGFP module from GCaMP6s for insertion into the human NTS1R. For screening linker variants, we generated linker libraries by first designing primers with 22 C saturated mutagenesis[70] for one amino acid on each side of the linker (LSS-XI-cpGFP-XH-DQL). To screen ntsLight2.0 from ntsLight1.0, based on common activation pathway of class A GPCRs[71], we generated libraries at region 5.61 and 6.33 for screening. Cloned constructs were amplified and purified with the Qiagen PCR purification kit before NEB 5α Competent *Escherichia coli* transformation. Competent cells were plated onto kanamycin-containing agar plate. After 24 h of growth at 37 °C, single colonies were picked into 96-well plates and grown overnight. Plasmids were purified using Wizard MagneSil Plasmid Purification System (Promega, A1630) with Opentrons OT-2 liquid handler. Top variants were sequenced by Genewiz. ntsLight1.1 was discovered after linker screening (LSS-XI-cpGFP-XH-DQL) and resulted in WI-EH. ntsLight2.0 screening resulted in I259M and G301T. To make AAV plasmids, NEB stable competent cells were transformed with pAAV plasmids. After overnight growth on an ampicillin-containing agar plate at 30 °C, a single colony was selected and sequenced. The colony with the correct sequencing result was then expanded at 30 °C in 100 ml of growth medium (2×YT), purified with a Qiagen Endo-free plasmid Maxi Kit, and sent to the UC Davis Virus Core for virus production.

## ntsLight1.1 measurements in cell culture assays

**Cell culture preparation.** Glass-bottom 96-well plates (P96-1.5H-N, Cellvis) were coated with 0.1 mg ml⁻¹ of poly-D-lysine (Sigma, P6407-5MG) overnight. Plates were washed with water and E18 rat hippocampal neurons (BrainBits; https://tissue.transnetyx.com/E18-Rat-Hippocampus_4; not authenticated; not tested for mycoplasma contamination) were plated in neurobasal culture media with Neurobasal Plus Meidum (Gibco, A35829-01-500mL), B27 Plus Supplement (Gibco, A3582801), Glutamax (Gibco, 35050-061) and Gentamicin Reagent (Gibco, 15710-064). Neurons were infected with AAV9-hSyn-ntsLight1.1 (see above) on DIV5 neurons and changed to new media on DIV7. Half media change was performed every two days before imaging on DIV12.

**Neuronal cell titration.** Neuronal cell titration was performed in 96-well plate. Prior to a titration experiment on DIV12 neurons, stock solution of 10 mM neurotensin (Phoenix Pharmaceuticals) in $H_2O$ were diluted to 333 µM (in HBSS and 0.1 mg ml⁻¹ BSA) and distributed in all of the first wells in 96-well plates. The following wells had serial dilution in HBSS for neuronal titration. For imaging with antagonist, stock solutions of 1 mM SR142948A (Millipore Sigma) in $H_2O$ were diluted to 200 nM in imaging media distributed across an empty 96-well plate (ligand plate) in triplicate. The imaging media consisted of 1× HBSS (Fisher, 14175103) containing HEPES buffer. Neurons grown in a separate 96-well plate (imaging plate) were gently washed 3x with imaging media, and the wells were filled with an appropriate volume of imaging media for the respective experiment. For titration experiments, 50 µl of imaging media was added to each well of the assay plate. Wells were then imaged with ImageXpress MicroConfocal High-Content Imaging system at 40× (NA 0.6) with 4 regions of interest (ROI) taken per well with no overlap of the ROIs (exposure = 300 ms) with MetaXpress software (version V6.6.3.55). Next, 50 µl from the ligand plate was transferred to the imaging plate containing a doubled desired final concentration. After 5 min of incubation, the same sites were re-imaged using the same settings. Titration was done with final concentration ranging from 150 µM to 1 pM, with tenfold serial dilution each time.

**Ligand specificity test and validation.** Neurons were plated and cultured in a 4-chamber glass-bottom dish (35 mm, Cellvis) following the same protocol as described above. Neurons were imaged using 60× oil objective on a Leica Stellaris Confocal. The neurons were imaged in imaging buffer and 10 µl of the following ligands were applied directly to each chamber: NTS (10 µM, Phoenix Pharmaceuticals), GABA (100 µM, Tocris), dopamine (100 µM, Sigma), acetylcholine (Sigma, 100 µM), 5-HT (100 µM, Fisher), oxytocin (10 µM, Phoenix Pharmaceuticals), somatostatin (10 µM, Phoenix Pharmaceuticals), neuropeptide Y (10 µM, Phoenix Pharmaceuticals), cholecystokinin (10 µM, Phoenix Pharmaceuticals), dynorphin (10 µM, Phoenix Pharmaceuticals) and neuromedin U-25 (10 µM, Phoenix Pharmaceuticals). We observed a concentration-dependent increase in fluorescence in the presence of NTS that was attenuated by application of an NTS receptor antagonist (SR142948A) (Extended Data Fig. 6a).

**Image processing and analysis.** Once imaging was complete, the images were exported and analysed using a customized MATLAB script (available at: https://github.com/lintianlab). In brief, segmentation was performed on individual images and a mask highlighting the membrane of the neurons was generated. Pixel intensities were obtained from the mask-highlighted area and exported into Excel. The $\Delta F/F$ values for each well were calculated.

## Brain-slice recordings using ntsLight1.1

**Slice preparation and imaging.** Acute coronal midbrain slices were prepared (same procedure as described in ex vivo electrophysiology), transferred to a recording chamber and perfused continuously at 2–4 ml min⁻¹ with oxygenated ACSF. Slices were visualized under a custom-built, open source macroscope (https://github.com/Llamero/DIY_Epifluorescence_Macroscope) fitted with high power LEDs and a Teledyne Kinetix sCMOS camera. Custom drawn regions of interest were imaged at a rate of 20 Hz with a 10 ms exposure of 474 nm LED stimulation (5.2 mW mm⁻²) for a total of 20 s. In the middle of the recording, 1 s of 635 nm stimulation (17 mW mm⁻²) consisting of 5 ms pulses at 20 Hz was delivered to the slice between each camera exposure, so

that none of the Chrimson stimulation light was recorded. Green light stimulation experiments were performed similarly to the red-light stimulation, with 1 s of 554 nm stimulation (8 mW mm$^{-2}$) consisting of 5 ms pulses at 20 Hz. For pulse-width modulation experiments, red-light stimulation was delivered at 20 Hz with varying pulse widths. To determine d$F/F$ in the lateral VTA, the fluorescence from an ROI drawn away from sensor and Chrimson-expressing regions was divided from the fluorescence in lateral VTA. Because photobleaching curves were not identical between different regions of the tissue, an additional baseline subtraction was performed. A window of d$F/F$ signal around Chrimson stimulation time was removed, the remaining d$F/F$ data was smoothed, and an estimated polynomial fit trendline was drawn through the smooth data and across the removed stimulation time window. This trendline was subtracted from the complete d$F/F$ signal. AUC was calculated as an approximate trapezoidal integral during stimulation time.

**Ex vivo validation.** To examine NAcLat→VTA specific NTS release, mice were injected with ntsLight1.1 into the VTA and AAV-hSyn-ChrimsonR-tdTomato (Chrimson) into the NAcLat; a separate group of mice was infused with only ntsLight1.1 into the VTA (sensor only). Six weeks later, we recorded ntsLight1.1 fluorescence from VTA brain slices during light stimulation. Red-light stimulation increased ntsLight1.1 fluorescence in VTA slices of Chrimson mice, but not in 'sensor only' mice (Extended Data Fig. 6b–d). Additional optical control experiments revealed that the increase in ntsLight1.1 fluorescence reaches a maximum at 10 ms red-light pulse widths suggesting ntsLight fluorescence reflects dynamics of NTS release rather than total light delivered to tissue. Delivering blue-light or red-light stimulation in isolation was insufficient to increase ntsLight1.1 fluorescence (Extended Data Fig. 6e,f).

## ntsLight2.0 measurements in cell culture assays

**Cell culture preparation.** HEK 293 T cells (ATCC, CRL-3126; not authenticated; not tested for mycoplasma contamination) were plated and concurrently transfected with pCMV-ntsLight2.0 using Lipofectamine 2000 (Invitrogen, 2980874) according to the manufacture's protocol. 24 h after transfection, cells were lifted using trypLE Express (Thermo Fisher, 12604021), pelleted (200 g for 2 min) and resuspended in 1 ml culture media containing DMEM (Gibco, 11995-065), fetal bovine serum (Gibco, 26140079) and Pen-Strep (Gibco, 15140148). Cells were then plated onto 4-chamber glass-bottom dishes and imaged the next day.

**Spectral measurements.** For spectral analysis to determine the optimal excitation wavelength for ntsLight2.0, we used the Leica Stellaris 8 confocal microscope to perform both excitation and emission spectrum measurement. After washing each plate with HBSS (Sigma Aldrich, H8264-500ML), 90 µl of imaging media with 1× HBSS (Fisher, 14175103) and 10 mM HEPES buffer was added to the centre of each quadrant. For emission spectrum measurement, we used λ-scan mode (xyλ) by exciting at 470 nm and imaged with a range of emission wavelength from 480–610 nm with 10 nm step size and 10 frame accumulation. For excitation spectrum measurement, we used excitation lambda scan mode (xyΛ) by exciting with white light laser in a range of wavelength from 440–540 nm with step size at 10 nm. The detection range of the detector precedes the excitation wavelength during the lambda scan emission wavelength. For emission and excitation spectrum with neurotensin, 10 µl of 20 µM NTS was added prior to imaging. Analysis was done using custom code to calculate change in fluorescence (Δ$F/F$) with before (apo) and after (+NTS) ligand addition. Fluorescence changes were then normalized to the maximum fluorescence in each group (Extended Data Fig. 7b).

**Primary hippocampal neuron with antagonist imaging experiment.** Glass-bottom 96-well plates (Cellvis) were coated with 0.1 mg ml$^{-1}$ of poly-D-lysine (Sigma, P6407-5MG) overnight. Plates were washed with UltraPure Distilled Water (Invitrogen, 10977015) and air dried. E18 rat primary hippocampal neurons (BrainBits, https://tissue.transnetyx.com/E18-Rat-Hippocampus_4; not authenticated; not tested for mycoplasma contamination) were dissociated and plated with 38 thousand cells per well in FBS-based neuronal medium containing Neurobasal Plus Medium (Gibco, A35829-01), FBS, GlutaMAX (Gibco, 35050-061) and B27 Plus (Gibco, A3582801). On the next day, medium was removed and replaced with FBS-free neuronal media. On DIV4, half of the neuronal media was changed with new media containing virus AAV9-hSyn-ntsLight2 and removed three days later. Neuronal cultures were imaged on DIV12. Immediately before an imaging experiment, stock peptide solution was prepared in a 96-well treatment plate and serial dilutions (from 300 µM to 3 pM final in HBSS) were prepared across each row. 1 nM and 100 nM final concentration of SR 142948 A (Sigma, SML0015) were then added to the treatment plate. Before adding drug treatment, the 96-well assay plate were washed with HBSS three times and 50 µl imaging medium (vehicle) was added to each well. Baseline imaging was done using ImageXpress Micro Confocal High-Content Imaging System with MetaXpress software (version V6.6.3.55) using a 20× objective and capturing four regions of interest per well. Next, 50 µl of ligand per well from the treatment plate was transferred to the assay plate. After 10 min incubation, the same sites were re-imaged using the same settings. For titration controls without antagonist, only neurotensin from 300 µM to 3 pM dissolved in HBSS were used. Blank controls with vehicle were present in every condition. The images were exported and analysed using a custom MATLAB script (available at https://github.com/lintianlab) to determine changes in fluorescence (Δ$F/F$). Segmentation was performed on individual images and a mask highlighting the membrane of the HEK293T cells was generated. Pixel intensities were obtained from the mask-highlighted area and the Δ$F/F$ values for each well were calculated and exported (Extended Data Fig. 7c).

## Fibre photometry recordings using ntsLight2.0

**Signal recording and processing.** ntsLight2.0 transients were measured using a custom-built fibre photometry (FIP) system[63]. In brief, fluorescent signals were obtained by stimulating cells expressing ntsLight2.0 with a 470 nm LED (60 µW at fibre tip). 470 nm LED light signals were released at 20 Hz and light emission was recorded using a sCMOS camera that acquired video frames containing the fibre (5 mm in length, NA 0.48, 400 µm core, Doric Lenses). A TTL signal generated by an Arduino Uno board was used to synchronize the camera and the FIP signal. Video frames were analysed online, and fluorescent signals were acquired using a custom acquisition code (available at https://github.com/handejong/Fipster) and later analysed in GraphPad Prism (version 10.3.1).

**Opto-photometry experiments.** ntsLight2.0 signals in the VTA were recorded during optogenetic stimulation of NAcLat cells. Specifically, mice were placed in a head-fixed apparatus and connected to fibre-optic patch cords. A 640 nm collimated diode laser was controlled by an Arduino Uno, and laser stimulation times were recorded in the FIP acquisition system with a TTL signal. Each session had 30 trials, 60-s duration, composed of a short laser pulse followed by a delay period. For analysis, we generated peri-event plots of the $z$-scored data around the laser stimulation onset and analysed the timeframe of −10 to +30 s around the laser stimulation onset across all trials. We then quantified AUC at specific time points: baseline (−3 to −1 s), during laser stimulation (0 to 2 s) and post-laser stimulation (3 to 5 s).

**In vivo validation.** NAcLat laser stimulation produced bidirectional ntsLight2.0 transients in the VTA composed of a fast negative peak that occurred during laser stimulation (0 to 2 s) followed by a slower positive peak (3 to 5 s). We performed several experiments to test

which component of the signal reflects a change in NTS release: First, we compared the observed response in mice expressing both Chrimson and ntsLight2.0 (sensor) to mice that lack one of these components (that is, sensor only, Chrimson only). The positive peak observed in the post-laser stimulation period was increased only when both sensor and Chrimson were expressed (Extended Data Fig. 7d–h). Second, we tested the sensitivity of the signal to varying stimulation patterns. When testing different stimulation durations (1, 3 or 5 s, 10 mW intensity, 20 Hz) we observed that the post-laser stimulation signal had the strongest increase during the 3-s stimulation duration and a moderate change with 1- or 5-s durations (Extended Data Fig. 7i–l). Increasing the stimulation intensities (0.5, 5 or 10 mW intensity for 3 s, 20 Hz) produced a proportional increase in sensor signal (Extended Data Fig. 7m–p). Third, we analysed the sensor signal in response to laser stimulation (3 s, 10 mW, 20 Hz) following intraperitoneal injection of an NTSR1 antagonist (SR48692, Sigma Aldrich, 5 mg kg$^{-1}$) or an equal amount of DMSO, which were injected 10 min before recording onset. We found that the slower positive, but not the fast negative peak, was blocked by the NTSR1 antagonist (Extended Data Fig. 7d–u), suggesting that laser stimulation produces a brief artefact that is followed by an increase of NTS release. While the nature of the artefact is currently unclear, it is also observed in recordings of other neuropeptide sensors, such as recently developed opioid sensors[72], and adenosine sensor (GRAB$_{Ado}$)[73]. Future experiments are required to test if recruitment of additional neurochemical signalling processes during stimulation may suppress NTS release. Lastly, to test the sensitivity of the sensor to NTS, we continuously recorded the sensor for 20 min, and then the mice were injected with an NTSR1 agonist (PD149163, Sigma Aldrich, 0.3 mg kg$^{-1}$, intraperitoneal injection). A TTL signal was delivered to the FIP acquisition system from a button pressed on an Arduino Uno board and the signal was recorded for an additional 50 min. The signal was aligned to injection timepoint, detrended and Z-scored averaged using a custom MATLAB code. AUC was analysed during the baseline period (−500 to 0 s before injection) and post injection (1500 to 2000 s). We observed a significant increase in sensor signal in response PD149163 but not saline injection (Extended Data Fig. 7v–x).

### Single-cell patch RNA sequencing

**Sample collection.** The procedure was described previously[74]. To minimize interference with subsequent molecular experiments, only a small amount of intracellular solution (~1 µl; not autoclaved or treated with RNase inhibitor) was used in the glass pipette during electrophysiological recordings. Before and during recording, all surface areas—including manipulators, microscope knobs and computer keyboard—that the experimenter needed to contact during the experiment were cleaned with RNaseZAP solution (Sigma). After whole-cell patch-clamp recordings, the cell's cytosol was aspirated via the glass pipette used during the recording. Although the aspirated cytosol may have contained genomic DNA, our choice of cDNA preparation, which involved poly-A based mRNA selection, virtually eliminated the possibility of genomic contamination in the RNA-sequencing data. Cell collection microtubes were stored on ice until they were used. For sample collection, we quickly removed the pipette holder from the amplifier headstage and used positive pressure to expel samples into microtubes containing 1 µl cell collection buffer (1× Lysis Buffer and RNase inhibitor from Takara's SMART-seq kits) while gently breaking the glass pipette tip. The sample was spun briefly, snap-frozen on dry ice and stored at −80 °C until further processing.

**cDNA library preparation.** As described previously[74]. Single-cell mRNA was processed using Takara's SMART-Seq v4 and SMART-Seq Single Cell kit according to the manufacturer's protocol. In brief, the samples were reverse transcribed to cDNA and subsequently amplified. The samples were purified with AMPure XP beads (Beckman Coulter), and the quality and quantity were analysed on a Fragment Analyzer (Advanced Analytical). Library preparation was performed using Nextera XT DNA Sample Preparation Kit (Illumina) as described in the protocol. In short, cDNA samples were fragmented and amplified using index adapters. The samples were then purified using AMPure XP beads and quality and quantity was assessed with the Fragment Analyzer (Advanced Analytical). Following library preparation, cells were pooled and sequenced using Illumina Novaseq 6000 with 150 bp paired-end reads.

**Bioinformatics.** After sequencing, raw reads were de-multiplexed using Illumina bc12fastq (version 2.20), and pseudo-aligned to the Ensembl GRCm38.95 reference transcriptome and normalized using kallisto (version 0.45.1). All data analysis was performed using Python (version 3.6.7) and R (version 3.5.1). For quality control, we calculated the median absolute deviation of each cell for the reads and the gene counts. Cells with less reads or gene counts than three times the median absolute deviation from the median were excluded. Differential gene expression analysis was performed using edgeR (version 3.24.3). Only genes that were expressed in at least 5 cells with a CPM value above 15 and the average CPM value has to be higher than 4 to be considered for differential gene expression analysis.

**Identification of gene families.** The Gene Ontology (GO) terms for identification of specific gene families was obtained from QuickGO platform (https://www.ebi.ac.uk/QuickGO/).

Synaptic molecules gene family: GO:0005179 hormone activity, GO:0032098 regulation of appetite, GO:0007218 neuropeptide signalling pathway, GO:0007269 neurotransmitter secretion and GO:0007268 chemical synaptic transmission.

Feeding gene family: GO:0007631 regulation of feeding behaviour and GO:0007631 feeding behaviour.

Synaptic gene family: GO:0043083 synaptic cleft, GO:0097060 synaptic membrane and GO:0099536 synaptic signalling.

Ion channels gene family: GO:0006816 calcium ion transport, GO:0006817 phosphate ion transport, GO:0006814 sodium ion transport, GO:0006811 monoatomic ion transport and GO:0006813 potassium ion transport.

Endoplasmic reticulum gene family: GO:0005783 endoplasmic reticulum.

Vesicle fusion family: GO:0031338 regulation of vesicle fusion, GO:0006906 vesicle fusion and GO:0098992 neuronal dense core vesicle.

Transcription gene family: GO:0010468 regulation of gene expression, GO:0031564 transcription, GO:0090293 antitermination nitrogen catabolite regulation of transcription, GO:0045990 carbon catabolite regulation of transcription, GO:0000409 regulation of transcription by galactose and GO:0046015 regulation of transcription by glucose.

### Histology and microscopy

**Immunohistochemistry.** Immunohistochemistry was performed as described previously[26,64]. In brief, after intracardial perfusion with 4% PFA in PBS, pH 7.4, the brains were post-fixed overnight and coronal brain sections (50 or 100 µm) were prepared. Sections were stained overnight in a primary antibody solution: rabbit anti-TH (Millipore), chicken anti-GFP (Abcam), rabbit anti-DS Red (Living Colors), all at 1:1,000 dilution. Twenty-four hours later, sections were stained for 2 h in secondary antibody solution (goat anti-rabbit Alexa Fluor 546, (all Thermo Fisher Scientific), goat anti-chicken Alexa Fluor 488 (Abcam), all 1:750). Stained sections were mounted with DAPI-containing mounting medium on microscope slides. Image acquisition was performed with Zeiss LSM710 laser scanning confocal microscope using 20× or 40× objectives or on a Zeiss AxioImager M2 upright widefield fluorescence/differential interference contrast microscope with charge-coupled device camera using 5×, 10× and 20× objectives. Zen Software 2.3 (Zeiss) was used for acquiring confocal and epifluorescence images of brain slices. Images were analysed using ImageJ (NIH, 64-bit Java 1.8.0_172).

Sections were labelled relative to bregma using landmarks and neuro-anatomical nomenclature as described in *The Mouse Brain in Stereotaxic Coordinates*[65]. All images presented with multiple colours represent a composite of images collected with different excitation wavelengths.

**Fluorescent in situ hybridization.** Fluorescent in situ hybridization experiments were conducted using a commercially available RNAscope Multiplex Fluorescent Reagent Kit V2 (ACD Bio). Brains were extracted and snap-frozen by submerging them into frozen isopentane (−70 to −50 °C). They were stored in an airtight container in an −80 °C freezer. 16 µm coronal NAcLat brain slices were prepared using a cryostat, placed on Superfrost Plus microscope slides (Fisher Scientific) and stored in a −80 °C freezer. On the next day, brain slices were fixed in 4% PFA in PBS (30 min) followed by an ethanol dehydration procedure (20 min). Slices were then bathed in hydrogen peroxide (10 min), followed by protease IV from the RNAscope kit (15 min). Next, probe mixes were made for *Nts* (Mm-Nts-C2), *Ntsr1* (Mm-Ntsr1-C2) or *Th* (Mm-Th). A custom-made probe (Mm-Nts-O1-C2), targeting only axon 4 of NM_024435.2 (*Nts*) was designed to assess NTS expression after conditional knockout of *Nts* in *Nts^flox^* mice (Fig. 4a–e). Probe mixes were applied to the brain slices for hybridization (2 h at 40 °C). After amplification of the signal (using AMP1, AMP2 and AMP3 from the RNAscope kit), channel C1 was developed using green Opal 520 (Akoya Biosciences, USA) and channel C2 was developed using orange Opal 570 (Akoya Biosciences). Finally, nuclei were stained using DAPI (from the RNAscope kit) and brain slices were sealed with ProLong Gold Antifade mountant (Thermo Fisher Scientific) and a glass coverslip. Images were taken using a confocal microscope (LSM710, Carl Zeiss) at 5 different *z* depths (spanning 4.4 µm), and images were flattened by taking the maximum projection across the *z* direction. ROIs were identified using a machine learning-based segmentation algorithm nucleAIzer based on the DAPI channel[75]. The amount of visible mRNA across the DAPI-identified region was used as a proxy for total mRNA in the cell. All identified regions of interest were manually sorted by an investigator who was blind to virus expression, diet, and probe mix. ROIs were removed if they: (1) showed overlap with other regions of interest; or (2) were segmented inadequately by the algorithm. The remaining cells were analysed based on the percentage of DAPI-positive pixels that were also positive for targeted mRNA or based on average fluorescence of targeted mRNA in DAPI-positive cells. To adjust for potential differences in staining and/or image quality, we compared pixels in all regions of interest to background fluorescence levels in each image. To do this, we first established a 'null distribution' that quantifies the distribution in pixel intensity values for cells putatively negative for targeted mRNA. Each cell's distribution of pixel intensities was compared to the null distribution for the targeted mRNA and a correlation coefficient *R* was calculated. If the *R* of a cell's distribution compared to the null distribution was less than 0.85, then a cell was labelled as positive for the targeted mRNA. For experiments in Extended Data Fig. 5i–m, mice exposed to regular diet (REG), 4-week HFD, or mice switched from a 4-week HFD to a 3-week regular diet were injected with a retrograde tracer (fluorescent retrobeads, red) into the VTA. 10 days later, the brains were extracted, and *Nts* mRNA was assessed in DAPI-positive NAcLat cells labelled with retrobeads.

## Statistics and reproducibility

**Main effect.** Student's *t*-tests (paired and unpaired), one-way or two-way ANOVA tests, and mixed-effect model analyses (for normally distributed data) and Friedman, Kruskal–Wallis or Mann–Whitney test (for non-normally distributed data) were used to determine statistical differences using GraphPad Prism 9 (version 9.5.1) and 10 (version 10.3.1) (Graphpad Software). Differential gene expression analysis was performed using Python (version 3.6.7), R (version 3.5.1) and edgeR (version 3.24.3). Wilcoxon's signed-rank test ($\alpha$ set to 5%) and MATLAB (version R2024a) were used for the analysis of in vivo electrophysiology data.

**Multiple comparisons.** When a main effect or interaction were reported, Holm–Šídák (for normally distributed data) or Dunn (for non-normally distributed data) post hoc analysis were applied. Spearman correlation coefficient and linear regression were used to measure the strength and direction of the linear relationship between two variables. Statistical significance was denoted by $*P < 0.05$, $**P < 0.01$, $***P < 0.001$. Data are presented as mean ± s.e.m. for parametric tests, and as median, 25th percentile and 75th percentile for non-parametric tests.

**Replication.** Several experiments were replicated in at least two technical replicates from each of at least two mice and similar results were obtained. For example, establishing the HFD mouse model (Fig. 1a–d), optogenetics (Figs. 2 and 4i) and NTS-OE (Fig. 5f–l and Extended Data Fig. 10e–k). For anatomical experiments, wherever representative examples are shown (Fig. 4g,k and Extended Data Figs. 1a–d, 3a,b,n, q–t, 4a,b, 6h, 7aa,ab, 8e,f,i,j and 9a,d), similar results were obtained in at least two technical replicates from each of at least two mice.

### Reporting summary

Further information on research design is available in the Nature Portfolio Reporting Summary linked to this article.

## Data availability

The RNA-sequencing datasets generated during this study are available at the Gene Expression Omnibus under accession GSE287548. Source data are provided with this paper.

## Code availability

Custom code used for the processing of fibre photometry data is available at https://github.com/handejong/Fipster. MouseActivity5.m (https://github.com/HanLab-OSU/MouseActivity/blob/master/Mouse Activity5.m) was used to analyse open-field behaviour. Code for analysis of ntsLight sensors in cultured cells is available at https://github.com/lintianlab. NeuralynxMatlabImportExport_v6.0.0 MATLAB package (available at https://neuralynx.fh-co.com/research-software/) was used for analysis of in vivo electrophysiology data. Custom MATLAB (version R2024a), Python (version 3.6.7) and edgeR (version 3.24.3), which were used for the processing of in vivo electrophysiology data and RNA-sequencing data are available at https://github.com/lammellab/.

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

**Acknowledgements** S.L. is a John P. Stock Faculty Fellow, Weill Neurohub Investigator and Rita Allen Scholar. This work was supported by the McKnight Foundation (S.L.), the One Mind Foundation (047483, S.L.), the Weill Neurohub (S.L.), the Rita Allen Foundation (S.L.), the Wayne and Gladys Valley Foundation (S.L.), a NARSAD Young Investigator Award (30374, N.G.S.), the NIH (R01DA042889, S.L.; U01NS120820 and U01NS113295, L.T.; R01NS121231 and R01DA049787, B.L.), the Swiss National Science Foundation (310030_219390, C.F.), the Dr Eric Slack-Gyr-Stiftung (C.F.) and an NSF GRFP (A.J.T.). The development of the macroscope and illuminator was funded by the following grant: NIH P30EY003176. Research reported in this publication was supported in part by the National Institutes of Health S10 programme under award number 1S10RR026866-01. The content is solely the responsibility of the authors and does not necessarily represent the official views of the National Institutes of Health.

**Author contributions** Stereotaxic injections were performed by N.G.S. and A.J.T. In vivo electrophysiology was performed by N.G.S. and J.R. DLC analysis was performed by N.G.S. Ex vivo electrophysiology was performed by A.J.T., C.L. and H.Y. RNAscope was performed by N.G.S., A.J.T. and J.P.H.V. ntsLight was developed and tested in culture by Y.J. and L.T. Ex vivo ntsLight1.1 recordings were performed by A.J.T. and Y.J. Immunohistochemistry was performed by N.G.S., A.J.T., E.H. and J.R. In vivo ntsLight2.0 assays were performed by N.G.S. and M.T. Behavioural experiments were performed by N.G.S., A.J.T., E.H., L.W.T. and J.R. RNA sequencing was performed by C.S. and C.F. NTS-OE construct design and cloning was done by T.L. and C.F. B.L. provided RG-EIAV-Cre virus. Data were analysed by N.G.S. and A.J.T. The study was designed by N.G.S., A.J.T. and S.L. The manuscript was written by N.G.S. and S.L. and edited by all authors.

**Competing interests** The authors declare no competing interests.

**Additional information**
**Correspondence and requests for materials** should be addressed to Stephan Lammel.

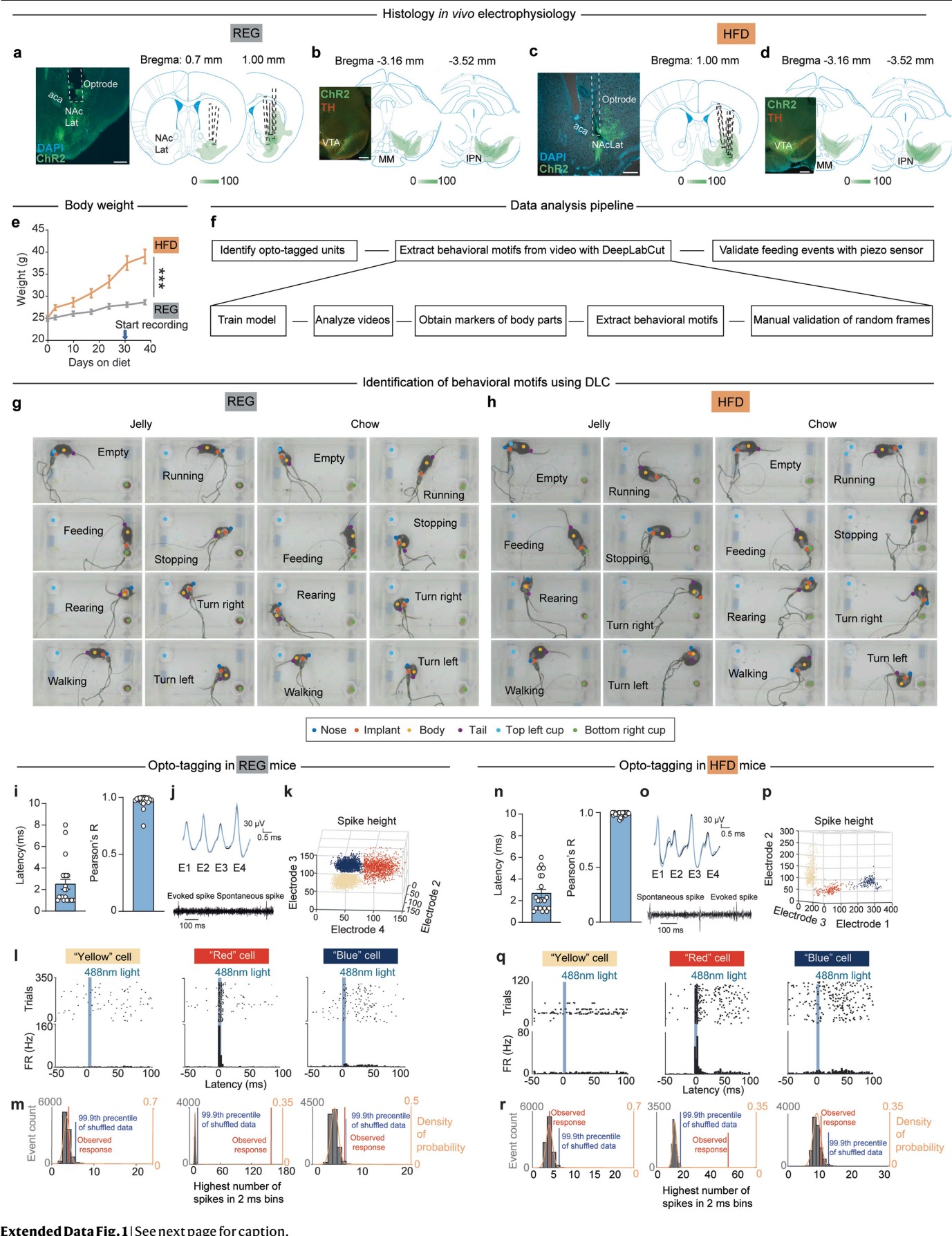

**Extended Data Fig. 1** | See next page for caption.

**Extended Data Fig. 1 | In vivo electrophysiological recordings of NAcLat →
VTA neurons in REG and HFD mice. (a)** Left: sample fluorescent image
showing recording location in the NAcLat for mice subjected to a regular diet
(REG) (green: ChR2, blue: DAPI; aca: anterior commissure; scale bar 500 μm).
Right: schematic overviews showing recording locations of all animals. Green
indicates ChR2 expression. Refers to animals shown in Fig. 1. **(b)** Left: Sample
fluorescent image showing ChR2-expressing (green) NAcLat terminals in the
VTA of REG mice (red: tyrosine hydroxylase (TH); scale bar 500 μm). Right:
schematic overviews showing ChR2 expression in NAcLat terminals (green)
for all recorded animals (MM: medial mammillary nuclei, IPN: interpeduncular
nucleus). Refers to animals shown in Fig. 1. **(c, d)** Same as in (a, b) but for mice
subjected to a high-fat diet (HFD). Refers to animals shown in Fig. 1. **(e)** Mice
with *ad libitum* access to high fat chow (HFD, orange) gained significantly
more weight than mice who received a regular diet (REG, grey). Arrow indicates
timepoint when electrophysiological recordings started (***p = 0.0009,
$n_{REG}$ = 8 mice, $n_{HFD}$ = 7 mice, 2-way RM ANOVA with Holm-Šídák's test). **(f)** Overview
of data analysis pipeline for analysing opto-tagged and non-tagged single unit
activity across behavioural motifs in freely behaving mice. **(g, h)** Sample frames
from different behavioural motifs obtained from REG (g) and HFD (h) mice
when they were presented with either jelly or chow. **(i)** Left: mean response
latency to light stimulation for all recorded units in mice subjected to a regular
diet (REG). Right: all opto-tagged units showed high correlation between
evoked and spontaneous action potential waveforms ($n_{REG}$ = 21 units from
n = 8 mice). **(j)** Top: spontaneous (black) and evoked (blue) averaged action

potential waveforms for a sample NAcLat→VTA unit. Bottom: sample recording
of an isolated unit with light-evoked and spontaneous spikes. **(k)** Sample plot
showing spike sorting. A three-dimensional plot of simultaneously recorded
units sorted according to action potential height detected at each recording
electrode. Colours indicate action potentials that belong to separate sample
units "Yellow", "Red", and "Blue". **(l)** Sample units during opto-tagging trials
based on plot shown in panel (k). Top: raster plots show example unit action
potentials detected during each light delivery trial at time = 0 ms (blue line
indicates delivery of 473 nm laser light; 5 ms pulse; 2 Hz). Bottom: corresponding
histograms show sum of all recorded action potentials across light delivery
trials relative to onset of blue light stimulation (5 ms light delivered at time = 0 ms).
"Red" unit is light responsive, "Blue" and "Yellow" units are not light responsive
(FR: firing rate). **(m)** Corresponding histograms of 2 ms bins (grey) showing
the max extracted spike count from shuffled unit spike data collected during
opto-tagging trials. The 99.9th percentile from the shuffled data distribution
(orange) is indicated by a blue vertical line. The observed max extracted spike
count from any 2 ms bin spikes is indicated by a vertical red line. If the observed
max extracted spike count (red line) was greater than the max extracted spike
count from the shuffled data (blue line), then the recorded unit was classified
as light responsive (i.e., opto-tagged). "Red" unit is light responsive, "Blue"
and "Yellow" units are not light responsive. **(n-r)** Same as in (i-m) but for mice
subjected to a high-fat diet ($n_{HFD}$ = 20 units from n = 7 mice). All data represented
as mean ± SEM. All statistical significance was determined using a two-sided
hypothesis.

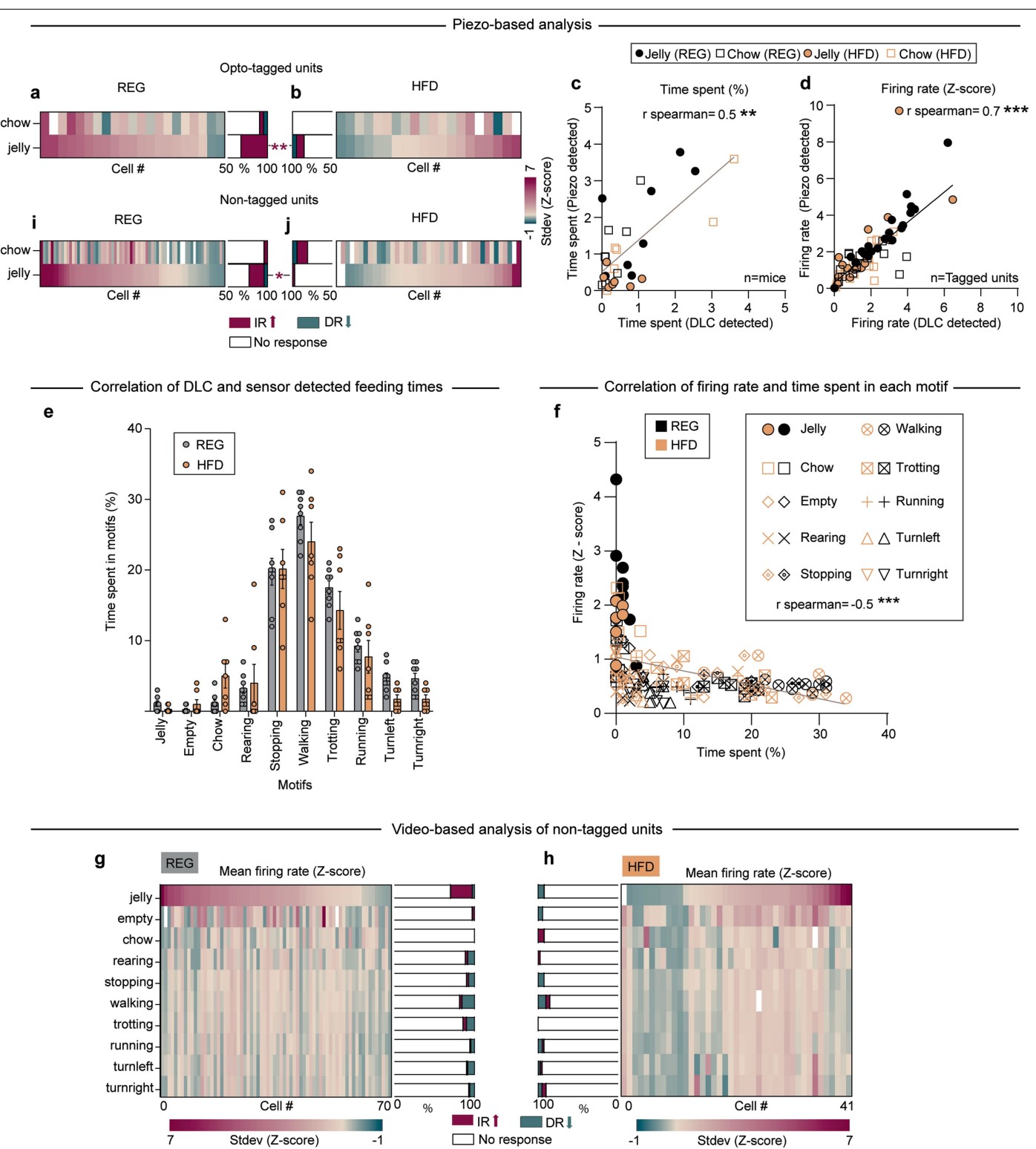

**Extended Data Fig. 2** | See next page for caption.

**Extended Data Fig. 2 | Piezo- and DLC-based analyses of opto-tagged and non-tagged units. (a, b)** Heatmaps showing the relative Z-score average of individual NAcLat→VTA (opto-tagged) neurons (columns) during piezo sensor activation for jelly and chow feeding (rows) for REG (a) and HFD (b) mice. The bar graphs show the percentage of increased response type (IR, red), decreased response type (DR, blue) and non-responsive (white) opto-tagged units (**p = 0.009 for relative proportions of IR responses in REG versus HFD mice, $n_{REG}$ = 21 units from n = 8 mice; $n_{HFD}$ = 20 units from n = 7 mice; two-sided Chi-square test for proportions with Bonferroni correction for multiple comparisons). **(c, d)** Diagrams showing a significant correlation between feeding events detected across all conditions with DLC analysis and piezo sensor for time spent feeding (c) (**p = 0.0044, n = 30 XY pairs from $n_{REG}$ = 8 mice; $n_{HFD}$ = 7 mice; r Spearman: 0.5, linear regression) and for opto-tagged unit firing rate (d) (***p < 0.001, n = 55 XY pairs from $n_{REG}$ = 8 mice; $n_{HFD}$ = 7 mice; r Spearman: 0.7, linear regression). **(e)** DLC-based analysis of average time spent in each motif for REG (grey) and HFD (orange) mice. There was no significant difference in the time REG and HFD mice spent in either motif (p > 0.05, $n_{REG}$ = 8 mice, $n_{HFD}$ = 7 mice, 2-way RM ANOVA). Data represented as mean ± SEM. **(f)** Diagram showing an inverse correlation between time spent and Z-scored firing rate across all behavioral motifs (***p < 0.001, $n_{REG}$ = 8 mice; $n_{HFD}$ = 7 mice, r Spearman: −0.5, linear regression). **(g, h)** Heatmap showing the relative Z-score average of individual NAcLat (non-tagged) neurons (columns) during different behavioral motifs (rows) for REG (g) and HFD (h) mice. The bar graph shows the percentage of IR (red), DR (blue) response types and non-responsive (white) non-tagged units in each behavioural motif ($n_{REG}$ = 70 units from n = 8 mice; $n_{HFD}$ = 41 units from n = 7 mice; Chi-square test for proportions with Bonferroni correction for multiple comparisons). **(i, j)** As in (a, b) but for non-tagged units (*p = 0.0312 for relative proportions of IR responses in REG versus HFD mice, $n_{REG}$ = 70 units from n = 8 mice; $n_{HFD}$ = 41 units from n = 7 mice; two-sided Chi-square test for proportions with Bonferroni correction for multiple comparisons).

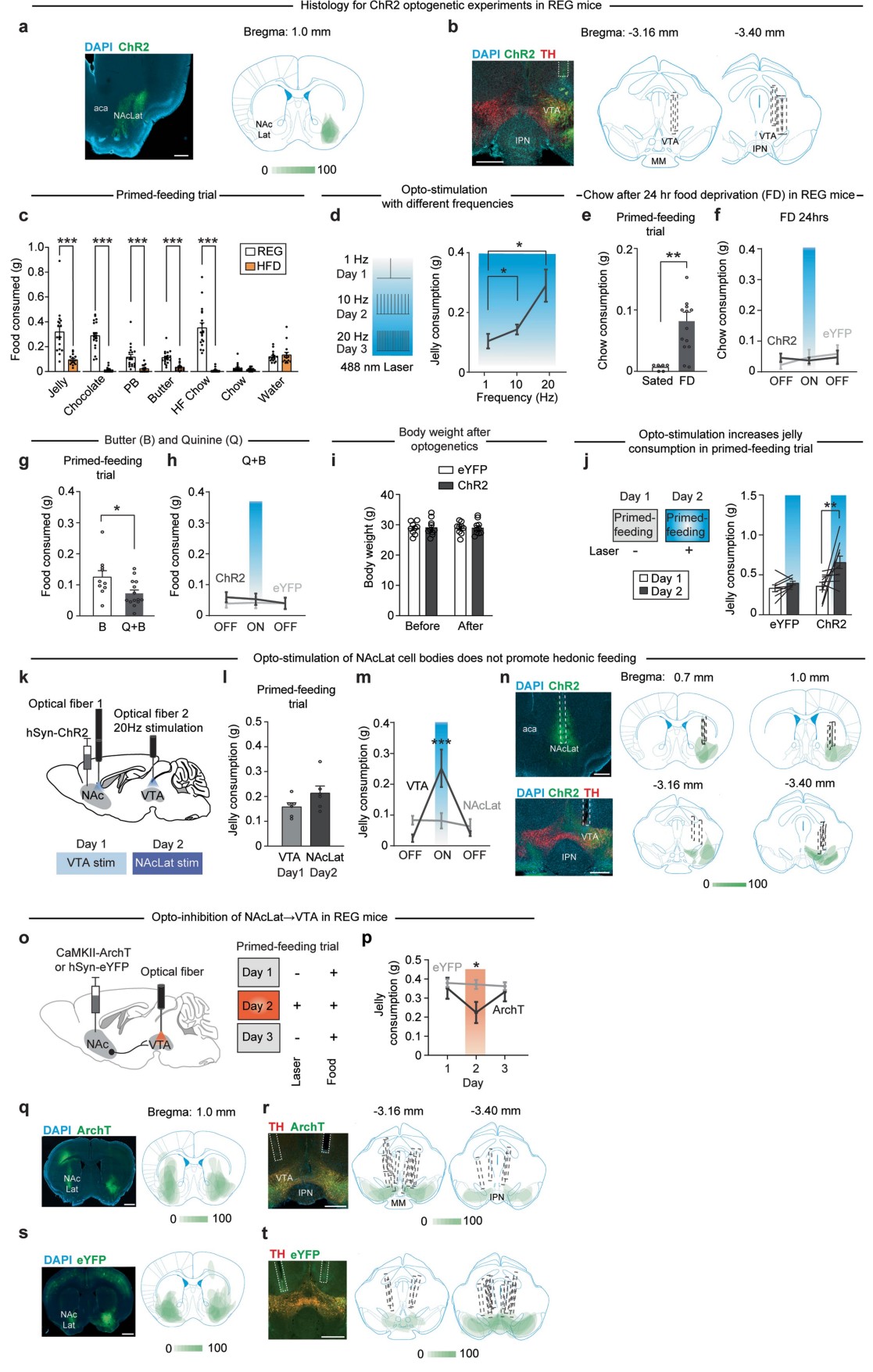

**Extended Data Fig. 3 |** See next page for caption.

**Extended Data Fig. 3 | Optogenetic manipulations of NAcLat→VTA pathway during feeding behaviors in REG mice.** (**a**) Left: sample fluorescent image showing ChR2 expression (green) in the NAcLat (DAPI: blue; aca: anterior commissure; scale bar 500 μm). Right: schematic overview showing ChR2 expression in NAcLat for all REG mice. (**b**) Left: sample fluorescent image showing optical fibre location and ChR2 expression (green) in NAcLat terminals in the VTA (TH: tyrosine hydroxylase, red; IPN: interpeduncular nucleus; scale bar 500 μm). Middle and right: schematic overviews showing locations of optical fibers for all REG mice (MM: medial mammillary nuclei). (**c**) Food consumption during the 15 min primed-feeding trial in REG and HFD mice. During the primed-feeding trial, a specific food type (jelly, chocolate, peanut butter (PB), butter, high-fat (HF) chow, chow) or water was placed in one of the cups and mice were allowed *ad libitum* access for 15 min. REG mice consumed greater amounts of hedonic foods than HFD mice. (***p < 0.001, jelly: $n_{REG}$ = 18 mice, $n_{HFD}$ = 15 mice; chocolate: $n_{REG}$ = 19 mice, $n_{HFD}$ = 15 mice; peanut butter: $n_{REG}$ = 19 mice, $n_{HFD}$ = 15 mice; butter: $n_{REG}$ = 19 mice, $n_{HFD}$ = 15 mice; HF chow $n_{REG}$ = 18 mice, $n_{HFD}$ = 14 mice; chow: $n_{REG}$ = 19 mice, $n_{HFD}$ = 15 mice; water: $n_{REG}$ = 16 mice, $n_{HFD}$ = 15 mice; two-sided unpaired Student's t-test for each food type). (**d**) Optogenetic stimulation of NAcLat→VTA promotes hedonic feeding behaviour in a frequency-dependent manner. Jelly consumption was measured on 3 different days in response to light stimulation of NAcLat terminals in the VTA. Same assay as in Fig. 2, except that, on each day, a different light stimulation frequency was used (1 Hz, 10 Hz, or 20 Hz). High frequency (10 Hz and 20 Hz) but not low frequency (1 Hz) light stimulation increased jelly consumption in REG mice (*p < 0.05, n = 10 mice, 1-way RM ANOVA with Holm-Šídák's multiple comparisons test). (**e**) 24-hour food deprivation (FD) in REG mice significantly increases chow consumption in the primed-feeding trial when compared to sated mice (**p = 0.0041, $n_{sated}$ = 6 mice, $n_{FD}$ = 14 mice, two-sided unpaired Student's t-test). (**f**) Optogenetic stimulation of NAcLat→VTA does not increase consumption of standard chow in food-deprived mice that are subjected to a regular diet ($n_{ChR2}$ = 7 mice, $n_{eYFP}$ = 8 mice, 2-way RM ANOVA). (**g**) Mice consumed significantly less butter (B) adulterated with quinine (Q) during the primed-feeding trial when compared to regular butter consumption (*p = 0.017, $n_{butter}$ = 10 mice, $n_{butter+quinine}$ = 14 mice, two-sided unpaired Student's t-test). (**h**) Optogenetic stimulation of NAcLat→VTA does not affect consumption of butter adulterated with quinine (p > 0.05, $n_{ChR2}$ = 7, $n_{eYFP}$ = 7 mice, 2-way RM ANOVA). (**i**) Body weight of mice expressing ChR2 or eYFP in the NAcLat→VTA pathway was similar before and after the series of optogenetic stimulation experiments shown in Fig. 2 (p > 0.05, $n_{eYFP}$ = 9 mice,

$n_{ChR2}$ = 10 mice, 2-way RM ANOVA). (**j**) Left: on day 1, jelly consumption was measured for 15 min during the primed-feeding trial. 24 h later (day 2), the primed-feeding trial was repeated, but this time, mice received optogenetic stimulation of the NAcLat→VTA pathway. Right: jelly consumption was higher with optogenetic stimulation of the NAcLat→VTA pathway (on day 2) relative to baseline levels of jelly consumption measured on day 1 (**p = 0.008, $n_{ChR2}$ = 10 mice, $n_{eYFP}$ = 8 mice, 2-way RM ANOVA with Holm-Šídák's multiple comparisons test). (**k**) Experimental design for optogenetic stimulation of NAcLat cell bodies and NAcLat terminals in VTA. AAV-hSyn-ChR2 was injected into the NAcLat, and optical fibers were implanted above the NAcLat (i.e., NAcLat fibre) and above the VTA (i.e., VTA fibre). Mice were subjected to the same behavioural paradigm as shown in Fig. 2, except that on Day 1 NAcLat terminals in the VTA were stimulated and, on Day 2, NAcLat cell bodies were stimulated. (**l**) Jelly consumption during the primed-feeding trial did not differ between Days 1 and 2 (n = 6 mice, two-sided paired Student's t-test). (**m**) Optogenetic stimulation (20 Hz, 5 ms pulses) of NAcLat terminals in the VTA, but not NAcLat cell bodies, increased jelly consumption (**p = 0.001, n = 6 mice, 2-way RM ANOVA with Holm-Šídák's multiple comparisons test). (**n**) Left: Fluorescent images showing location of optical fibre in NAcLat (top, ChR2: green, DAPI: blue; scale bar 500 μm) and VTA (bottom, TH: red; scale bar 500 μm) (aca: anterior commissure, IPN: interpeduncular nucleus). Middle and right: schematic overview showing locations of optical fibres in NAcLat (top) and VTA (bottom) for all mice. Green indicates ChR2 expression. (**o**) Left: experimental design for optogenetic inhibition of NAcLat→VTA in REG mice. AAV-CaMKII-ArchT or AAV-hSyn-eYFP was injected into the NAcLat, and two optical fibers were implanted above the VTA. Right: experimental timeline: Mice were subjected for 3 consecutive days to the primed-feeding trial, but constant laser light was only delivered on day 2. (**p**) Jelly consumption in mice expressing ArchT or eYFP, with (day 2) and without (day 1 and day 3) constant laser light ($n_{ArchT}$ = 7 mice, $n_{eYFP}$ = 8 mice, *p = 0.032, 2-way RM ANOVA). (**q, s**) Left: sample fluorescent image showing ArchT (q) or eYFP (s) expression (green) in the NAcLat (DAPI: blue, scale bars 1 mm). Right: schematic overviews showing ArchT (q) ot eYFP (s) expression (green) for all mice. (**r, t**) Left: sample fluorescent image showing optical fibre location and ArchT (r) or eYFP (t) expression (green) in NAcLat terminals in the VTA (TH: tyrosine hydroxylase, red; IPN: interpeduncular nucleus; scale bars 500 μm). Middle and right: schematic overviews showing locations of optical fibres and ArchT (r) or eYFP (t) expression (green) for all mice (MM: medial mammillary nuclei). All data represented as mean ± SEM.

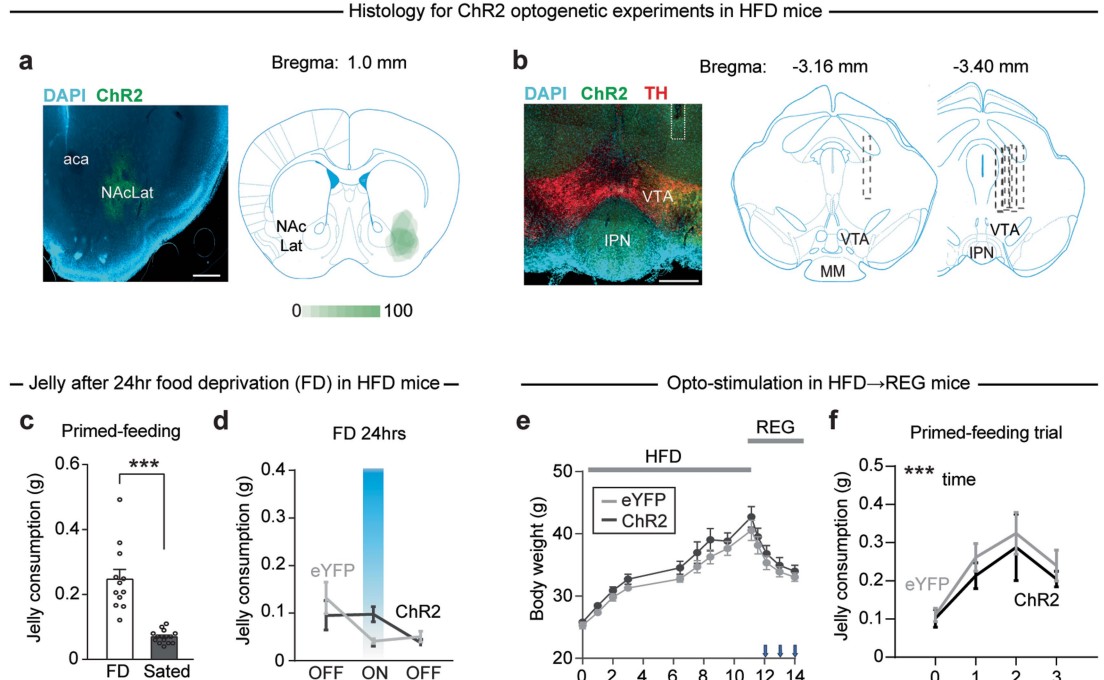

**Extended Data Fig. 4 | Optogenetic stimulation of NAcLat → VTA does not promote hedonic feeding behavior in HFD mice. (a)** Left: sample fluorescent image showing ChR2 expression (green) in the NAcLat (DAPI: blue; aca: anterior commissure; scale bar 500 µm). Right: schematic overview showing ChR2 expression (green) in NAcLat for all HFD mice. (**b**) Left: sample fluorescent image showing optical fibre location and ChR2 expression (green) in NAcLat terminals in the VTA (TH: tyrosine hydroxylase (red), IPN: interpeduncular nucleus; scale bar 500 µm). Middle and right: schematic overviews showing locations of optical fibres for all HFD mice (MM: medial mammillary nuclei). **(c)** 24-hour food deprivation (FD) in HFD mice significantly increases jelly consumption in the primed-feeding trial when compared to sated HFD mice (***p < 0.001, $n_{sated}$ = 15 mice, $n_{FD}$ = 12 mice, two-sided unpaired Student's t-test).

**(d)** Optogenetic stimulation of NAcLat→VTA does not increase consumption of jelly in food deprived HFD mice ($n_{HFD-ChR2}$ = 6 mice, $n_{HFD-eYFP}$ = 6 mice, 2-way RM ANOVA). **(e)** Body weight measured at different time points when ChR2 or eYFP mice were subjected to HFD and then returned to REG (1-way RM ANOVA, $n_{ChR2}$ = 8 mice, $n_{eYFP}$ = 7 mice). Blue arrows indicate time points when optogenetic experiments shown in Fig. 2g were performed. **(f)** Consumption of jelly during the primed-feeding trial in mice expressing ChR2 (black) or eYFP (grey) when mice were removed from HFD and returned to REG; tested at different time points over 3 weeks (***p = 0.0005, $n_{ChR2}$ = 8 mice, $n_{eYFP}$ = 7 mice, 2-way RM ANOVA with Holm-Šídák's multiple comparisons test; refers to Fig. 2g). All data represented as mean ± SEM, except panel (c), which is presented as median, 25th percentile and 75th percentile.

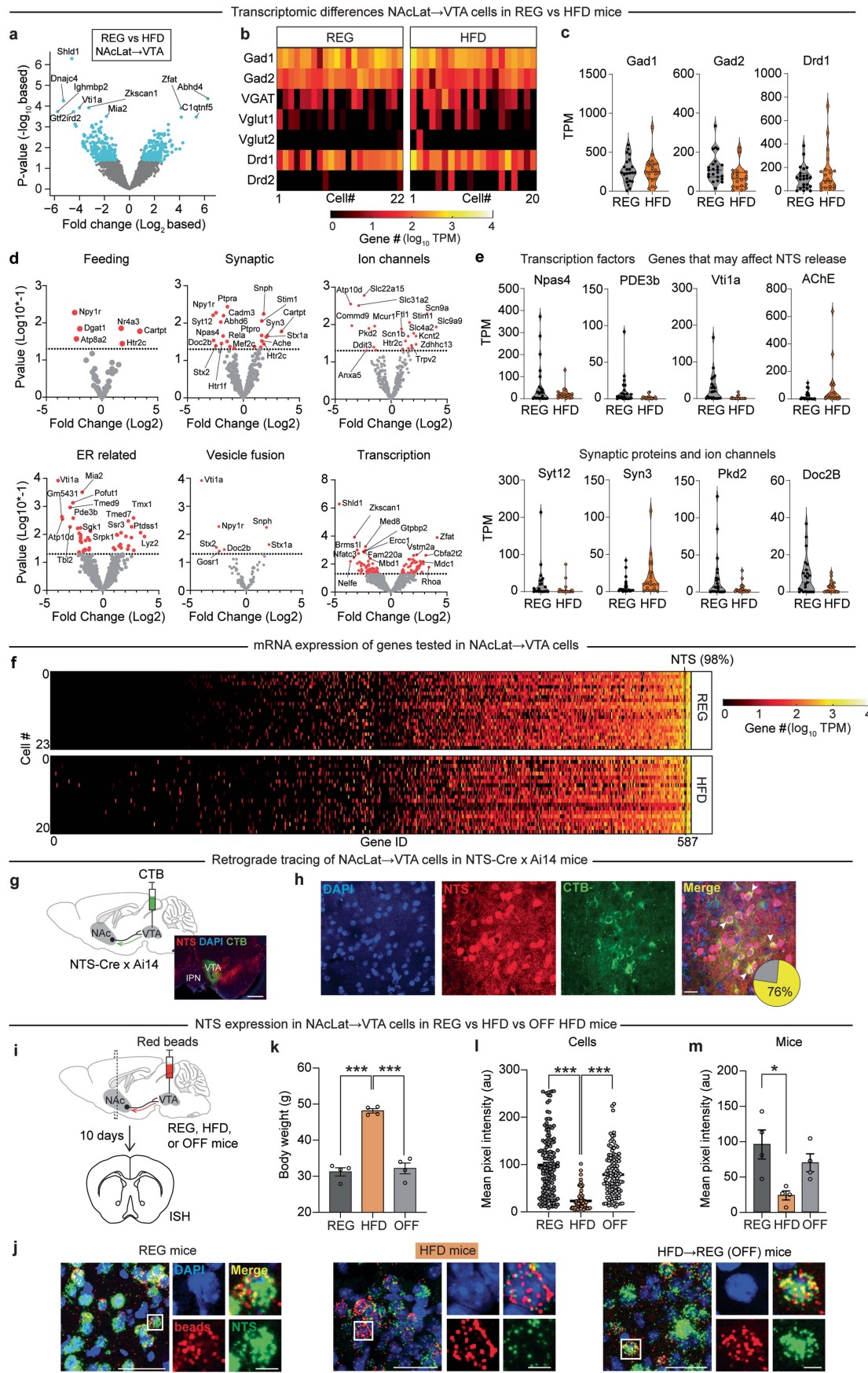

**Extended Data Fig. 5** | See next page for caption.

**Extended Data Fig. 5 | Gene expression analysis in NAcLat → VTA cells from REG and HFD mice. (a)** Volcano plot displaying differential gene expression in NAcLat→VTA cells between REG and HFD mice. Light blue data points indicate genes that are significantly differentially expressed in cells from REG versus HFD mice (i.e., absolute value of $\log_2$ (fold change) >1 and p < 0.05). The 10 genes with the highest expression differences are highlighted. **(b)** Each column displays the relative expression of GABAergic (Gad1, Gad2, VGAT) and glutamatergic (Vglut1, Vglut2) cell markers as well as dopamine receptors (Drd1, Drd2) for individual NAcLat→VTA cells (rows) from REG (left) and HFD (right) mice. Statistical significance was determined using a two-sided hypothesis. Values were not corrected for multiple comparisons. **(c)** Violin plots show expression of individual genes: Gad1 (left), Gad2 (middle), Drd1 (right) (REG: n = 23 cells from n = 10 mice, HFD: n = 20 cells from n = 11 mice). **(d)** Volcano plots showing differential gene expression between NAcLat→VTA cells from REG and HFD mice (REG: n = 23 cells from n = 10 mice, HFD: n = 20 cells from n = 11 mice). Points plotted indicate genes that are (red) or are not (gray) significantly differentially expressed between REG and HFD cells. Individual plots show subsets of genes related to feeding (top left), synaptic transmission (top middle), ion channels (top right), endoplasmic reticulum (ER) (bottom left), vesicle fusion (middle right), transcription factors (bottom right). Genes marked in red are significantly differentially regulated by the criteria: absolute value of $\log_2$ (fold change) > 1 and p < 0.05. Statistical significance was determined using a two-sided hypothesis. Values were not corrected for multiple comparisons. **(e)** Violin plots showing expression of selected genes expressed in NAcLat→VTA cells that significantly differ between REG and HFD mice. Top: Npas4, PDE3b, Vti1a, ACHE; bottom: Syt12, Syn3, Pdk2, doc2b (REG: n = 23 cells from n = 10 mice, HFD: n = 20 cells from n = 11 mice). **(f)** Each row displays the relative expression of genes for individual NAcLat→VTA cells from REG (top) and HFD (bottom) mice. Each column displays a single gene that is related to synaptic signalling and feeding (587 genes are sorted according to median value; NTS gene is in the 98 percentiles, highlighted by vertical line) (REG: n = 23 cells from n = 10 mice, HFD: n = 20 cells from n = 11 mice). **(g)** Schematic showing retrograde tracing of VTA-projecting NAcLat neurons in NTS-Cre mice crossed to an Ai14 (tDTomato) reporter mouse line. Cholera toxin subunit B (CTB, 300 nl) was injected into the VTA. CTB and tdTomato (i.e., NTS, red) labeling was analysed in the NAcLat using a confocal microscope. Inset: fluorescent image showing injection-site of CTB (green) in the VTA (DAPI: blue; IPN: interpeduncular nucleus; scale bar 1 μm). **(h)** High magnification confocal images showing DAPI (blue), NTS (red), and CTB (green) labelling in the NAcLat. White arrows indicate example NAcLat cells that express tdTomato (i.e., NTS-expressing) and are labelled with CTB (i.e., projecting to VTA); scale bar 50 μm. Inset: ~76% of NAcLat→VTA cells express NTS (yellow, 469/618 cells, n = 3 mice). **(i)** Schematic showing injection of red fluorescent retrobeads into the VTA in different cohorts of mice that had been on a regular diet (REG) or 4 weeks of high-fat diet (HFD) or have been switched from 4 weeks of HFD back to a regular diet for 3 weeks (OFF). Ten days later, brains were extracted, and NTS mRNA expression was assessed in retro-beads labelled NAcLat cells using in situ hybridization (ISH). **(j)** Confocal images showing DAPI (blue), NTS (green) and retrobead (red) labeling in the NAcLat for REG, HFD, OFF mice. White boxes indicate areas of magnification (scale bars 50 μM in lower magnification (left) and 10 μm in higher magnification (right) images). **(k)** Body weight of mice on regular diet (REG), 4 weeks on high-fat diet (HFD), or 4 weeks on HFD and then returned to regular diet for 3 weeks (OFF) (n = 4 mice in each group, ***p < 0.001; 1-way ANOVA with Holm-Šídák's multiple comparisons test). **(l)** Mean NTS pixel intensity in retrobead-positive NAcLat cells for REG, HFD, and OFF mice (REG: n = 148 cells from n = 4 mice, HFD: n = 76 cells from n = 4 mice, OFF: n = 123 cells from n = 4 mice, ***p < 0.001, Kruskal-Wallis test with Dunn's multiple comparisons test). **(m)** Averaged mean pixel intensity for NTS for each mouse from different diet groups (n = 4 mice in each group, *p = 0.018, Kruskal-Wallis test with Dunn's multiple comparisons test). All data represented as mean ± SEM.

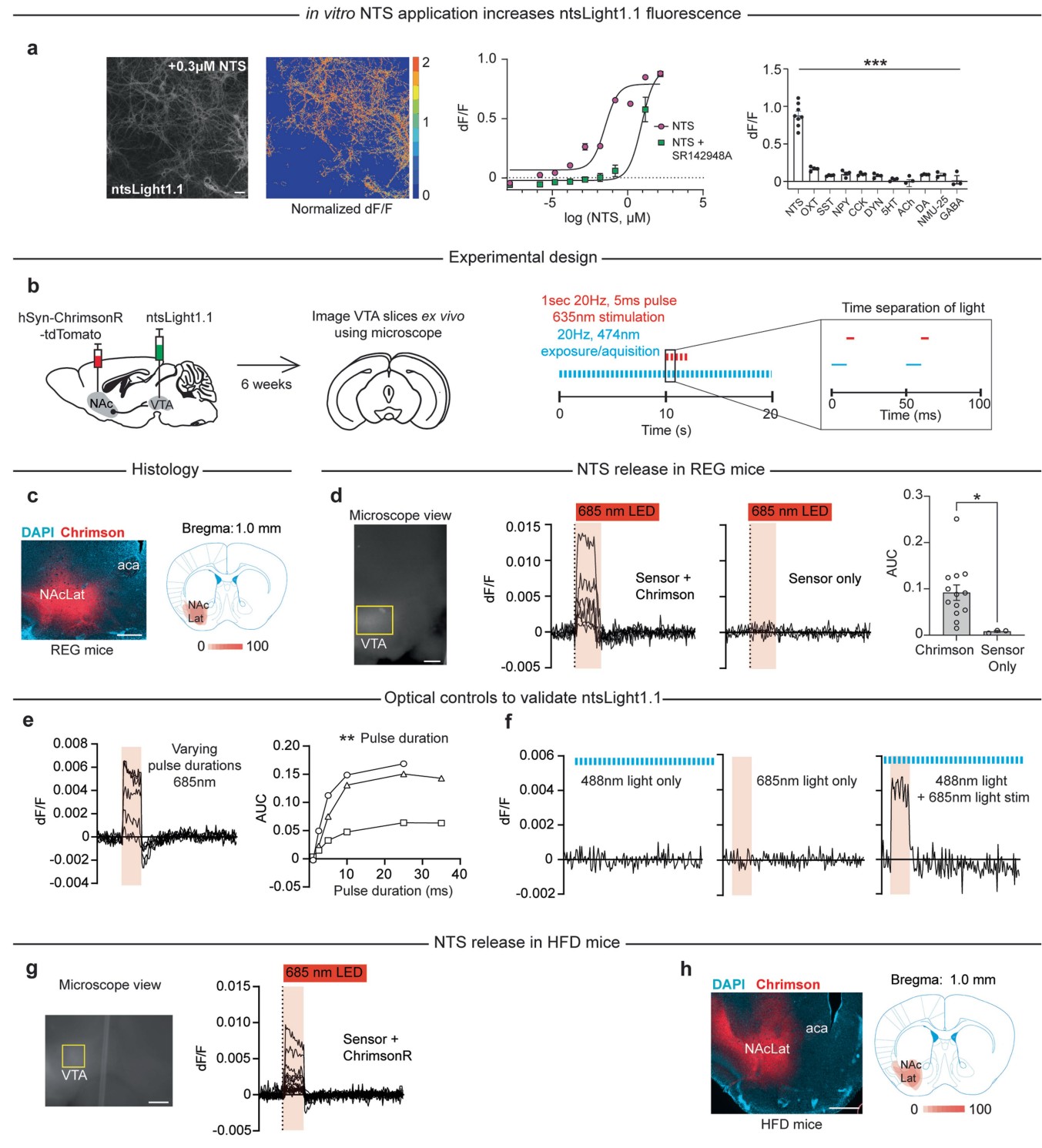

**Extended Data Fig. 6** | See next page for caption.

**Extended Data Fig. 6 | In vitro and ex vivo recordings of NTS release using ntsLight1.1.** (a) Left: ntsLight1.1 expression in hippocampal primary cultured neurons; scale bar 50 μm. Middle: ntsLight1.1 fluorescence increased in a NTS concentration dependent manner (pink). The dose-response sensitivity of ntsLight1.1 fluorescence was reduced in the presence of NTS receptor antagonist (green, 100 nM SR142948A). Right: ntsLight1.1 increased fluorescence in response to NTS but with low response to Dynorphin, GABA, Neuromedin N, and other neuropeptides/transmitters. (NTS: Neurotensin n = 8, OXT: Oxytocin, n = 4; SST: Somatostatin, n = 4; NPY: Neuropeptide Y, n = 4; CCK: Cholecystokinin, n = 4; DYN: Dynorphin, n = 4; 5HT: Serotonin (5-hydroxytryptamine), n = 4; ACh: Acetylcholine, DA: Dopamine, n = 4; NMU-25: Neuromedin U-25, n = 3; GABA: Gamma-aminobutyric acid) n = 3 (***$p < 0.001$; 1-way RM ANOVA with Holm-Šídák's multiple comparisons test). (b) Left: experimental design for ex vivo imaging of NTS release using ntsLight1.1. AAV-hSyn-ChrimsonR-tdTomato was injected into the NAcLat and ntsLight1.1 was injected into the VTA of REG mice. Six weeks later, acute VTA slices were prepared for fluorescent imaging using an epifluorescence macroscope. Right: light stimulation and imaging protocol. (c) Left: sample microscopy image of Chrimson (red) injection site in NAcLat from a REG mouse (DAPI: blue; aca: anterior commissure; scale bar 500 μm). Right: schematic showing Chrimson expression in NAcLat across all REG mice. (d) Left: sample microscope image showing ntsLight1.1 recordings in REG mice highlighting the region of interest (ROI) in the lateral VTA (scale bar 200 μm). Middle: example ntsLight1.1 fluorescence extracted from the lateral VTA of REG mice injected with Chrimson in the NAcLat and ntsLight1.1 in the VTA (middle-left) or for REG mice injected with only ntsLight1.1 in the VTA (middle-right). Right: only mice expressing both Chrimson in the NAcLat and ntsLight1.1 in the VTA show increased fluorescence during red light stimulation (*$p = 0.035$, Chrimson and ntsLight1.1: n = 13 slices from n = 5 mice; ntsLight1.1 only: n = 3 slices from n = 1 mouse; two-sided unpaired Student's t-test). (e) Left: mean ntsLight1.1 response to 1 s of 20 Hz red light stimulation with varying pulse durations. ntsLight1.1 fluorescence reaches apparent max fluorescence at ≥10 ms pulse duration. Right: ntsLight1.1 fluorescence corresponds with max activation of Chrimson rather than max amount of light delivered to VTA (**$p = 0.0012$, effect of pulse duration on ntsLight1.1 AUC, n = 3 slices from n = 1 mouse, 2-way RM ANOVA). (f) Recordings from a single VTA ROI expressing ntsLight1.1. 20 Hz continuous blue (473 nm) light alone, which only excites ntsLight1.1 (left), and 1 s of 20 Hz red (685 nm) light alone, which only activates Chrimson (middle), and did not increase ntsLight1.1 fluorescence. Only the combination of blue and red light increased ntsLight1.1 fluorescence (right). (g) Left: sample image from ntsLight1.1 imaging experiment showing ROI in lateral VTA for measuring ntsLight1.1 fluorescence in a HFD mouse (scale bar 200 μm). Right: graph showing that red light stimulation of NAcLat terminals in VTA increased ntsLight1.1 fluorescence in the lateral VTA of HFD mice. (h) Left: Sample microscopy image of Chrimson (red) injection site in NAcLat from a HFD mouse (DAPI: blue; aca: anterior commissure; scale bar 500 μm). Right: schematic showing Chrimson expression in NAcLat across all HFD mice. All data represented as mean ± SEM.

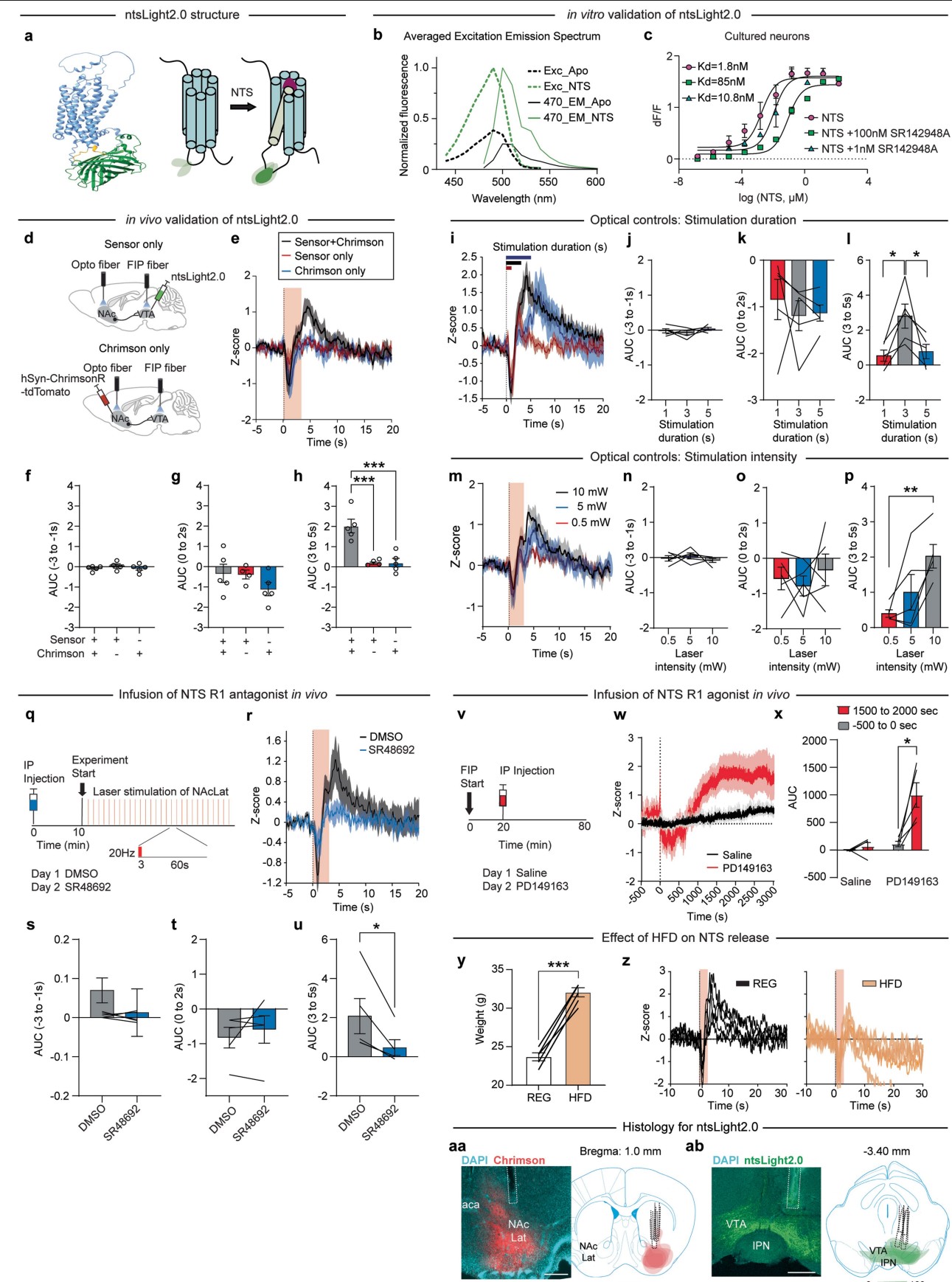

**Extended Data Fig. 7 |** See next page for caption.

**Extended Data Fig. 7 | In vitro and in vivo recordings of NTS release using ntsLight2.0. (a)** Left: a simulated structure of ntsLight2.0. Right: schematic showing ntslight2.0 activation in that NTS binding induces a conformational change in the sensor that increases fluorescence. **(b)** Excitation and emission spectrum of ntsLight2.0 with (green) or without (black) NTS. **(c)** ntsLight2.0 fluorescence increased in a NTS concentration dependent manner (pink). The dose-response sensitivity of ntsLight2.0 fluorescence was reduced proportionally to the concentration of the added NTS receptor antagonist (green, 100 nM (squares) and 1 nM (triangles) SR142948A). **(d)** Experimental design for ntsLight2.0 validation using in vivo fibre photometry (FIP). Top: sensor only: ntslight2.0 was injected into the VTA. Bottom: Chrimson only: AAV-hSyn-ChrimsonR-tdTomato was injected into the NAcLat). Not shown: sensor + Chrimson: AAV-hSyn-ChrimsonR-tdTomato was injected into the NAcLat and ntslight2.0 was injected into the VTA. Optical fibres were implanted above the NAcLat and above the VTA in all experimental conditions. Six weeks later, NTS release was recorded in the VTA using fibre photometry during optogenetic simulation of the NAcLat in head-fixed mice. **(e)** ntsLight2.0 transients recorded in the VTA 5 s before and 20 s after 3-second opto-stimulation of the NAcLat in mice for 3 different experimental conditions: sensor only (red, n = 4 mice), Chrimson only (blue, n = 5 mice), and sensor + Chrimson (grey, n = 5 mice). Data are Z-scored, averaged across 30 trials, t = 0 s indicates stimulation onset. **(f-h)** AUCs for sensor only (red), Chrimson only (blue), and both sensor and Chrimson (gray) groups when analysing the time period before stimulation onset (f) (−3 to −1 s), during laser stimulation (g) (0 to 2 s) and after stimulation (h) (3 to 5 s). Mice expressing both sensor and Chrimson showed a significant increase in the Z-scored average of ntsLight2.0 fluorescence in the 3 to 5 s interval (***$p < 0.0009$, $n_{sensor}$ = 4 mice, $n_{Chrimson}$ = 5 mice, $n_{sensor+Chrimson}$ = 5 mice, one-way ANOVA). **(i)** ntslight2.0 transients during optogenetic stimulation of NAcLat with different durations (red: 1 s, black: 3 s, blue: 5 s stimulation duration). Experiments were performed in mice expressing both sensor and Chrimson (n = 5 mice). The graph displays Z-scored averages from n = 30 trials. **(j-l)** AUCs for 1 s (red), 3 s (grey) and 5 s (blue) stimulation duration analysed before stimulation onset (j) (−3 to −1 s), during laser stimulation (k) (0 to 2 s), and after stimulation (l) (3 to 5 s). 3 s light stimulation produced the largest increase in ntsLight2.0 transients in the 3 to 5 s interval (*$p_{1-3}$ = 0.03, *$p_{3-5}$ = 0.034, n = 5 mice, one-way ANOVA). **(m)** ntsLight2.0 transients during optogenetic stimulation of NAcLat with different stimulation intensities (red: 0.5 mW, blue: 5 mW, black: 10 mW). Experiments were performed in mice expressing both sensor and Chrimson (n = 5 mice). The graph displays Z-scored

averages from n = 30 trials. **(n-p)** AUCs for 0.5 mW (red), 5 mW (blue), and 10 mW (grey) stimulation intensities analyzed before stimulation onset (n) (−3 to −1 s), during laser stimulation (o) (0 to 2 s) and after stimulation (p) (3 to 5 s). 10 mW stimulation produced the largest ntsLight2.0 transients in the 3–5 s interval (**$p = 0.009$, n = 5 mice, one-way ANOVA). **(q)** Mice received either DMSO or NTS R1 antagonist SR48692 injections (IP, 5 mg/kg) 10 min prior to FIP recordings. Experiments were performed in sensor + Chrimson mice (n = 5 mice) with n = 30 trials of NAcLat Chrimson stimulation (3 s, 10 mW, 20 Hz) while recording ntsLight2.0 transients in the VTA. **(r)** ntsLight2.0 transients recorded in the VTA 5 s before and 20 s after 3 s optogenetic stimulation of NAcLat in DMSO (back) and SR48692 (blue) injected animals. The graph displays Z-scored averages from n = 30 trials. Experiments were performed in mice expressing both sensor and Chrimson (n = 5 mice). **(s-u)** AUCs before (s) (−3 to −1 s), during (t) (0 to 2 s), and after (u) (3 to 5 s) NAcLat stimulation in mice injected with DMSO (gray) or SR48692 (blue). In SR48692 injected animals, ntsLight2.0 transients were significantly decreased in the 3 to 5 s interval compared to DMSO injected mice (*$p = 0.0358$, n = 5 mice, two-sided paired Student's t-test). **(v)** Experimental design. Baseline ntsLight2.0 signal was recorded for 20 min. Subsequently, mice were injected with either saline (day 1, IP) or an NTS R1 agonist (PD149163, 0.3 mg/kg IP, on day 2) and ntslight2.0 transients were recorded for an additional 50 min. **(w)** ntslight2.0 transients (averaged Z-scores) in response to saline (black) and PD149163 (red) injections. Experiments were performed in mice expressing both sensor and Chrimson (n = 5 mice). **(x)** AUCs before (−500 to 0 s, gray) and after (1500 to 2000 s, red) injection of saline or PD149163. PD149163, but not saline, significantly increased ntsLight2.0 transients in the VTA (*$p = 0.017$, n = 5 mice, 2-way RM ANOVA with Šídák's multiple comparisons test). **(y)** Body weight of mice on REG and 4 weeks of HFD mice for experiment shown in Fig. 3n–p, ***$p = 0.0001$, n = 5 mice, two-sided paired Student's t-test). **(z)** Individual ntsLight2.0 traces during optogenetic stimulation of NAcLat in REG (left, black) and HFD (right, orange) mice (refers to Fig. 3o; n = 5 mice). **(aa)** Left: sample fluorescent image of Chrimson (red) injection site in NAcL at and optical fibre location (DAPI: blue; aca: anterior commissure; scale bar 500 μm). Right: schematic showing Chrimson expression (red) in NAcLat and optical fibre locations across all mice. **(ab)** Left: Sample fluorescent image showing optical fibre location and ntsLight2.0 expression (green) in the VTA (IPN: interpeduncular nucleus; scale bar 500 μm). Right: schematic overviews showing locations of optical fibres and Chrimson expression in NAcLat terminals (green) in the VTA across all mice. All data represented as mean ± SEM (error bars or shading).

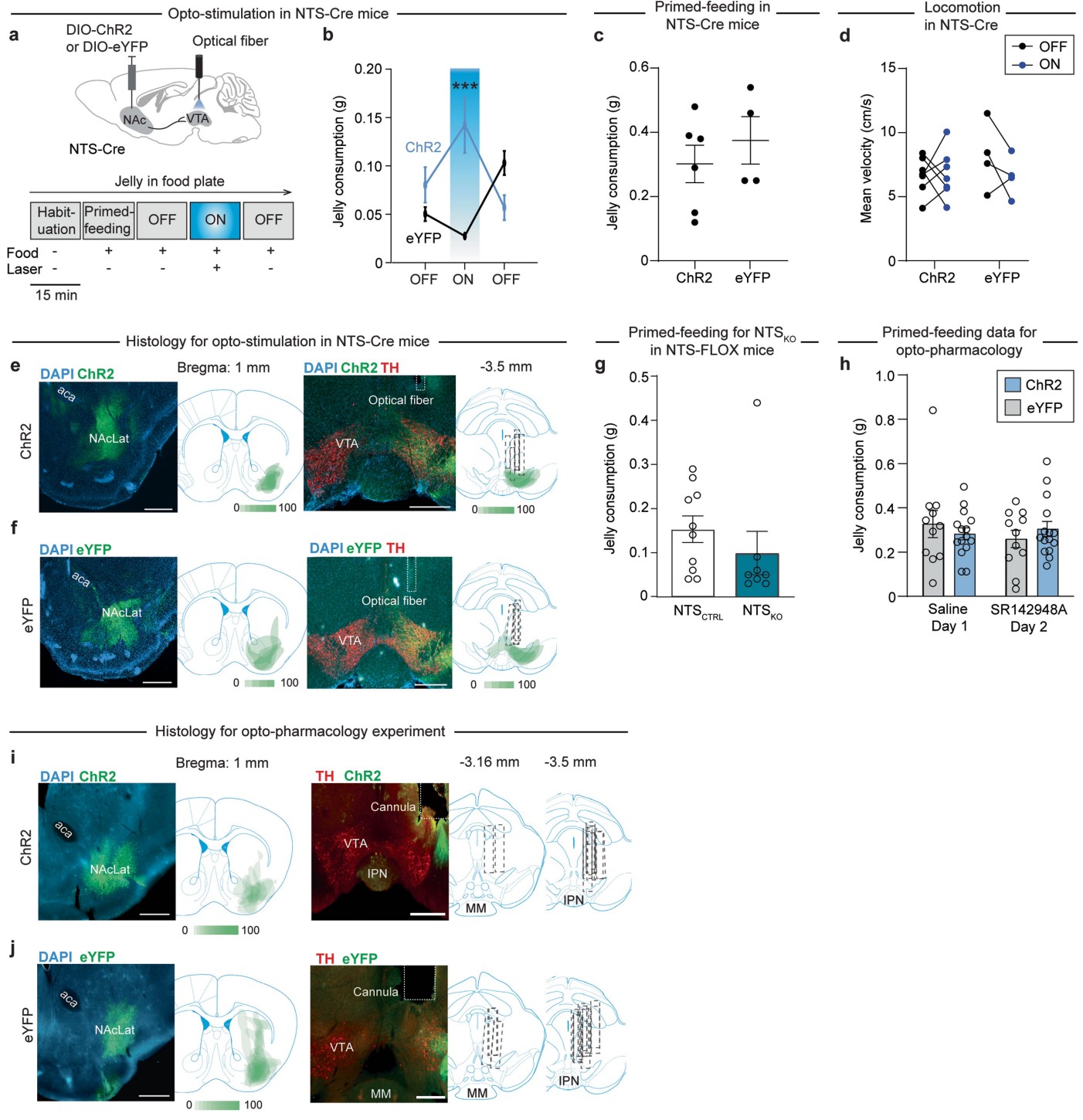

**Extended Data Fig. 8** | See next page for caption.

**Extended Data Fig. 8 | NTS release in the NAcLat → VTA pathway promotes hedonic feeding behavior. (a)** Top: experimental design showing AAV-DIO-ChR2 (ChR2) or AAV-DIO-eYFP (eYFP) injections into the NAcLat and implantation of an optical fiber above the VTA in NTS-CRE mice. Bottom: experimental timeline of acute feeding assay (same as in Fig. 2). **(b)** Optogenetic stimulation (20 Hz, 5 ms pulses) of NAcLat terminals in the VTA increased jelly consumption in NTS-Cre mice expressing ChR2 but not eYFP (***$p = 0.0009$, $n_{ChR2} = 6$ mice, $n_{eYFP} = 4$ mice, 2-way RM ANOVA with Holm-Šídák's multiple comparisons test). **(c)** NTS-Cre mice expressing ChR2 and eYFP consumed similar amounts of jelly during the primed-feeding trial ($p > 0.05$, $n_{ChR2} = 6$ mice, $n_{eYFP} = 4$ mice, two-sided unpaired Student's t-test). **(d)** No significant change in mean velocity of mice in the open-field test during stimulation of NAcLat inputs for both NTS-Cre mice expressing ChR2 or eYFP ($p > 0.05$, $n_{ChR2} = 6$ mice, $n_{eYFP} = 4$ mice, 2-way RM ANOVA). **(e-f)** Histology for optogenetic stimulation in NTS-Cre mice. Left: sample fluorescent images showing ChR2 (e) or eYFP (f) expression (green) in the NAcLat and the corresponding schematics for ChR2 or eYFP expression analyses across all animals. Right: sample fluorescent images showing optical fibre locations and ChR2-expressing NAcLat terminals (green) in the VTA and the corresponding schematics across all mice (scale bars 500 μm, DAPI: blue, TH: red, aca: anterior commissure). **(g)** NTS-FLOX mice expressing Cre and ChR2 ($NTS_{KO}$) or ChR2 ($NTS_{CTRL}$) in the NAcLat consumed similar amounts of jelly during the primed-feeding trial ($p > 0.05$, $n_{NTS-KO} = 8$ mice, $n_{NTS-CTRL} = 10$ mice, two-sided unpaired Student's t-test). **(h)** Jelly consumption during the primed-feeding trial for mice in the opto-pharmacology experiment. There was no significant difference in jelly consumption between ChR2 and eYFP mice infused with either saline or SR142948A ($p > 0.05$, $n_{ChR2} = 11$ mice, $n_{eYFP} = 15$ mice, 2-way RM ANOVA). **(i, j)** Histology for opto-pharmacology experiment. Left: sample fluorescent images showing ChR2 (i) or eYFP (j) expression (green) in the NAcLat and the corresponding schematics across all mice. Right: sample fluorescent images showing cannula implant locations and ChR2 (i) or eYFP (j) expressing NAcLat terminals in the VTA and the corresponding schematics across all mice (scale bars 500 μm, DAPI: blue, TH: red; MM: medial mammillary nuclei, IPN: interpeduncular nucleus, aca: anterior commissure). All data represented as mean ± SEM.

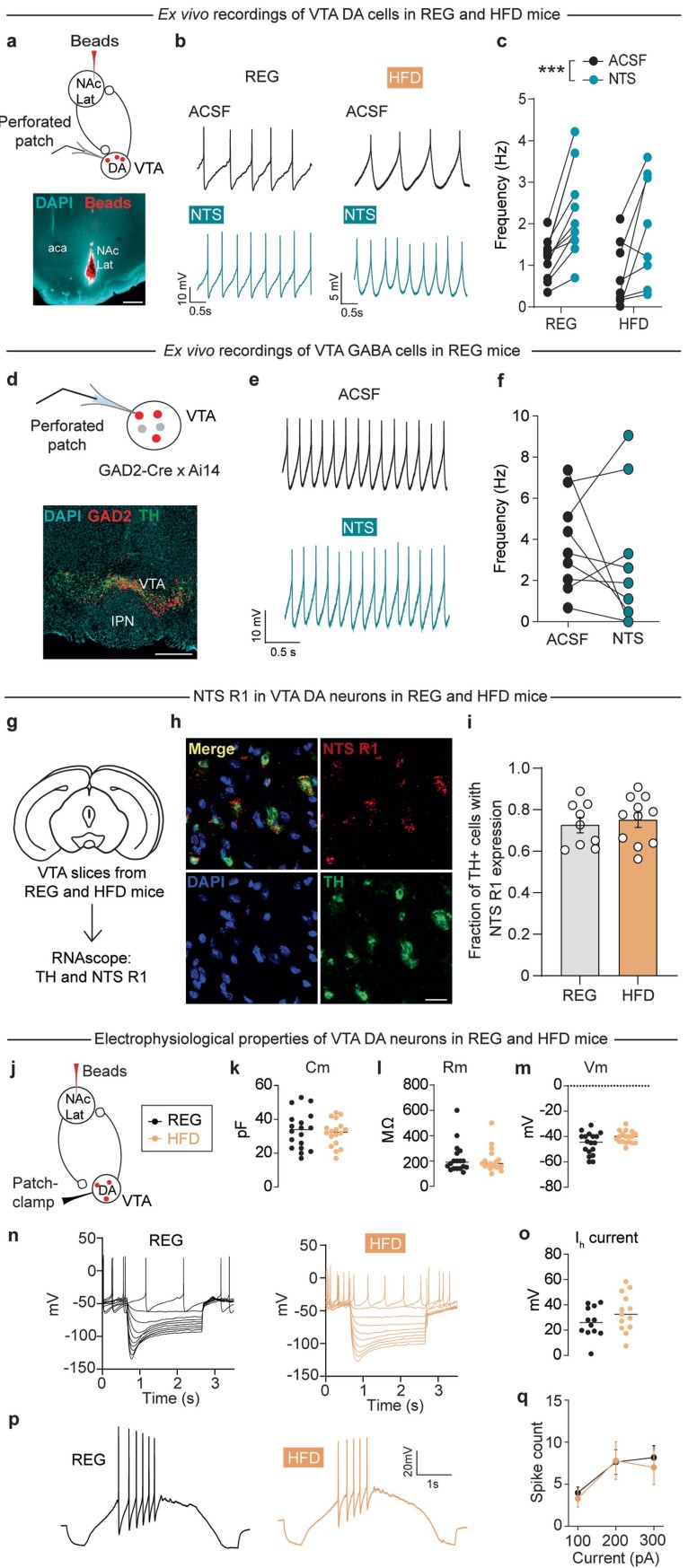

**Extended Data Fig. 9** | See next page for caption.

**Extended Data Fig. 9 | HFD does not induce changes in NTS function in VTA dopamine and GABAergic cells. (a)** Top: schematic of experimental design. Red fluorescent retrobeads were injected into the NAcLat. One week later, retrobeads-labelled VTA cells were recorded in acute brain slices using perforated-patch. Bottom: injection-site of beads (red) in NAcLat (DAPI: turquoise; scale bar 500 μm). **(b)** Sample recordings from a retrobead-labelled VTA dopamine (DA) cell from REG (left) and HFD (right) mice in ACSF (black) and during bath application of neurotensin (NTS, 1 μM, turquoise). **(c)** Firing rate of NAcLat-projecting VTA DA neurons before (ACSF, black) and during NTS (turquoise) application for REG and HFD mice (***p = 0.001 effect of NTS, no effect for diet type, $n_{REG}$ = 11 cells from n = 5 mice, $n_{HFD}$ = 8 cells from n = 6 mice, 2-way RM ANOVA with Holm-Šídák's multiple comparisons test). **(d)** Top: schematic showing that perforated patch recordings were performed from tdTomato-expressing cells in the VTA of GAD2-Cre mice crossed to an Ai14 reporter mouse line. Bottom: sample fluorescent image showing tdTomato expression (i.e., GAD2-expressing cells, red) in the VTA (DAPI: blue, TH: green; scale bar 500 μm, IPN: interpeduncular nucleus). **(e)** Sample perforated patch recording from a VTA GABA cell before (ACSF) and after bath application of neurotensin (1 μM NTS, turquoise). **(f)** Application of 1 μM NTS did not significantly change the firing rate of VTA GABA cells when compared to baseline (p = 0.4191, n = 9 cells from n = 5 mice, two-sided paired Student's t-test). **(g)** Experimental design to analyse TH and NtsR1 mRNA expression in VTA cells from REG and HFD mice. **(h)** Sample microscopy image of fluorescent in situ hybridization showing DAPI (blue), NtsR1 (neurotensin receptor 1, red), and TH (tyrosine hydroxylase, green) in the VTA. Scale bar: 20 μm. **(i)** Fraction of VTA TH-positive cells that co-localize with NtsR1 for REG and HFD mice (p = 0.62, $n_{REG}$ = 42 slices from n = 9 mice, $n_{HFD}$ = 41 slices from n = 11 mice,

two-sided unpaired Student's t-test). **(j)** Schematic of experimental design showing that red fluorescent retrobeads were injected into the NAcLat and patch clamp recordings were performed from beads-containing VTA dopamine (DA) cells in mice subjected to a regular (REG) or high-fat diet (HFD). **(k)** No significant difference in the membrane capacitance of beads-labelled VTA DA cells between REG and HFD mice (p = 0.99, REG: n = 23 cells from n = 5 mice, HFD: n = 25 cells from n = 3 mice, two-sided unpaired Student's t-test). **(l)** No significant difference in the membrane resistance of beads-labelled VTA DA between REG and HFD mice (p = 0.27, REG: n = 23 cells from n = 5 mice, HFD: n = 25 cells from n = 3 mice, two-sided unpaired Student's t-test). **(m)** No significant difference in the resting membrane potential of beads-labeled VTA DA cells between REG and HFD mice (p = 0.23, REG: n = 23 cells from n = 5 mice, HFD: n = 25 cells from n = 3 mice, two-sided unpaired Student's t-test). **(n)** Whole cell, current clamp recordings of beads-labeled VTA DA neurons from REG (left) and HFD (right) mice. Note, slow regular discharge and sag components induced by injections of hyperpolarizing currents of increasing amplitudes (steps of −25 pA). **(o)** No significant difference in the sag amplitudes, a measure for the hyperpolarization-activated current ($I_h$), recorded in beads-labeled VTA DA neurons from REG and HFD mice, analysed at −150 pA (p = 0.125, REG: n = 15 cells from n = 6 mice, HFD: n = 13 cells from n = 4 mice, two-sided unpaired Student's t-test). **(p)** Whole cell, current clamp responses to 2-sec ramps of depolarizing currents ( + 150 pA) of beads-labelled VTA DA neurons from REG (left) and HFD (right) mice. **(q)** No significant difference in the number of spikes evoked by injecting depolarizing current ramps in beads-labelled VTA DA neurons between REG and HFD mice (p = 0.77, REG: n = 11 cells from n = 5 mice, HFD: n = 10 cells from n = 3 mice, 2-way RM ANOVA). All data represented as mean ± SEM.

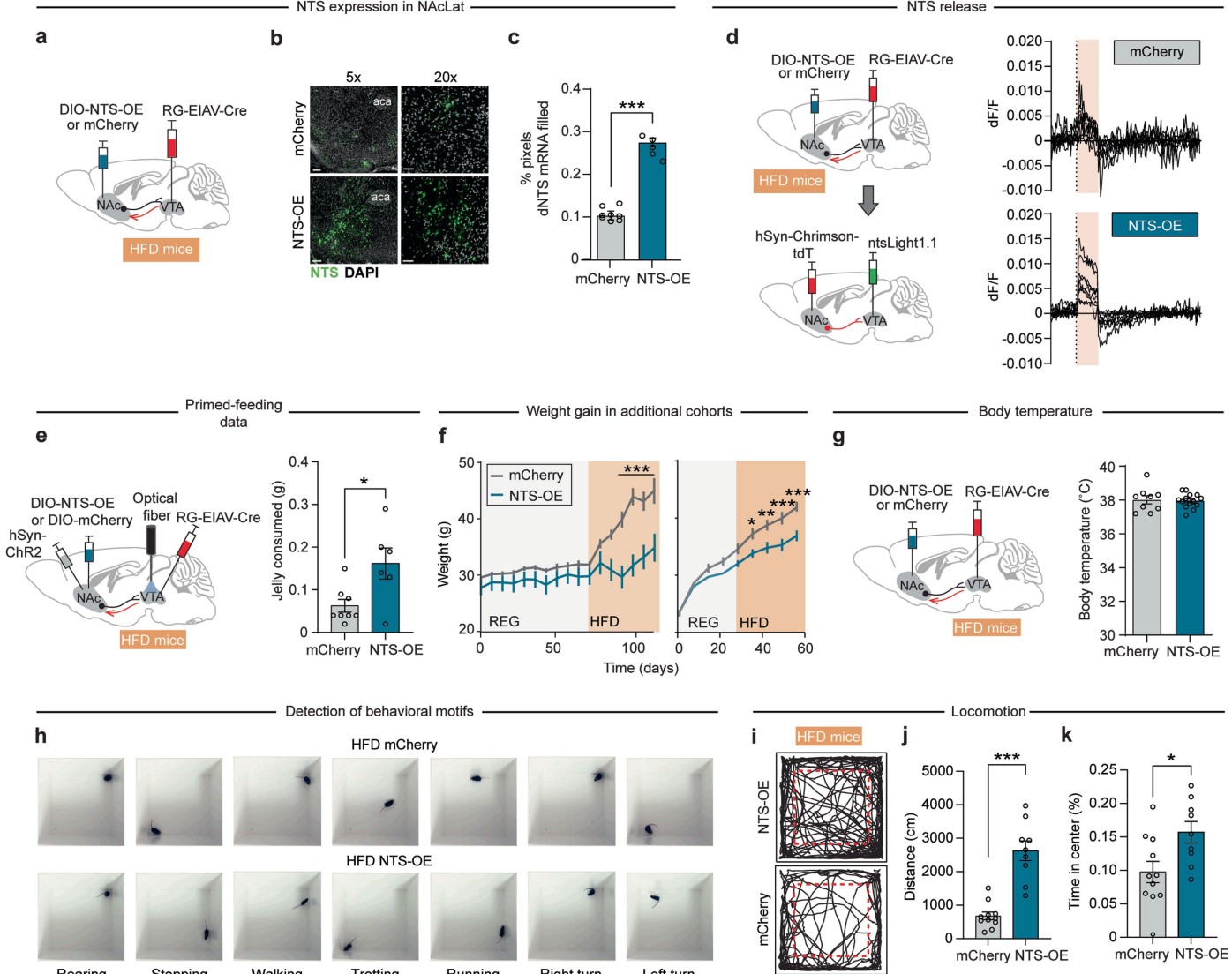

**Extended Data Fig. 10** | See next page for caption.

**Extended Data Fig. 10 | NTS overexpression in HFD mice. (a)** Schematic of experimental design to selectively overexpress NTS in NAcLat→VTA neurons. A retrograde virus (RG-EIAV-Cre) was injected into the VTA and either Cre-dependent NTS overexpression virus (AAV-DIO-NTS-OE; see methods for detailed virus description) or Cre-dependent fluorescent reporter (AAV-DIO-mCherry) was injected into the NAcLat. Mice were subjected to HFD, and 4 weeks later, NTS expression was analysed in NAcLat→VTA neurons using RNAscope. **(b)** Sample images of fluorescent in situ hybridization in the NAcLat of NTS-OE and mCherry control HFD mice (DAPI: grey, NTS: green; aca: anterior commissure; scale bars 200 μm (5x), 50 μm (20x)). **(c)** Significantly increased expression of NTS in the NAcLat of HFD mice injected with NTS-OE virus when compared to mice injected with a mCherry control vector (***$p < 0.001$, $n_{mCherry} = 14$ slices from $n = 7$ mice, $n_{NTS-OE} = 8$ slices from $n = 4$ mice, two-sided unpaired Student's t-test). **(d)** Top: schematic of experimental design showing that a retrograde virus (RG-EIAV-Cre) was injected into the VTA, and either AAV-DIO-NTS-OE or AAV-DIO-mCherry was injected into the NAcLat. 4 weeks later, these mice were injected with AAV-hSyn-Chrimson into the NAcLat and AAV9-hSyn-ntsLight1.1 into the VTA. Mice were then subjected to HFD, and 4 weeks later, ntsLight1.1 was measured in acute VTA slices. Right: sample ntsLight1.1 fluorescent traces in response to Chrimson stimulation (685 nm) extracted from the lateral VTA in slices from HFD mice expressing mCherry (top) or NTS-OE (bottom) in the NAcLat→VTA pathway ($n_{mCherry} = 10$ slices from $n = 2$ mice; $n_{NTS-OE} = 9$ slices from $n = 2$ mice; refers to Fig. 5a–c). **(e)** Left: schematic showing experimental design to optogenetically stimulate the NAcLat→VTA pathway in HFD mice that overexpress NTS in the NAcLat→VTA pathway. Mice were injected with RG-EIAV-Cre into the VTA and either AAV-DIO-NTS-OE or AAV-DIO-mCherry into the NAcLat. 4 weeks later, these mice were injected with AAV-hSyn-ChR2 into the NAcLat and an optical fibre was implanted above the VTA. Mice were then subjected to HFD and optogenetic experiments were performed 4 weeks later. Right: bar graph showing increased jelly consumption during primed-feeding trial in HFD mice that overexpress NTS in the NAcLat→VTA pathway when compared to mCherry control mice (*$p = 0.017$, $n_{NTS-OE} = 6$ mice, $n_{mCherry} = 8$ mice, two-sided unpaired Student's t-test; refers to data shown in Fig. 5d, e).
**(f)** Analysis of the effects of varying experimental conditions (i.e., baseline body weight (BW) of mice, housing scheme, and virus expression time) on BW in two additional cohorts of NTS-OE and control mice. Left (cohort 1): mice expressing mCherry or NTS-OE in the NAcLat→VTA pathway were maintained for 10 weeks on a regular diet before switching to HFD (mean BW at beginning of experiment: mCherry: $29.82 \pm 0.37$ g, $n = 4$ mice; NTS-OE: $28.2 \pm 1.48$ g, $n = 5$ mice, injected at age of 10 weeks). Right (cohort 2): mice injected with different viruses are mixed within the same home cages (mean BW at beginning of experiment: mCherry: $23.3 \pm 1.22$ g, $n = 9$ mice; NTS-OE: $23.4 \pm 0.97$ g, $n = 10$ mice, injected at age of 6 weeks). Despite varying experimental conditions across the two cohorts, NTS-OE mice on HFD consistently gained significantly less weight compared to mCherry control mice when switching from regular to high-fat diet (*$p = 0.0117$, **$p = 0.0023$, ***$p < 0.0003$, 2-way RM ANOVA with Holm-Šídák's multiple comparisons test). **(g)** Left: schematic of experimental design showing that a retrograde virus (RG-EIAV-Cre) was injected into the VTA and either AAV-DIO-NTS-OE or AAV-DIO-mCherry was injected into NAcLat. Mice were subjected to HFD and 4 weeks later the body temperature was measured. Right: bar graph showing no significant difference between the body temperatures of NTS-OE and mCherry mice ($p = 0.61$, $n_{mCherry} = 9$ mice, $n_{NTS-OE} = 14$ mice, two-sided unpaired Student's t-test; same mice as shown in Fig. 5f). **(h)** Sample behavioural motifs obtained using DeepLabCut from HFD mice that are freely behaving in an open-field chamber. These mice either express mCherry (top) or overexpress NTS (bottom) in the NAcLat→VTA pathway (refers to Fig. 5j–l). **(i)** Representative trajectories of mice expressing NTS-OE (top) and mCherry (bottom) in NAcLat→VTA in an open-field chamber for 10 min. **(j)** Bar graph showing distance travelled for NTS-OE (grey) and mCherry control (blue) groups during open-field exploration ($p < 0.001$, $n_{mCherry} = 11$ mice, $n_{NTS-OE} = 9$ mice, two-sided unpaired Student's t-test). **(k)** Bar graph showing percentage of time mice spent in the centre of the open-field chamber for NTS-OE (grey) and mCherry control (blue) groups (*$p = 0.018$, $n_{mCherry} = 11$ mice, $n_{NTS-OE} = 9$ mice, two-sided unpaired Student's t-test). All data represented as mean ± SEM.

# Reporting Summary

## Statistics

For all statistical analyses, confirm that the following items are present in the figure legend, table legend, main text, or Methods section.

| n/a | Confirmed | |
|---|---|---|
| ☐ | ☒ | The exact sample size (*n*) for each experimental group/condition, given as a discrete number and unit of measurement |
| ☐ | ☒ | A statement on whether measurements were taken from distinct samples or whether the same sample was measured repeatedly |
| ☐ | ☒ | The statistical test(s) used AND whether they are one- or two-sided<br>*Only common tests should be described solely by name; describe more complex techniques in the Methods section.* |
| ☐ | ☒ | A description of all covariates tested |
| ☐ | ☒ | A description of any assumptions or corrections, such as tests of normality and adjustment for multiple comparisons |
| ☐ | ☒ | A full description of the statistical parameters including central tendency (e.g. means) or other basic estimates (e.g. regression coefficient) AND variation (e.g. standard deviation) or associated estimates of uncertainty (e.g. confidence intervals) |
| ☐ | ☒ | For null hypothesis testing, the test statistic (e.g. *F*, *t*, *r*) with confidence intervals, effect sizes, degrees of freedom and *P* value noted<br>*Give P values as exact values whenever suitable.* |
| ☒ | ☐ | For Bayesian analysis, information on the choice of priors and Markov chain Monte Carlo settings |
| ☒ | ☐ | For hierarchical and complex designs, identification of the appropriate level for tests and full reporting of outcomes |
| ☐ | ☒ | Estimates of effect sizes (e.g. Cohen's *d*, Pearson's *r*), indicating how they were calculated |

*Our web collection on statistics for biologists contains articles on many of the points above.*

## Software and code

Policy information about availability of computer code

Data collection | PClamp 10.5 (Molecular Devices) was used for acquisition of brain slice electrophysiology data. Zen Software 2.3 (Zeiss) was used for acquiring confocal and epifluorescence images of brain slices. For validation and spectral analysis experiments of ntsLight1.1 and 2.0 cultured neurons were imaged with a Leica Stellaris 8 Confocal. For imaging of ntsLight1.1 in brain slices, a custom-built, open source macroscope (https://github.com/Llamero/DIY_Epifluorescence_Macroscope) fitted with high power LEDs and a Teledyne Kinetix sCMOS camera. Imaging of neuronal cultures was performed using MetaXpress software (Version 6.6.3.55) on the ImageXpress MicroConfocal system. Neural signals were recording used using a Digital Lynx 4SX system. ntsLight2.0 signals were recorded using a MATLAB-based custom fiber photometry system https://github.com/handejong/Fipster. For in vivo electrophysiology experiments, neural signals were recorded using a Digital Lynx 4SX system. Laser output was controlled using a Master-8 pulse stimulator (A.M.P.I.). Spikes were sorted offline using SpikeSort3D 2.5.4 (Neuralynx) software. Video-based offline tracking was performed via DeepLabCut.py (Version 2.0.7). Biobserve (Version 3.0.1.442) video tracking system was used for the real-time place preference test. In RNAseq experiments, cells were pooled and sequenced using the Illumina NovaSeq 6000 with 150 bp paired-end reads. After sequencing, raw reads were de-multiplexed using Illumina bc12fastq (Version 2.20), and pseudo-aligned to the Ensembl GRCm38.95 reference transcriptome and normalized using kallisto (Version 0.45.1).

Data analysis | Comparative statistical tests, correlation, and regression were performed in GraphPad Prism Versions 9 (Version 9.5.1) and 10 (Version 10.3.1). Gene expression analysis for RNAseq was performed using Python (Version 3.6.7), R (Version 3.5.1) and edgeR (Version 3.24.3). MouseActivity5.m was used to analyze open-field behavior (https://github.com/HanLab-OSU/MouseActivity/blob/master/MouseActivity5.m). DeepLabCut.py (Version 2.0.7) was used to analyze mouse behavior. For analysis of in vivo electrophysiology data, NeuralynxMatlabImportExport_v6.0.0 MATLAB package was used, which is available at https://neuralynx.fh-co.com/research-software/. Custom MATLAB (Version R2024a), Python (Version 3.6.7) and edgeR (Version 3.24.3) were used for the processing of in vivo electrophysiology data and RNAseq are available at https://github.com/lammellab/. ImageJ (NIH, 64-bit Java 1.8.0_172) was used for analysis of fluorescence and confocal images. For Ligand specificity test in cell cultures, images from Leica Stellaris Confocal were exported and

analyzed using a customized MATLAB (Version R2023b) script available at https://github.com/lintianlab. Brain slice electrophysiology data were analyzed offline using Clampfit (Molecular Devices, Version 10.5).

For manuscripts utilizing custom algorithms or software that are central to the research but not yet described in published literature, software must be made available to editors and reviewers. We strongly encourage code deposition in a community repository (e.g. GitHub). See the Nature Portfolio guidelines for submitting code & software for further information.

## Data

Policy information about availability of data

All manuscripts must include a data availability statement. This statement should provide the following information, where applicable:
- Accession codes, unique identifiers, or web links for publicly available datasets
- A description of any restrictions on data availability
- For clinical datasets or third party data, please ensure that the statement adheres to our policy

All source data are available in the accompanying source data files. The RNAseq datasets generated during this study are available at NCBI GEO; GSE287548.

## Research involving human participants, their data, or biological material

Policy information about studies with human participants or human data. See also policy information about sex, gender (identity/presentation), and sexual orientation and race, ethnicity and racism.

| | |
|---|---|
| Reporting on sex and gender | N/A |
| Reporting on race, ethnicity, or other socially relevant groupings | N/A |
| Population characteristics | N/A |
| Recruitment | N/A |
| Ethics oversight | N/A |

Note that full information on the approval of the study protocol must also be provided in the manuscript.

# Field-specific reporting

Please select the one below that is the best fit for your research. If you are not sure, read the appropriate sections before making your selection.

☒ Life sciences      ☐ Behavioural & social sciences      ☐ Ecological, evolutionary & environmental sciences

For a reference copy of the document with all sections, see nature.com/documents/nr-reporting-summary-flat.pdf

# Life sciences study design

All studies must disclose on these points even when the disclosure is negative.

| | |
|---|---|
| Sample size | No statistical methods were used to predetermine sample size, which were based on work in previous publications (Lammel et al., 2008; Neuron; Lammel et al., 2012; Nature; Cerniauskas et al., 2019; Neuron; Yang et al., 2021; Nature Neuroscience; Cardozo Pinto et al., 2019; Nature Communications). |
| Data exclusions | For recordings of NAcLat-VTA units during behavior (Fig. 1), data was excluded from the statistical analyses in the following cases: For DeepLabCut-based analysis: if either of the 10 behavioral motifs was not detected, it was not included in the total cell count for evaluating proportions of DR, IR and non-responsive in the group analysis. For Piezo-based analysis: a trial was excluded if the sensor was not triggered. For the RNAseq experiment (Fig. 3), genes were not included in the differential gene expression analysis unless they met the following criteria: at least 5 cells with a CPM value above 15 and the average CPM value higher than 4. |
| Replication | Experiments were designed so that the data is based on at least two independent subjects per group. This applies to the following experiments: establishing HFD mouse model (Fig. 1a-d), in vivo electrophysiology (Fig. 1e-m, ED Fig. 1e-r, ED Fig. 2), optogenetics (Fig. 2, Fig. 4e, i; Fig. 5e; ED Fig. 3c-m, p; ED Fig. 4c-f; ED Fig. 8a-d, g, h), RNAseq (Fig. 3a-j; ED Fig. 5a-f), NTS release (Fig. 3k-p; Fig. 5a-c; ED Fig 6a-g; ED Fig. 7a-z; ED Fig. 10d), brain slice electrophysiology (Fig. 4l, m; ED Fig. 9b, c, e, f, j-q), tracing (ED Fig. 5g, h), in situ hybridization (ED Fig. 5i-m; ED Fig. 9g-i; ED Fig. 10a-c) NTS-OE (Fig. 5f-l, ED Fig. 10e-k), but with the only exception of the data shown in ED Fig. 6d, e, which was obtained from one subject. Several of these experiments were replicated in at least two technical replicates from each of at least two mice and similar results were obtained. For example, establishing the HFD mouse model (Fig. 1a-d), optogenetics (Fig. 2, Fig. 4i) and NTS-OE (Fig. 5f-l, ED Fig. 10e-k). For anatomical experiments, wherever representative examples are shown (Fig. 4g, k; ED Fig. 1a-d; ED Fig. 3a, b, n, q-t; ED Fig. 4a, b; ED Fig. 6h; ED Fig. 7aa, ab; ED Fig. 8e, f, i, j; ED Fig. 9a, d), similar results were obtained in at least two technical replicates from each of at least two mice. |
| Randomization | Animals were randomly assigned to different diet and virus injection groups from litter mates. For recordings of NAcLat-VTA units during |

| | |
|---|---|
| Randomization | behavior (Fig. 1), the order of jelly and chow exposure was randomized between animals. During optogenetic experiments, the order of mice expressing ChR2 or control fluorophore was randomized for each experimental condition. |
| Blinding | For quantification of NTS-OE using in situ hybridization (ED Fig. 10a-c), all identified regions of interest were manually sorted by an investigator who was blind to virus expression, diet, and probe mix. For quantification of NTS mRNA expression using in situ hybridization following NTS-KO (Fig. 4 b, c) and manipulation of diets (ED Fig. 5i-m), the images were shuffled, and the experimental conditions were hidden during image processing and pixel intensity evaluation. Blinding was not used in other experiments because the experimental conditions were obvious to the researchers and the analysis was performed objectively and not subjective to human bias or analyses were automated. |

# Reporting for specific materials, systems and methods

We require information from authors about some types of materials, experimental systems and methods used in many studies. Here, indicate whether each material, system or method listed is relevant to your study. If you are not sure if a list item applies to your research, read the appropriate section before selecting a response.

## Materials & experimental systems

| n/a | Involved in the study |
|---|---|
| ☐ | ☒ Antibodies |
| ☐ | ☒ Eukaryotic cell lines |
| ☒ | ☐ Palaeontology and archaeology |
| ☐ | ☒ Animals and other organisms |
| ☒ | ☐ Clinical data |
| ☒ | ☐ Dual use research of concern |
| ☒ | ☐ Plants |

## Methods

| n/a | Involved in the study |
|---|---|
| ☒ | ☐ ChIP-seq |
| ☒ | ☐ Flow cytometry |
| ☒ | ☐ MRI-based neuroimaging |

## Antibodies

| | |
|---|---|
| Antibodies used | Primary antibodies: rabbit anti-TH (Millipore, 657012), chicken anti-GFP (Abcam, ab13970), rabbit anti-DS Red (Living Colors, Takara Bio 632496). Secondary antibodies: goat anti-rabbit Alexa Fluor 546 (Life Technologies, A11010), goat anti-chicken Alexa Fluor 488 (Abcam, ab150169), all 1:750. |
| Validation | All antibodies mentioned above have been extensively validated in mice in each of these previous studies (Lammel et al., 2008; Neuron; Lammel et al., 2012; Nature; Cerniauskas et al., 2019; Neuron, Yang et al., 2021; Nature Neuroscience; Cardozo Pinto, et al., 2019), except rabbit anti-DS red antibody, which was validated by the manufacturer: https://www.takarabio.com/documents/Certificate%20of%20Analysis/632496/632496-101717.pdf |

## Eukaryotic cell lines

Policy information about cell lines and Sex and Gender in Research

| | |
|---|---|
| Cell line source(s) | HEK293T (ATCC, CRL-3126, https://www.atcc.org/products/crl-3216). E18 rat hippocampal neuron (BrainBits, https://tissue.transnetyx.com/E18-Rat-Hippocampus_4). |
| Authentication | These cell lines were not authenticated. |
| Mycoplasma contamination | These cell lines were not tested for mycoplasma contamination. |
| Commonly misidentified lines (See ICLAC register) | No commonly misidentified cell lines were used in the study. |

## Animals and other research organisms

Policy information about studies involving animals; ARRIVE guidelines recommended for reporting animal research, and Sex and Gender in Research

| | |
|---|---|
| Laboratory animals | The following mouse lines (6-8 weeks old) were used: C57BL/6J mice (Jackson Laboratory, stock number: 000664), NTS-Cre (Jackson Laboratory; stock number: 017525, strain code: Ntstm1(cre)Mgmj), GAD2-Cre (Jackson Laboratory, stock number: 010802, strain code: Gad2tm2(cre)Zjh/J), Ai14 (Jackson Laboratory, stock number: 007914, strain code: B6.Cg-Gt(ROSA)26Sortm14(CAG-tdTomato)Hze/J). NTS-FLOX (Jackson Laboratory, stock number: 036262, strain code: B6;FVB-Ntsem1Evdr/J). Mice were housed on a 12:12 light cycle (lights on at 07:00) and a room temperature of 22-25°C and 55% humidity. All procedures complied with the animal care standards set forth by the National Institutes of Health and were approved by University of California Berkeley's Administrative Panel on Laboratory Animal Care. |
| Wild animals | The study did not involve wild animals. |
| Reporting on sex | Sex was not considered as a co-variable. All experiments were performed in mixed cohorts of mice. No between-mice comparisons |

| Reporting on sex | were made in the study. |
| --- | --- |
| Field-collected samples | No field-collected samples were used. |
| Ethics oversight | All procedures complied with the animal care standards set forth by the National Institutes of Health and were approved by University of California, Berkeley's Administrative Panel on Laboratory Animal Care. |

Note that full information on the approval of the study protocol must also be provided in the manuscript.

## Plants

| Seed stocks | N/A |
| --- | --- |
| Novel plant genotypes | N/A |
| Authentication | N/A |

