## [Peer Review File · Nature]

Changes in neurotensin signaling drive hedonic devaluation in obesity

Corresponding Author: Dr Stephan Lammel

Version 1:

Reviewer comments:

Referee #1

(Remarks to the Author)

Shimoni, Tose and colleagues performed a systematic study of the relationship between hedonic feeding as regulated by projections from the lateral part of the nucleus accumbens (NAcLat) to the ventral tegmental area (VTA) and how these functions are altered following prolonged consumption of high fat diet (HFD). Using automated behavior tracking combined with photo-tagging of VTA-projecting NAcLat neurons, the authors argue that the neurons respond most strongly while approaching palatable food and that this relationship is disrupted following HFD consumption. They then use optogenetics to activate this pathway, showing that activation selectively increases consumption of highly palatable foods, which is absent in mice maintained on HFD. Using a single-cell patch-seq approach, they identify neurotensin (Nts) as being especially down-regulated in HFD fed mice. This results in reduced Nts release within the VTA. Through a combination of in vivo and ex vivo pharmacological experiments, the authors argue that presynaptic, but not postsynaptic, Nts is critical for mediating the effects of HFD on hedonic behavior. Finally, the authors show that hedonic behavior elicited by NAcLat to VTA stimulation can be partially rescued by overexpressing Nts in this pathway in HFD fed mice.

Overall, this study elegantly tackles an important question regarding brain control of hedonic behavior in a context of overconsumption of highly palatable food. The authors employ a comprehensive set of techniques designed to elucidate the genetic, cellular, and circuit level functions of this pathway as they relate to hedonic feeding. The experiments are well designed and executed (though see comments below), and the results are presented clearly. The reliability of the results could be enhanced with more rigorous analysis of data presented in Fig. 1 and the addition of critical control experiments.

Major concerns:

1. The analysis of neural data in Figure 1 should be revised to directly compare neural responses across behavioral events for individual neurons.

- The thoughtful experimental design allows the authors to directly assess jelly and chow response selectivity within each neuron, yet this analysis appears to be absent. A direct comparison of jelly vs chow responsivity for neurons that are significantly and specifically responsive to food approach should be shown (i.e., do individual jelly-responding neurons actually fire less during chow approach?). A similar analysis should be conducted for the HFD condition.
- Average time series responses to relevant behaviors (e.g., approaching the food cup, approaching empty cup) for example opto-tagged and non-opto-tagged units should be included. How selective are the opto-tagged neurons responses to a particular behavior? The two example neurons shown appear to respond to multiple behaviors, including food approach. In Fig. 1i,j,m,n, most neurons appear to be quite unresponsive for each of the behaviors. What proportion of neurons are significantly responsive during each behavioral motif? Does the proportion of 'jelly responsive' neurons significantly decrease in HFD mice compared with REG mice?
- An assumption about these results is that neurons that were not tagged (putatively not projecting to the VTA) show a different response profile than those confirmed to project to the VTA. Response profiles of non-opto-tagged neurons should be analyzed in a similar manner to opto-tagged neurons. Are the VTA projecting neurons functionally distinct from the putatively non-VTA projecting neurons?

2. Fig. 2: One alternative explanation for the finding that HFD mice fail to show increased consumption of palatable foods is that the HFD potentiates novelty-induced food suppression. To test this possibility, the same optogenetic stimulation paradigm could be applied using HFD in both HFD and REG groups. If the HFD group fails to increase HFD consumption, it would corroborate the interpretation that the mice are responding to hedonic value as opposed to novelty.

3. NTS expression changes in NAcLat neurons following HFD should be independently validated with a method other than sequencing (e.g., in situ hybridization). Does the NTS expression rebound in a way that is similar to the functional recovery upon placing the mice back onto chow after HFD?

4. An important untested prediction that arises from the Nts results is that transient NTS:NAcLat to VTA stimulation should recapitulate the hedonic feeding phenotype observed during stimulation of the NAcLat to VTA pathway (Fig. 2). The results of Fig. 5i-l would also suggest that activation of Nts projections to the VTA should increase locomotion velocity.

Minor concerns:

5. Line 103-104: data measuring adiposity are not shown. Please either include the data or remove this claim.

6. It has previously been shown that some NAcLat neurons exhibit pauses in activity during palatable food consumption (e.g., Krause et al., 2010 J. Neurosci). Are any such neurons observed in the recording experiments (Fig. 1)? If so, some link with existing literature and brief discussion of the implications within the context of the broader results would be helpful.

7. How long were mice maintained on HFD prior to performing patch clamp recordings? Please clarify in the text.

8. The legend for Fig. 5h is missing.

9. Fig. 5j: are the example behavioral traces taken from the same period within the session? For example, one might expect the behavior to be quite different at the beginning of the session compared with the end of the session. If so, it is important to show behavior from the same epoch within the session.

Referee #2

(Remarks to the Author)

In this study, Shimoni et al. explore a neural mechanism by which chronic exposure to high-fat diet (HFD) attenuates the drive for calorie-dense food. The authors observed that enhanced responses of ventral tegmental area (VTA)-projecting lateral nucleus accumbens (NAcLat) neurons to high-calorie foods, as well as increase in feeding of such foods following activation of the NAcLat-VTA circuit, were diminished in HFD-induced obese mice. Further investigation using patch-seq revealed a reduced neurotensin expression and release within this circuit in HFD mice, implying the role of neurotensin signaling in this effect. Remarkably, blocking neurotensin signaling abolished the effect of NAcLat-VTA optogenetic stimulation in caloric food intake and VTA dopamine neuron activity. Conversely, restoring neurotensin signaling in this circuit reversed the decrease in caloric food intake and mitigated weight gain in HFD mice.

How chronic exposure to high-calorie foods impacts feeding behavior and contributes to weight gain is a topic of tremendous interest given the obesity epidemic. The authors examine a paradoxical phenomenon where HFD-induced obese mice exhibit lowered desire for calorie-rich foods. Then they present compelling evidence showing that plasticity in neuropeptide signaling may underlie this phenomenon at least in part. The present study not only sheds light on a neural mechanism behind the progression of obesity, but also introduces a novel therapeutic avenue for obesity treatment, marking a noteworthy progress. The manuscript is logically structured, clearly and concisely written, and provides an in-depth discussion. However, there are few limitations to the interpretation of the data that will require corrective steps.

Major points:

1. The authors claim in the title and throughout the manuscript that "diet induced changes in hedonic feeding shape obesity progression", yet this claim lacks direct experimental support. What is demonstrated is how (diet-induced) changes in neurotensin signaling shape changes in hedonic feeding and obesity progression (e.g. Figs. 4a-c, 5f-h), not a causal link between decreased appetite for high-calorie foods and obesity progression. The authors are recommended to either conduct additional experiments to substantiate their claim or revise the manuscript to accurately reflect their findings. For example, the title can be rephrased to "diet-induced changes in neurotensin signaling shape hedonic feeding and obesity progression".

2. The authors note that increase in firing rate of VTA dopamine neurons, triggered by optogenetic stimulation of NAcLat-VTA circuit, was abolished after application of an NTS receptor antagonist (Fig. 4d-f). This is unexpected and particularly confusing considering the previous findings from the group of corresponding author that NAcLat neuron activation leads to the disinhibition of VTA dopamine neurons through GABAergic inhibition of VTA GABA neurons. Explanation for this apparent contradiction should be incorporated in the manuscript.

3. The authors did show in Extended Data Fig. 8c that infusion of neurotensin receptor antagonist did not increase caloric food consumption in regular chow-fed mice, but this method may not be ideal due to its non-specific effect. Would inhibition of neurotensin signaling in this circuit in regular chow-fed mice recapitulate reduced drive to consume food with high calories? Loss-of-function experiments using optogenetics may reveal the importance of this circuit in the development of

impaired hedonic feeding.

Minor points:

1. The authors posit an intriguing observation regarding the restoration of hedonic feeding behavior by neurotensin overexpression leading to increased consumption of hedonic food yet resulting in decreased overall food intake and weight gain. While potential explanations for this seemingly paradoxical phenomenon are discussed (lines 394-417), a detailed examination of the daily feeding patterns in neurotensin-overexpressing mice and control mice under HFD conditions would provide invaluable insights into understanding the relationship between changes in hedonic feeding and the progression of obesity.

Referee #3

(Remarks to the Author)

The article by Shimoni and colleagues investigates the role of neurotensin (NTS) signaling, within a lateral nucleus accumbens to ventral tegmental area disinhibitory feedback loop (NAcLat VTA), in hedonic feeding. They first show that mice given HFD show a reduced preference for palatable food. Then, using drivable optoelectrode, they show that NAcLat  VTA neurons from chow fed animals respond to hedonic feeding, but not in high-fat diet (HFD) animals. Then they strengthen this observation by demonstrating that optogenetic stimulation of the NAcLat  VTA pathway is sufficient to induce hedonic feeding in chow- but not HFD-fed animals. Interestingly, only stimulation of NAcLat  VTA and not the NAc neurons themselves produces this effect, indicating that another population of neurons might control behavior in an opposing manner. Using patch-seq, they then describe the same electrical properties in both chow and HFD group, but highlight differential expression of multiple genes including a decreased NTS expression in the HFD group. Associated with this, they unveil a reduction in NTS release within the VTA using ntsLight1.1 sensor. To better characterize the involvement of NTS in hedonic feeding, they used the NTSR antagonist injection within the VTA to show that it prevents hedonic feeding despite the opto-stimulation. Overexpressing NTS in HFD mice increases consumption of jelly in the timed-feeding assay, but paradoxically reduces bodyweight overall. Altogether, Shimoni and colleagues describe high-fat diet impact on reward system circuitry and NTS signaling and the consequence on hedonic feeding.

Overall, I think that this is a very nicely organized story that will add to the ongoing story of the role of the mesolimbic dopamine system to feeding and obesity. However, I have some concerns regarding the interpretation of the main point of the paper, which significantly diminishes my enthusiasm. Moreover, regarding the title, I think that 'shape obesity progression' is too strong of a statement based the data shown and should be rephrased to be more closely aligned to the data.

1. The biggest issue that I see overall is that the authors do not account for the possibility that the effects in Figure 2 are due to novelty rather than to feeding per se. Because jelly is not a food that the mice have extensive experience with, but rather just a few pre-exposures, it is possible that HFD mice have increased anxiety towards novel foods (meaning, increased novelty suppressed feeding). This would align with a lot of previous literature regarding increased anxiety in HFD mice. Moreover, this might also explain the paradoxical data in Figure 5, in which overexpression of NTS leads to decreased weight and homecage consumption of HFD while increased jelly consumption in the time-feeding assay. The easiest way to explore this would be to repeat the timed-feeding experiments using HFD, a palatable food that they are highly familiar with.
2. Closely related to point 1, I don't see any justification for using "real food" substances like jelly, peanut butter, butter etc. instead of HFD, HFHS and/or sucrose pellets or any other standardized food substance? The optogenetic stimulation experiments should be conducted with some standardized food substances as well so that the nutritional components are very clear. In addition, the brands and relative nutritional components of the "real food" substances should be listed. Even the amount of quinine added to the butter was not listed, nor the method of mixing them.
3. The second biggest issue that I see is that the authors show that opto activation increases NTS release and that NTSR antagonist prevents opto-induced feeding. But what they do not show is whether palatable food consumption endogenously increases NTS release, which would greatly strengthen the conclusions of the paper.
4. In general, I find the discussion of neurotensin and feeding sorely lacking. There is abundant evidence in many brain regions that neurotensin is anorexigenic and decreases feeding, including VTA, lateral septum, hypothalamus and nucleus of tract solitarius. The only other exception that I'm aware of being the CeA, where activation of Nts neurons led to increase consumption of ethanol and sucrose liquids, but did not change solid food consumption (PMID: 31744862). That activation of a (potentially) subpopulation of Nts neurons leads to increased palatable consumption is therefore surprising and the implications of that should be discussed thoroughly.
5. In Figure 1, the authors show neuronal activity depending on the behavior. They mention in the text l.119 'we found increased firing rate in opto-tagged units during jelly, but not chow, consumption', however they show 'approaching food' and not feeding. In the method they mention that feeding events were verified by analyzing manually the video. Why don't they dissociate approach from actual feeding? Besides, the animals are barely approaching the chow so the statement in that sentence should be reconsidered. Same thing for l.123-124, HFD mice barely approach the food, but the authors stipulate that there is no increase during jelly eating. More detailed analysis is required or the statement should be moderated.
6. There was a lot of evidence cited from rodents that HFD leads to reduced preference for reward. What is the evidence in humans that there is reduced preference for reward?
7. The authors provide in the Introduction a rationale for studying VTA  NAcc, but then instead pivot to studying NAcc -> VTA without any rationale. Moreover, the justification for NAcc-Lat and not core or medial shell is also not justified in any way, nor is any of that previous literature cited.
8. Figure 1i: The figure legend states that it is 18 units from 8 mice. There are many units with low firing rates and only a few units with high firing rates that seem to drive the statistical difference. Are these units treated statistically independently? Are all of the units with high firing rates from just one or two mice? I think this needs to be considered very carefully because the

claim is not necessarily supported if the high firing rates are derived only from a few animals.

9. Similar to point 7, but for Extended Data figure 3c: the legend says 12 experiments from 8 mice. Does this mean that the same mouse was tested multiple times but they are being treated as independent samples? This can artificially inflate the data. If one mouse has multiple trials, either they should be averaged together for a nested analysis or just one trial be used.

10. Extended fig. 6h. The authors show that 76% of NacLat->VTA neurons express NTS, but how many NTS neurons project to VTA? From the sample image it seems there are many more NTS neurons than CTB-labeled. This is important to know whether NTS is a true marker for this projection or just a general marker in NAcLat neurons.

11. Figure 5: the value seems too high for one mouse consumption over a week, the authors should double-check that it is actually normalized per animal per cage. I also wonder if there is an effect of NTS over-expression on stress/anxiety as you show an increased jelly consumption in a restricted feeding in a non-home cage associated with a decrease food consumption of HFD in home cage. Since you performed an open field, I recommend to analyze anxiety and add it in the extended data in order to rule out any stress effect in addition to the studies on novelty detailed in point 1.

12. Figure 5f shows the body weight evolution depending on the diet, it would be interesting to have the parallel with food consumption evolution (in kcal in order to have both combine all phases in the same graph) as the authors only show the food consumption during HFD+Chow phase.

13. I find this sentence of the abstract confusing and opaque "Altogether, our results suggest that the transition from hedonically-driven eating to habitual consumption involves an intricate neural mechanism, where the brain's reward system dynamically adapts to dietary influences. This adaptability is a crucial factor in the progression of obesity." What data does this refer to? Which adaptability is a crucial factor in the progression of obesity. I don't think I saw any data that changes the progression of obesity, rather most of the data showed that obesity that already occurs has altered reward dynamics.

Minor comm

1. Extended Figure 1F: why is this normalized to 5 mice / cage? Does that mean that mice are eating on average only 1.5g of chow a day? This doesn't seem to make sense ... In addition, it might be helpful to provide kcal as well as g since the kcal/g differ for chow and HFD.

2. Figure 1e and extended figure 1G: are these supposed to be the same data presented differently? Because the data points don't align. If they aren't the same data, what two distinct time points are they from?

3. Extended data figure 2 has no description in the manuscript text to say what the purpose of the figure is.

4. Figure 2d and similar: it is very confusing to the reader for the individual data to be underneath the summarized data. This individual data should either be moved to supplementary or put in more transparent underneath the summary data in one graph.

5. Extended data fig 10a-c: the authors should also compare to chow-fed mice to determine if NTS OE is restoring NTS levels or perhaps surpassing normal NTS levels.

6. Comparing Figure 3L to figure 5b the NTS release in HFD mice looks starkly different and much higher in Figure 5b. What is the reason for this?

7. Extended data figure 5f: for consistency, please replace pre-feeding with primed-feeding trial. Additionally, why is there only one group? It would be interesting to compare the 3week evolution with eYFP group.

8. Figure 5h: the color legend is missing; I recommend using the same color as 5g for consistency

Version 2:

Reviewer comments:

Referee #1

(Remarks to the Author)

I commend the authors on a thorough revision that has addressed my concerns. I have no further comments.

Referee #2

(Remarks to the Author)

The authors have successfully addressed my concerns. The manuscript is significantly improved, and I have no further suggestions.

Referee #3

(Remarks to the Author)

I was pleased to see the revision for the manuscript by Gazit Shimoni and colleagues. Overall, I was incredibly impressed with the responses to critiques and I think the added experiments significantly improve the paper, in particular the addition of the inhibition experiments and in vivo application of Nts-light. Although I still think it is a bit of a missing link that Nts release is not shown from physiological behavior and only from opto stim, I am satisfied with the authors explanation of why this was not feasible at this time. The manuscript is now a very comprehensive investigation that provides an important mechanism for linking obesity with the devaluation of reward, although the potential therapeutic benefit from neurotensin agonism on weight loss is already known. However, the finding that this is linked also to devaluation and reverses devaluation provides a previously unknown angle. I have just two easily addressable points to still make.

1. I still think the title is somewhat misleading. I do not see much in the paper about obesity progression per se, and I believe the title should reflect more about the devaluation of reward, which is actually the main focus.

2. In addition, perhaps the authors plan to upload it before publication, but I did not see any mention of submitting the RNA sequencing data to the GEO database.

Minor points:

1. Pg. 7. States that upregulated genes are Log Fold Change >-1 , but I presume they mean 1?
2. Extended Fig 4J legend seems to be wrong. The schematic states Day 2 has laser but below day 2 is white and on the graph it's the dark one which has the laser
3. P11: the last paragraph states that there is no change in metabolic rate because there was no change in body temperature, but they have an increase in locomotion and per se energy expenditure. This would have to be examined more closely to determine no change in metabolic rate.
4. Please check all references. For example, ref. 35 is not the first demonstration that Nts in the LS controls feeding, but rather is Azevedo et al., 2020.

November 22, 2024

Re: Decision on Nature manuscript 2024-01-01245A

RESPONSE TO REVIEWERS

We are grateful to the reviewers for their detailed reading of our manuscript and for their constructive comments. Based on the referees' feedback, we have substantially revised our manuscript. Changes in the manuscript have been highlighted in yellow.

We hope that the revised manuscript and the responses included in this letter will satisfy the reviewers in that we have carried out a systematic and rigorous study on how neurotensin influences hedonic feeding behavior and how changes in neurotensin expression and release may influence various aspects of diet-induced obesity. We look forward to answering any further questions about our work and eagerly await a decision on our paper. Please find below a detailed response to the reviewers' comments which are reproduced in italics.

REVIEWER COMMENTS

Referee #1:

Shimoni, Tose and colleagues performed a systematic study of the relationship between hedonic feeding as regulated by projections from the lateral part of the nucleus accumbens (NAcLat) to the ventral tegmental area (VTA) and how these functions are altered following prolonged consumption of high fat diet (HFD). Using automated behavior tracking combined with phototagging of VTA-projecting NAcLat neurons, the authors argue that the neurons respond most strongly while approaching palatable food and that this relationship is disrupted following HFD consumption. They then use optogenetics to activate this pathway, showing that activation selectively increases consumption of highly palatable foods, which is absent in mice maintained on HFD. Using a single-cell patch-seq approach, they identify neurotensin (Nts) as being especially down-regulated in HFD fed mice. This results in reduced Nts release within the VTA. Through a combination of in vivo and ex vivo pharmacological experiments, the authors argue that presynaptic, but not postsynaptic, Nts is critical for mediating the effects of HFD on hedonic behavior. Finally, the authors show that hedonic behavior elicited by NAcLat to VTA stimulation can be partially rescued by overexpressing Nts in this pathway in HFD fed mice. Overall, this study elegantly tackles an important question regarding brain control of hedonic behavior in a context of overconsumption of highly palatable food. The authors employ a comprehensive set of techniques designed to elucidate the genetic, cellular, and circuit level functions of this pathway as they relate to hedonic feeding. The experiments are well designed and executed (though see comments below), and the results are presented clearly. The reliability of the results could be enhanced with more rigorous analysis of data presented in Fig. 1 and the addition of critical control experiments.

Major concerns:

1. *The analysis of neural data in Figure 1 should be revised to directly compare neural responses across behavioral events for individual neurons.*

We thank the referee for their thoughtful commentary, which has substantially improved our study. In response to the suggestion regarding the electrophysiology data in **Figure 1**, we have thoroughly re-analyzed the data. Specifically, we analyzed both opto-tagged and non-opto-tagged units using a time-series approach to directly compare neural responses across different behavioral events and piezo sensor activation for individual neurons.

We quantified unit firing rates before and during the first three seconds of each motif occurrence and used the Wilcoxon signed-rank test to determine whether the neural response was unchanged (non-responsive), significantly increased, or significantly decreased. Additionally, we assessed the proportion of each response type in REG and HFD mice and evaluated whether these proportions differed between diets using a Chi-square test.

Our analysis revealed that, in REG mice, a substantial number of opto-tagged units exhibited significantly increased firing rates during jelly consumption compared to baseline activity. This increase was predominantly observed during jelly consumption, while other behavioral motifs frequently showed decreased responses. In contrast, opto-tagged units in HFD mice displayed lower firing rates during jelly consumption, with none of the tagged units reaching statistical significance. Similar results were observed in piezo-based analyses, with increased firing rates during jelly consumption in REG mice and a marked reduction in HFD mice. Non-tagged units also exhibited higher firing rates during jelly consumption compared to other motifs, as well as reduced proportions of IR responses in HFD mice compared to REG mice for both DLC- and piezo-based analyses, although the effect size was smaller.

While we observed units with both statistically significant decreases and increases in firing rates during other behavioral motifs, the proportions of these responses did not differ significantly between diets. ***These revisions have been included in Figure 1e-m and Extended Data Figure 3.***

- *The thoughtful experimental design allows the authors to directly assess jelly and chow response selectivity within each neuron, yet this analysis appears to be absent. A direct comparison of jelly vs chow responsivity for neurons that are significantly and specifically responsive to food approach should be shown (i.e., do individual jelly-responding neurons actually fire less during chow approach?). A similar analysis should be conducted for the HFD condition.*

We thank the reviewer for their insightful comment. To address this point, we have now included additional analyses for both REG and HFD mice that directly compare jelly versus chow responsivity in individual opto-tagged and non-opto-tagged neurons. ***As shown in Extended Data Fig. 3a, b (for opto-tagged units) and Extended Data Fig. 3i, j (for non-tagged units), the majority of units that respond significantly to jelly exhibit little to no responsivity to chow.***

- *Average time series responses to relevant behaviors (e.g., approaching the food cup, approaching empty cup) for example opto-tagged and non-opto-tagged units should be included.*

We thank the reviewer for this suggestion. As noted above, ***our analysis now incorporates average time-series responses*** to determine whether a unit responds to specific behavioral

motifs and to classify the type of change (i.e., increase, decrease, or no response). **Figure 1k** includes an example time-series plot of the same neuron during different motifs (e.g., jelly consumption and walking) demonstrating opposing response types.

How selective are the opto-tagged neurons responses to a particular behavior? The two example neurons shown appear to respond to multiple behaviors, including food approach. In Fig. 1i,j,m,n, most neurons appear to be quite unresponsive for each of the behaviors.

We thank the reviewer for raising this important point. To address it, we have included heatmaps for both REG and HFD mice showing the Z-scored firing rates of opto-tagged units (**Fig. 1l, m**). These data demonstrate high responsivity to jelly-feeding behavior, although some units also respond to other behavioral motifs. Notably, while opto-tagged units in HFD mice retain some responsiveness to jelly feeding, the proportion of cells showing a statistically significant increase during jelly feeding is significantly reduced compared to REG mice.

What proportion of neurons are significantly responsive during each behavioral motif? Does the proportion of 'jelly responsive' neurons significantly decrease in HFD mice compared with REG mice?

The reviewer's assumption is correct. As shown in **Figure 1l, m** (bar graphs), the proportion of opto-tagged cells in HFD mice that exhibit a statistically significant increase during jelly feeding is indeed significantly reduced compared to REG mice.

• An assumption about these results is that neurons that were not tagged (putatively not projecting to the VTA) show a different response profile than those confirmed to project to the VTA. Response profiles of non-opto-tagged neurons should be analyzed in a similar manner to opto-tagged neurons. Are the VTA projecting neurons functionally distinct from the putatively non-VTA projecting neurons?

Thank you for this valuable suggestion. **We have now analyzed the non-opto-tagged neurons using the same approach applied to the opto-tagged cells (Extended Data Fig. 3g-j)**. In general, we observe similar activity patterns for the different behavioral motifs in non-opto-tagged cells as in opto-tagged cells, though with some quantitative differences. For instance, as in the opto-tagged group, a substantial number of non-opto-tagged cells exhibit increased responses during jelly consumption. However, while the proportion of non-opto-tagged units with a statistically significant increase in firing rate during jelly consumption was reduced in HFD mice compared to REG mice, this reduction did not reach statistical significance ($p = 0.06$). A possible explanation for this discrepancy is the uncertainty regarding the specific projection targets of non-opto-tagged cells. While our data suggest that the non-opto-tagged group likely includes a significant proportion of neurons projecting to the VTA, it is reasonable to assume that it also contains other cell populations, such as interneurons or neurons with distinct projection targets. Differences in the composition of these cell populations between REG and HFD mice may account for the observed discrepancy between the opto-tagged and non-tagged data.

2. *Fig. 2: One alternative explanation for the finding that HFD mice fail to show increased consumption of palatable foods is that the HFD potentiates novelty-induced food suppression. To test this possibility, the same optogenetic stimulation paradigm could be applied using HFD in both HFD and REG groups. If the HFD group fails to increase HFD consumption, it would corroborate the interpretation that the mice are responding to hedonic value as opposed to novelty.*

We appreciate the reviewer's important suggestion regarding the potential role of novelty-induced food suppression in our findings. To address this, we conducted an additional experiment in which both HFD and regular chow-fed (REG) mice were offered high-fat chow (HF chow), and we applied the same optogenetic stimulation paradigm targeting the NAcLat→VTA pathway. Our results demonstrate that optogenetic stimulation significantly increased HF chow consumption in the REG but not HFD group. These findings support the idea that mice are responding to the hedonic value of food rather than to novelty. **The new data has been included in Figure 2d, e.**

3. *NTS expression changes in NAcLat neurons following HFD should be independently validated with a method other than sequencing (e.g., in situ hybridization). Does the NTS expression rebound in a way that is similar to the functional recovery upon placing the mice back onto chow after HFD?*

We thank the reviewer for drawing our attention to this point. To address their suggestion to further validate NTS expression changes using an alternative method, **we performed additional in situ hybridization experiments, which are shown in ED Fig. 7c-g.** Specifically, we used fluorescent retrobeads for retrograde identification of NAcLat→VTA cells and then quantified NTS mRNA expression in beads-positive NAcLat cells from 3 different cohorts of mice (i.e., mice on a regular diet, mice on a high-fat diet and mice that have been subjected to a high-fat diet, but then returned to a regular diet). Our results show that NTS expression was significantly reduced in HFD mice. However, NTS expression was increased in mice returned to a regular diet, though it remained slightly lower on average compared to mice that had only been on a regular diet. These findings are consistent with the RNAseq data (**Fig. 3g-j**), and they align with the observed functional recovery shown in **Figure 2g**. Together with our previous findings, these new results support the conclusion that NTS expression in NAcLat→VTA cells is reduced in HFD mice, which may be reversible when the diet is changed.

4. *An important untested prediction that arises from the Nts results is that transient NTS:NAcLat to VTA stimulation should recapitulate the hedonic feeding phenotype observed during stimulation of the NAcLat to VTA pathway (Fig. 2). The results of Fig. 5i-l would also suggest that activation of Nts projections to the VTA should increase locomotion velocity.*

We appreciate the reviewer's insightful prediction and have conducted additional experiments to address this point. Specifically, **we optogenetically stimulated the NAcLat→VTA pathway in NTS-Cre mice** (i.e., we targeted Chr2 specifically to NTS-expressing NAcLat neurons and implanted an optical fiber above the VTA) to examine its effects on acute feeding behavior. Consistent with our findings in C57Bl/6 mice, we observed a significant increase in hedonic feeding behavior during optogenetic stimulation (**Extended Data Fig. 10a-c**).

Regarding locomotor activity, while NTS overexpression (NTS-OE) in NAcLat→VTA cells increased locomotion, we did not observe a similar increase when stimulating the NAcLat→VTA

pathway in NTS-Cre mice (**Extended Data Fig. 10d**). One possible explanation for this discrepancy is that under normal conditions, when mice are on a regular diet, NTS-OE does not appear to affect behaviors such as time-restricted feeding, home-cage feeding, or body weight. These changes only emerged after switching the diet from REG to HFD. We believe the same principle applies to locomotor activity. Therefore, the increased locomotion observed in NTS-OE mice on a high-fat diet may not necessarily be recapitulated when stimulating the NAcLat→VTA pathway in NTS-Cre mice on a regular diet.

Minor concerns:

5. Line 103-104: data measuring adiposity are not shown. Please either include the data or remove this claim.

We have removed this claim.

6. It has previously been shown that some NAcLat neurons exhibit pauses in activity during palatable food consumption (e.g., Krause et al., 2010 J. Neurosci). Are any such neurons observed in the recording experiments (Fig. 1)? If so, some link with existing literature and brief discussion of the implications within the context of the broader results would be helpful.

We thank the reviewer for this suggestion. In the revised manuscript, **we have adopted a similar classification approach to that used in Krause et al., 2010**. In their study, as well as in other work from the Fields group, two response types to hedonic food were identified: Type 1 (neurons with decreased responses during feeding) and Type 2 (neurons with increased responses). In our study, we identified cells displaying similar response types during hedonic feeding. Specifically, we found approximately 60% Type 2 (increased response) responses during jelly consumption when analyzing opto-tagged units, and around 30% Type 2 responses when analyzing non-tagged units. Notably, we did not observe any Type 1 (decreased response) responses in the opto-tagged group, but we did identify ~4% of this response type in the non-tagged group. The differences between our findings and those of the Fields group may be attributed to several factors, including differences in recording locations (as their work focuses more intensively on the medial shell and core of the NAc), differences in species, or, importantly, the use of food-restricted animals in their experiments. Food restriction induces a state of hunger, potentially activating homeostatic mechanisms that we deliberately avoided in our paradigm.

7. How long were mice maintained on HFD prior to performing patch clamp recordings? Please clarify in the text.

Mice were maintained on HFD for 4 weeks prior to patch clamp recordings. This has been clarified in the text (**p.10**).

8. The legend for Fig. 5h is missing.

Thank you. The legend has now been included.

9. Fig. 5j: are the example behavioral traces taken from the same period within the session? For example, one might expect the behavior to be quite different at the beginning of the session compared with the end of the session. If so, it is important to show behavior from the same epoch within the session.

We have **revised Figure 5j to display the entire session (10 minutes) for NTS-OE and CTRL mice**. Additionally, we conducted a detailed analysis of the behavioral motifs using DeepLabCut to gain a deeper understanding of the specific aspects of locomotion behavior affected. Our results indicate that the increased locomotion observed in NTS-OE mice is not confined to a specific segment of the session.

Referee #2:

In this study, Shimoni et al. explore a neural mechanism by which chronic exposure to high-fat diet (HFD) attenuates the drive for calorie-dense food. The authors observed that enhanced responses of ventral tegmental area (VTA)-projecting lateral nucleus accumbens (NAcLat) neurons to high-calorie foods, as well as increase in feeding of such foods following activation of the NAcLat-VTA circuit, were diminished in HFD-induced obese mice. Further investigation using patch-seq revealed a reduced neurotensin expression and release within this circuit in HFD mice, implying the role of neurotensin signaling in this effect. Remarkably, blocking neurotensin signaling abolished the effect of NAcLat-VTA optogenetic stimulation in caloric food intake and VTA dopamine neuron activity. Conversely, restoring neurotensin signaling in this circuit reversed the decrease in caloric food intake and mitigated weight gain in HFD mice. How chronic exposure to high-calorie foods impacts feeding behavior and contributes to weight gain is a topic of tremendous interest given the obesity epidemic. The authors examine a paradoxical phenomenon where HFD-induced obese mice exhibit lowered desire for calorie-rich foods. Then they present compelling evidence showing that plasticity in neuropeptide signaling may underlie this phenomenon at least in part. The present study not only sheds light on a neural mechanism behind the progression of obesity, but also introduces a novel therapeutic avenue for obesity treatment, marking a noteworthy progress. The manuscript is logically structured, clearly and concisely written, and provides an in-depth discussion. However, there are few limitations to the interpretation of the data that will require corrective steps.

Major points:

1. *The authors claim in the title and throughout the manuscript that "diet induced changes in hedonic feeding shape obesity progression", yet this claim lacks direct experimental support. What is demonstrated is how (diet-induced) changes in neurotensin signaling shape changes in hedonic feeding and obesity progression (e.g. Figs. 4a-c, 5f-h), not a causal link between decreased appetite for high-calorie foods and obesity progression. The authors are recommended to either conduct additional experiments to substantiate their claim or revise the manuscript to accurately reflect their findings. For example, the title can be rephrased to "diet-induced changes in neurotensin signaling shape hedonic feeding and obesity progression".*

We thank the reviewer for their constructive commentary and for pointing out that our study makes a **"noteworthy progress that sheds light on neural mechanisms behind the progression of obesity and introduces novel therapeutic avenues for obesity treatment"**. We agree that the title should more accurately reflect the findings presented in the manuscript. Accordingly, we have

revised the title to: "Diet-induced changes in neurotensin signaling shape hedonic feeding and obesity progression."

2. The authors note that increase in firing rate of VTA dopamine neurons, triggered by optogenetic stimulation of NAcLat-VTA circuit, was abolished after application of an NTS receptor antagonist (Fig. 4d-f). This is unexpected and particularly confusing considering the previous findings from the group of corresponding author that NAcLat neuron activation leads to the disinhibition of VTA dopamine neurons through GABAergic inhibition of VTA GABA neurons. Explanation for this apparent contradiction should be incorporated in the manuscript.

We thank the reviewer for drawing our attention to this important point. Indeed, in a previous study we demonstrated that optogenetic stimulation of NAcLat terminals in the VTA promotes excitation of VTA dopamine (DA) neurons (Yang et al., 2018). In the Yang et al. study, we assumed (but did not directly test) that excitation of VTA DA neurons involves disinhibition through local VTA GABAergic neurons. This assumption was based on our results that (i) NAcLat neurons make direct inhibitory connections onto VTA GABAergic neurons and (ii) VTA GABAergic neurons make functional synaptic connections onto VTA DA neurons. At that time, we felt that this assumption was justified, because the idea the VTA GABAergic neurons disinhibit DA neurons is well accepted in the field. In the present study, we show that, in addition to disinhibition, VTA DA neurons are also directly excited through NTS binding to NTS R1 receptors following its release from NAcLat terminals (Woodworth et al., 2017). NTS R1 receptors are selectively expressed on VTA DA neurons, but not on VTA GABAergic neurons (Woodworth et al., 2018, 2017). It is therefore unlikely that NTS release from NAcLat inputs affects VTA GABA cell firing, which could have indirectly influenced VTA DA cell firing. This aligns with our data demonstrating that in brain slice recordings, NTS bath application does not change the firing of VTA GABAergic neurons (**Extended Data Fig. 11d-f**). However, it is possible that in *ex vivo* brain slice preparations, when the net effect of NAcLat stimulation on VTA DA firing is tested, the observed excitation of VTA DA neurons is mainly mediated by NTS rather than disinhibition. In fact, some researchers in the field have long assumed that disinhibition of DA neurons may not be detected in brain slice preparations (personal communication with John Williams, Vollum Institute). This could explain our data showing that, in the presence of an NTS receptor antagonist, DA cells are not excited following optogenetic stimulation of NAcLat inputs (**Fig. 4j-m**). This, however, does not exclude the possibility that VTA DA neurons are disinhibited *in vivo*. It is likely, as demonstrated recently for lateral hypothalamus inputs to the VTA, that both NTS-induced excitation as well as disinhibition of VTA DA neurons are necessary components that determine the function of this pathway *in vivo* (Soden et al., 2023). ***We now touch upon these reflections in our discussion section.***

3. The authors did show in Extended Data Fig. 8c that infusion of neurotensin receptor antagonist did not increase caloric food consumption in regular chow-fed mice, but this method may not be ideal due to its non-specific effect. Would inhibition of neurotensin signaling in this circuit in regular chow-fed mice recapitulate reduced drive to consume food with high calories? Loss-of-function experiments using optogenetics may reveal the importance of this circuit in the development of impaired hedonic feeding.

We appreciate the reviewer's careful attention to this point and agree that loss-of-function experiments are important. To address this concern, we have performed two additional sets of experiments.

First, as suggested by the reviewer, we used optogenetics to inhibit the NAcLat→VTA pathway in an acute feeding assay. Our results show that optogenetic inhibition of this pathway significantly reduced hedonic feeding behavior during the primed feeding trial when mice normally show high motivation for consuming hedonic foods. This finding indicates that this circuit is necessary for promoting hedonic feeding. **These new results are shown in Extended Data Fig. 4o-t.**

Second, to directly assess whether NTS in this circuit is required for hedonic feeding, we performed a conditional knock out of NTS in NAcLat neurons by injecting hSyn-cre to NTS-FLOX mice and control mice without hSyn-cre injection. We then optogenetically stimulated NAcLat terminals in the VTA. Control mice responded as expected to optogenetic stimulation by increasing hedonic food consumption. However, in NTS knock out mice, optogenetic stimulation of the NAcLat→VTA pathway failed to promote hedonic feeding, demonstrating that NTS is necessary for this behavior. **These new results are shown in Figure 4a-e.** We again thank the reviewer for their suggestion as it strengthened our conclusions that NTS within the NAcLat→VTA pathway is critical for hedonic feeding behavior.

Minor points:

1. The authors posit an intriguing observation regarding the restoration of hedonic feeding behavior by neurotensin overexpression leading to increased consumption of hedonic food yet resulting in decreased overall food intake and weight gain. While potential explanations for this seemingly paradoxical phenomenon are discussed (lines 394-417), a detailed examination of the daily feeding patterns in neurotensin-overexpressing mice and control mice under HFD conditions would provide invaluable insights into understanding the relationship between changes in hedonic feeding and the progression of obesity.

To address the reviewer's suggestion, we have included additional time points in our feeding analysis (**Fig. 5h, i**), which provides a better understanding of how feeding behaviors evolve in the same mice during the progression of diet-induced obesity. It indeed has been shown that high-fat diets and obesity disrupt circadian feeding patterns, leading to increased food intake during the inactive phase (Hatori et al., 2012; Kohsaka et al., 2007). Moreover, adjusting feeding times, such as restricting food access to the active phase, has been shown to reduce weight gain and improve metabolic outcomes in diet-induced obesity models (Chaix et al., 2014). Therefore, the reviewer's suggestion for a more detailed characterization of daily feeding patterns in NTS-OE and control mice—such as examining feeding behavior over a 24-hour cycle, including day and night phases—is well received and represents an important future direction. While we acknowledge the value of this approach, and briefly touch on this in our discussion, we believe that our current dataset sufficiently supports our claims that NTS-OE in HFD mice alters both home cage feeding and time-restricted feeding patterns.

Referee #3:

The article by Shimoni and colleagues investigates the role of neurotensin (NTS) signaling, within a lateral nucleus accumbens to ventral tegmental area disinhibitory feedback loop (NAcLat VTA), in hedonic feeding. They first show that mice given HFD show a reduced preference for palatable food. Then, using drivable optoelectrode, they show that NAcLat  VTA neurons from chow fed animals respond to hedonic feeding, but not in high-fat diet (HFD) animals. Then they strengthen this observation by demonstrating that optogenetic stimulation of the NAcLat  VTA pathway is sufficient to induce hedonic feeding in chow- but not HFD-fed

animals. Interestingly, only stimulation of NAcLat  VTA and not the NAc neurons themselves produces this effect, indicating that another population of neurons might control behavior in an opposing manner. Using patch-seq, they then describe the same electrical properties in both chow and HFD group, but highlight differential expression of multiple genes including a decreased NTS expression in the HFD group. Associated with this, they unveil a reduction in NTS release within the VTA using *ntsLight1.1* sensor. To better characterize the involvement of NTS in hedonic feeding, they used the NTSR antagonist injection within the VTA to show that it prevents hedonic feeding despite the opto-stimulation. Overexpressing NTS in HFD mice increases consumption of jelly in the timed-feeding assay, but paradoxically reduces bodyweight overall. Altogether, Shimoni and colleagues describe high-fat diet impact on reward system circuitry and NTS signaling and the consequence on hedonic feeding.

Overall, I think that this is a very nicely organized story that will add to the ongoing story of the role of the mesolimbic dopamine system to feeding and obesity. However, I have some concerns regarding the interpretation of the main point of the paper, which significantly diminishes my enthusiasm. Moreover, regarding the title, I think that 'shape obesity progression' is too strong of a statement based the data shown and should be rephrased to be more closely aligned to the data.

We appreciate the reviewer's thoughtful feedback. In response to the comment about the title, we have revised it as suggested by Referee #1 to better reflect the data presented. **The new title, "Diet-induced changes in neurotensin signaling shape hedonic feeding and obesity progression,"** aligns more closely with our findings, emphasizing the role of NTS signaling in hedonic feeding behavior and its contribution to obesity progression, rather than making a broader statement about directly shaping obesity.

1. The biggest issue that I see overall is that the authors do not account for the possibility that the effects in Figure 2 are due to novelty rather than to feeding per se. Because jelly is not a food that the mice have extensive experience with, but rather just a few pre-exposures, it is possible that HFD mice have increased anxiety towards novel foods (meaning, increased novelty suppressed feeding). This would align with a lot of previous literature regarding increased anxiety in HFD mice. Moreover, this might also explain the paradoxical data in Figure 5, in which overexpression of NTS leads to decreased weight and homecage consumption of HFD while increased jelly consumption in the time-feeding assay. The easiest way to explore this would be to repeat the timed-feeding experiments using HFD, a palatable food that they are highly familiar with.

We appreciate the reviewer's concern regarding the potential role of novelty-induced food suppression in our findings. To address this, we performed additional experiments to explore whether the effects observed in **Figure 2** were due to novelty or feeding per se. Specifically, we repeated the timed-feeding experiments using high-fat chow (HF chow), a palatable food that the mice are highly familiar with, instead of jelly. In these experiments, we found that optogenetic stimulation of the NAcLat→VTA pathway significantly increased HF chow consumption, similar to the effects seen with jelly. These findings suggest that the observed effects are indeed related to feeding behavior driven by hedonic value, rather than anxiety or novelty-induced food suppression. This supports our original interpretation and indicates that the results are not solely due to the novelty of the jelly. **The new data have been included in Figure 2d, e.**

2. Closely related to point 1, I don't see any justification for using "real food" substances like

jelly, peanut butter, butter etc. instead of HFD, HFHS and/or sucrose pellets or any other standardized food substance? The optogenetic stimulation experiments should be conducted with some standardized food substances as well so that the nutritional components are very clear. In addition, the brands and relative nutritional components of the “real food” substances should be listed. Even the amount of quinine added to the butter was not listed, nor the method of mixing them.

We appreciate the reviewer’s feedback and the suggestion to clarify our choice of food substances. Our goal in using “real food” substances like jelly, peanut butter, and butter was to test foods of different natures, specifically high-fat (e.g., peanut butter) versus high-sugar (e.g., jelly, chocolate), to better understand the diverse hedonic drives behind feeding behavior. To address the reviewer’s concern about standardized food substances, **we have now included additional experiments using high-fat chow, which is a standardized food substance (Fig. 2d, e).**

Additionally, we have included in the Methods **detailed information on the nutritional components of all food substances** used in the study, which we believe adds further clarity (p. 33). We have also included further details on the preparation of quinine-butter mixtures, including the specific amount of quinine used and the method of mixing, in the Methods section on **page 31**.

Taken together, we believe that our results, based on testing seven different food substances, offer robust evidence to support our claims about hedonic feeding. We hope these additions address the reviewer’s concern.

3. The second biggest issue that I see is that the authors show that opto activation increases NTS release and that NTSR antagonist prevents opto-induced feeding. But what they do not show is whether palatable food consumption endogenously increases NTS release, which would greatly strengthen the conclusions of the paper.

We appreciate and share the reviewer’s desire to explore endogenous NTS release during food consumption. To address this, our first step was to establish and validate *ntsLight* for *in vivo* recordings. However, while *ntsLight1.1* provided robust readouts in brain slice recordings, its sensitivity was limited for reliably detecting signals *in vivo*. To overcome this challenge, we developed an improved version, *ntsLight2.0*, which exhibited enhanced sensitivity, making it more effective for *in vivo* applications. We conducted a substantial number of additional validation experiments for this new sensor, which are now included in **Extended Data Fig. 9**. We then used this updated sensor to demonstrate that optogenetic stimulation of the NAcLat→VTA pathway induces NTS release *in vivo* and that this release is significantly reduced following a high-fat diet (**Fig. 3n-p**). **These results further strengthen our claim that NTS is released in the NAcLat→VTA pathway and that chronic high-fat diet reduces NTS release, which is now supported by both *in vivo* and *ex vivo* experiments.**

To specifically address the referee’s question regarding whether palatable food consumption increases endogenous NTS release, we then performed *in vivo* recordings while mice consumed palatable foods such as jelly and measured endogenous NTS release in the VTA. However, we realized that this experiment, while important, has a significant limitation: it lacks pathway specificity. This is a critical issue, as NTS is released in the VTA from multiple sources, such as the lateral hypothalamus and IPAC (a nucleus of the central extended amygdala) (Furlan et al., 2022; Soden et al., 2023), and NTS release patterns may vary depending on its specific source during food consumption. Consistent with this, we observed considerable variability in the NTS

signals, with recordings showing both increases and decreases in NTS release in the VTA. Perhaps this variability is not surprising, as the referee correctly noted in their subsequent comment that NTS's role in feeding likely varies across different brain structures. But due to the difficulty in clearly interpreting these results, we decided not to include the data in the current study. Instead, we plan to explore the heterogeneity of NTS inputs to the VTA in hedonic feeding behavior in future studies, recognizing that this investigation may require several years of experimentation.

Acknowledging the importance of the referee's point, we sought alternative approaches to provide direct evidence for NTS's role in hedonic feeding behavior, beyond relying solely on *in vivo* pharmacology. To this end, we performed an additional experiment in which we knocked out NTS expression in NAcLat neurons and optogenetically stimulated NAcLat terminals in the VTA. In the acute feeding assay, control mice responded to optogenetic stimulation by increasing hedonic food consumption, as expected. However, in NTS KO mice, optogenetic stimulation of the NAcLat→VTA pathway failed to promote hedonic feeding, demonstrating that NTS is indeed necessary for this behavior. **These new results are now presented in Figure 4a-e.**

Together with our previous results, we believe that these new data now provide strong evidence that NTS is released in the NAcLat→VTA pathway, that it is critical for hedonic feeding behavior, and that chronic high-fat diet reduces NTS release in this pathway.

4. In general, I find the discussion of neurotensin and feeding sorely lacking. There is abundant evidence in many brain regions that neurotensin is anorexigenic and decreases feeding, including VTA, lateral septum, hypothalamus and nucleus of tract solitarius. The only other exception that I'm aware of being the CeA, where activation of Nts neurons led to increase consumption of ethanol and sucrose liquids, but did not change solid food consumption (PMID: 31744862). That activation of a (potentially) subpopulation of Nts neurons leads to increased palatable consumption is therefore surprising and the implications of that should be discussed thoroughly.

We thank the reviewer for their valuable comment. **We have expanded our discussion on this important topic and included references to the literature suggested by the reviewer (p. 12/13).** The reviewer is correct that NTS's role in feeding has been extensively studied across various brain regions, where it is often thought to inhibit feeding. Therefore, our findings demonstrating that NTS in NAcLat inputs to the VTA selectively promotes hedonic feeding behaviors, rather than non-hedonic ones, are indeed surprising and reveal a novel and unexpected role for NTS in the VTA. Given that previous studies have shown that NTS infusion into the VTA, as in other brain regions, decreases feeding, we have focused our discussion on NTS's specific actions within the VTA—the primary focus of this study—and proposed several potential explanations for this apparent contradiction.

5. In Figure 1, the authors show neuronal activity depending on the behavior. They mention in the text l.119 'we found increased firing rate in opto-tagged units during jelly, but not chow, consumption', however they show 'approaching food' and not feeding. In the method they mention that feeding events were verified by analyzing manually the video. Why don't they dissociate approach from actual feeding? Besides, the animals are barely approaching the chow so the statement in that sentence should be reconsidered. Same thing for l.123-124, HFD mice barely approach the food, but the authors stipulate that there is no increase during jelly eating. More detailed analysis is required or the statement should be moderated.

We thank the reviewer for this comment and apologize for the confusion caused by the wording in the original manuscript. To address this concern, we have **carefully revised both the analysis and terminology for the experiment shown in Figure 1**, specifically clarifying that we are referring to **feeding behavior** rather than just **food approach**.

The reviewer correctly notes that REG mice consume chow less frequently compared to jelly and that HFD mice consume jelly less frequently compared to REG mice. While these behavioral responses limit data acquisition by producing non-uniform event distributions, they also provide a valuable opportunity to detect neural activity patterns associated with naturalistic occurring feeding behaviors, minimizing potential confounds related to feeding motivation, which is a central focus of our study.

As shown in **Fig. 1k-m**, we have performed a more detailed analysis correlating individual behavioral motifs with neural activity. Importantly, we found that the classification of neural responses is not solely determined by the sheer number of events. If this were the case, we would expect to observe an inflated number of false positives for highly frequent behaviors such as rearing or running. However, our analysis indicates this is not the case. Moreover, in **Extended Data Fig. 3f**, we show an inverse correlation between the time spent in a given motif and the firing rate of individual units. Notably, many units with high firing rates are associated with motifs involving a smaller number of events (e.g., jelly consumption) compared to motifs with more frequent events (e.g., walking).

We hope the reviewer agrees that this re-analysis of the electrophysiology data shown in **Figure 1**, the additional data provided in **Extended Data Fig. 3**, and the reassessment of our terminology ensure that our conclusions better align with the behavioral observations and the corresponding neural activity dynamics.

6. There was a lot of evidence cited from rodents that HFD leads to reduced preference for reward. What is the evidence in humans that there is reduced preference for reward?

We appreciate the reviewer's request for clarification regarding the evidence for reduced preference for reward in humans following HFD. While much of the detailed mechanistic research has been conducted in rodent models, there is growing evidence from human studies that prolonged consumption of a high-fat or high-calorie diet can lead to alterations in reward processing and preference.

For example, studies in humans have shown that individuals with obesity or those consuming high-fat diets exhibit reduced sensitivity to the rewarding effects of palatable foods, as indicated by blunted activity in brain regions involved in reward, such as the striatum and prefrontal cortex (García-García et al., 2013; Stice et al., 2008). Additionally, research has demonstrated that overconsumption of high-calorie foods can lead to altered dopamine signaling in humans, which may contribute to a diminished response to reward (Volkow et al., 2008).

Moreover, clinical studies have reported that individuals consuming a diet high in fat and sugar often exhibit a reduced preference for previously rewarding stimuli, a phenomenon that parallels findings in rodent models (Small et al., 2003). These studies suggest that, similar to rodents, prolonged exposure to a high-fat diet in humans may lead to reduced preference for and sensi-

tivity to reward. **We have now incorporated references to human studies alongside the rodent data to provide a better view of how HFD impacts reward preferences across species (p. 14).**

7. The authors provide in the Introduction a rationale for studying VTA  NAcc, but then instead pivot to studying NAcc -> VTA without any rationale. Moreover, the justification for NAcc-Lat and not core or medial shell is also not justified in any way, nor is any of that previous literature cited.

We thank the reviewer for drawing attention to this important point. In our previous study, we found that optogenetic stimulation of the NAcLat→VTA pathway promotes robust reward-related behaviors, including place preference and intracranial self-stimulation (Yang et al., 2018). In contrast, stimulation of NAc medial shell inputs to the VTA induced a general state of behavioral suppression, which was not specific to either reward- or aversion-related behaviors (Yang et al., 2018). Furthermore, we observed that the connectivity between NAc medial shell and VTA neurons is more complex, involving predominantly direct inhibition of both VTA dopamine neurons and VTA GABAergic neurons, which may account for this behavioral outcome. Additionally, since the effect of NTS on dopamine cell activity is primarily excitatory, it is unlikely that NTS is co-released in this pathway. This aligns with our observation that NTS expression is sparse in the NAc medial shell.

Given this complexity, we recognized that interpreting the effects of optogenetic stimulation on feeding behavior in the NAc medial shell pathway would be more challenging. In contrast, the robust and consistent behavioral phenotype observed with stimulation of NAcLat inputs to the VTA provided a clear rationale for focusing on this pathway in our current study. **We have revised the introduction to clearly explain this rationale (p. 3). Additionally, we have included a brief description in the Methods section (p. 28) further outlining our decision to prioritize the NAcLat→VTA pathway over the NAc medial shell to VTA pathway, keeping the focus of the introduction on the NAcLat→VTA pathway.**

8. Figure 1i: The figure legend states that it is 18 units from 8 mice. There are many units with low firing rates and only a few units with high firing rates that seem to drive the statistical difference. Are these units treated statistically independently? Are all of the units with high firing rates from just one or two mice? I think this needs to be considered very carefully because the claim is not necessarily supported if the high firing rates are derived only from a few animals.

We thank the reviewer for their thoughtful comment. To address this concern, we conducted an additional nested analysis to confirm that the observed effect is robust and not disproportionately influenced by data from just a few mice. As shown below in **Figure R1**, the statistical difference remains consistent when averaging multiple sessions of each animal and analyzing REG/HFD differences (with n = number of mice). This additional analysis supports our findings independently of individual variations in firing rates. By accounting for the non-independence of repeated measures within individual mice, this analysis confirms that our results are not biased by repeated sampling and remain statistically robust.

Additionally, in **Fig. 1l, m**, we now present the Z-scored firing rates for all opto-tagged and non-tagged units, demonstrating that the increase in firing rate in response to hedonic feeding is common across most units rather than being driven by only a few cells. Lastly, we would like to note that while some mice were tested on more than one day, the location of the microdrive was

lowered by $>80 \mu\text{m}$ between days. This procedure strongly suggests that different cells were recorded, further minimizing the risk of over-representation from individual neurons.

Figure R1. Nested analysis confirms robustness of observed effects across individual animals. The percentage of responsive units (y-axis) during each behavioral motif is plotted for REG (gray) and HFD (orange) mice showing for increased response (IR, top) and decreased responses (DR, bottom) in neural activity. Each bar represents the percent of responsive units detected across all experimental days and averaged by mouse. The mean \pm SEM, and statistical differences were assessed using a nested ANOVA, accounting for the non-independence of repeated measures within individual mice. This analysis shows that the increase in firing rate during jelly consumption is consistent across multiple animals and not disproportionately driven by data from a small subset of mice. This robustness supports the conclusion that the observed effects are not biased by repeated sampling or individual variability ($n_{\text{REG}} = 8$ mice, $n_{\text{HFD}} = 7$ mice; 2-way RM ANOVA with Holm-Šidák's multiple comparisons test).

9. Similar to point 7, but for Extended Data figure 3c: the legend says 12 experiments from 8 mice. Does this mean that the same mouse was tested multiple times but they are being treated as independent samples? This can artificially inflate the data. If one mouse has multiple trials, either they should be averaged together for a nested analysis or just one trial be used.

We thank the reviewer for raising this important point. To address this concern, **we have revised the “time spent” analysis** to reflect the average time spent for each mouse across different experimental days, thereby accounting for repeated measurements within the same animal (**Extended Data Fig. 3c, e, f**). This approach ensures that our analysis is not artificially inflated by treating multiple trials from the same mouse as independent samples, providing a more accurate representation of the data.

10. Extended fig. 6h. The authors show that 76% of NAcLatVTA neurons express NTS, but how many NTS neurons project to VTA? From the sample image it seems there are many more NTS neurons than CTB-labeled. This is important to know whether NTS is a true marker for this projection are just a general marker in NAcLat neurons.

We thank the reviewer for bringing this important point to our attention. However, based on our current experimental design, it is challenging to determine whether the unlabeled NTS neurons in the NAcLat also project to the VTA or elsewhere. The reason for this limitation is that the use of any retrograde tracer, including CTB, does not allow for labeling the entire population of projection neurons, as the number of labeled cells depends on the spread and uptake of the tracer

at the injection site. Therefore, it is expected that we observe unlabeled NTS-expressing cells in the NAcLat.

That said, we feel confident in concluding that NTS is highly expressed in the NAcLat→VTA pathway, as both our RNA-seq and *in situ* hybridization data consistently show a significant overlap between NTS expression and NAcLat→VTA projection neurons. However, there is a small percentage of neurons that project to the VTA but do not express NTS, as indicated by the data. Whether NTS-expressing NAcLat neurons project to other brain regions remains an open question. We plan to address this question in future studies as the focus of our current study is on the NAcLat→VTA pathway. **Accordingly, we have revised the manuscript to specifically state that we cannot rule out that there are NTS-expressing neurons in the NAcLat that project to other brain structures than the VTA (p. 7).**

11. Figure 5: the value seems too high for one mouse consumption over a week, the authors should double-check that it is actually normalized per animal per cage.

We thank the reviewer for bringing this discrepancy to our attention. Upon re-evaluating the data, we identified and corrected the values to accurately reflect the consumption of a single mouse over a week. The updated data are now properly normalized per animal per cage.

I also wonder if there is an effect of NTS over-expression on stress/anxiety as you show an increased jelly consumption in a restricted feeding in a non-home cage associated with a decrease food consumption of HFD in home cage. Since you performed an open field, I recommend to analyze anxiety and add it in the extended data in order to rule out any stress effect in addition to the studies on novelty detailed in point 1.

We thank the reviewer for this suggestion. **We re-analyzed the open field test to assess anxiety-related behavior by measuring the time the animals spent in the center of the chamber (Extended Data Fig. 13k).** As expected, NTS-OE mice exhibited significantly increased locomotion compared to CTRL mice. Additionally, NTS-OE mice spent more time in the center of the chamber, which may indicate reduced anxiety-related behavior.

However, it remains challenging to determine whether the increased center time is due to a general increase in locomotion or represents a bona fide reduction in anxiety associated with neurotensin overexpression in the NAcLat→VTA pathway. If the latter is true, it is possible that reduced anxiety could contribute to the observed changes in feeding behavior in response to NTS overexpression.

Furthermore, there is evidence that obesity is associated with increased anxiety and other mood disorders (Rajan and Menon, 2017). If the increased center time observed in NTS-OE mice reflects a bona fide reduction in anxiety due to neurotensin overexpression, this could represent an important therapeutic benefit beyond the observed changes in feeding behavior. Reduced anxiety may contribute not only to healthier feeding patterns but also to an improved overall mental state, which is highly relevant for the treatment of obesity and its comorbidities. We have now included this possibility and its implications in the revised manuscript (**pp. 12 and 15**).

12. Figure 5f shows the body weight evolution depending on the diet, it would be interesting to

have the parallel with food consumption evolution (in kcal in order to have both combine all phases in the same graph) as the authors only show the food consumption during HFD+Chow phase.

We thank the reviewer for this valuable suggestion. In response, we have included additional time points for both time-restricted and home-cage feeding to provide more comprehensive details on food consumption evolution across all phases (**Fig. 5h, i**).

Additionally, we have now included data on ‘kcal consumption’ in **Figure 1c** to further validate our HFD mouse model. For **Figure 5**, however, we opted to present food intake in grams rather than kcal to better reflect the mice’s choices between the different diet types, as this distinction aligns more directly with our focus on feeding behavior rather than purely caloric intake. That said, if the reviewer strongly feels that including kcal information is necessary, we are happy to incorporate it into the figure or extended data. Importantly, this addition would not alter our conclusions.

13. I find this sentence of the abstract confusing and opaque “Altogether, our results suggest that the transition from hedonically-driven eating to habitual consumption involves an intricate neural mechanism, where the brain’s reward system dynamically adapts to dietary influences. This adaptability is a crucial factor in the progression of obesity.” What data does this refer to? Which adaptability is a crucial factor in the progression of obesity. I don’t think I saw any data that changes the progression of obesity, rather most of the data showed that obesity that already occurs has altered reward dynamics.

We agree with the reviewer that our wording was imprecise and confusing, and this has been changed in the abstract.

Minor comm

1. Extended Figure 1F: why is this normalized to 5 mice / cage? Does that mean that mice are eating on average only 1.5g of chow a day? This doesn’t seem to make sense ... In addition, it might be helpful to provide kcal as well as g since the kcal/g differ for chow and HFD.

We thank the reviewer for pointing out this discrepancy. We agree that the normalization used in the original figure was indeed confusing and incorrect. To address this, we have now adjusted the data to reflect food consumption per mouse per week (g/week) to avoid any misunderstandings.

Additionally, as suggested by the reviewer, we have included a new graph in **Fig. 1c** showing food consumption in kcal to account for the differing kcal/g values of chow and HFD. Across all conditions, the mice consumed 1.5–3.4 g of chow or HF chow per day, which is consistent with observations from other cohorts of C57BL/6 mice.

2. Figure 1e and extended figure 1G: are these supposed to be the same data presented differently? Because the data points don’t align. If they aren’t the same data, what two distinct time points are they from?

We thank the reviewer for this important observation. Figure 1e and Extended Data Fig. 1g are not the same data; they represent two distinct cohorts of mice. Specifically, Extended Data Fig. 1g shows data from a separate cohort of mice to illustrate the baseline HFD phenotype and how home-cage feeding and time-restricted feeding behaviors are altered in response to chronic HFD. This additional cohort was included because, for our *in vivo* electrophysiology studies, mice needed to be single-housed, making home-cage feeding measurements impractical for that group.

To avoid further confusion, **we have moved the data from Extended Data Fig. 1g to Figure 1 (panels a-d)** to introduce the HFD phenotype and its effects on home-cage and time-restricted feeding more clearly.

3. Extended data figure 2 has no description in the manuscript text to say what the purpose of the figure is.

As suggested by the referee, we have added a sentence in the main text (p. 4) describing the data shown in **Extended Data Fig. 2**.

4. Figure 2d and similar: it is very confusing to the reader for the individual data to be underneath the summarized data. This individual data should either be moved to supplementary or put in more transparent underneath the summary data in one graph.

We agree with the referee that the individual traces are confusing to the reader. In response, we have removed the individual data from the figure. In the revised manuscript, individual data can be assessed in the accompanying statistical tables and source data files.

5. Extended data fig 10a-c: the authors should also compare to chow-fed mice to determine if NTS OE is restoring NTS levels or perhaps surpassing normal NTS levels.

We thank the reviewer for this suggestion. While we agree that comparing the effects of NTS-OE in chow-fed mice could provide valuable insights, repeating the experiment would require substantial time and resources. Injecting new cohorts of mice with the NTS overexpression and control vectors, followed by exposure to both regular and high-fat diets, would take several months to complete and divert resources from other critical experiments.

After careful consideration, we evaluated the potential impact of this additional data on the conclusions of our study. While this experiment could provide interesting context, we believe the knowledge gained would not significantly alter the key conclusions derived from our current dataset. Therefore, we have prioritized experiments that directly address the primary objectives and conclusions of our study.

6. Comparing Figure 3L to figure 5b the NTS release in HFD mice looks starkly different and much higher in Figure 5b. What is the reason for this?

We thank the reviewer for pointing out this apparent inconsistency. The experiments depicted in Figure 3l and Figure 5b were performed in different cohorts of mice and at different times, which could account for the observed variability. To minimize the effects of inter-animal variability, we

standardized the experimental conditions, including the same virus expression time, identical macroscope settings, and conducting recordings for the two experimental groups (e.g., REG vs. HFD mice in Fig. 3k-m) on the same day. However, the recordings for HFD mice shown in Figure 5a-c were from a separate cohort of mice, and these recordings were conducted on a different day. While we applied the same experimental protocols to compare HFD CTRL mice and HFD NTS-OE mice in Figure 5, we cannot entirely rule out the possibility of small differences in the exact recording location in the VTA or expression time when comparing the HFD mice used in Figure 3 to those in Figure 5.

To better address the reviewer's concern, we have now included new experiments directly comparing ***NTS release in vivo within the same mice exposed to different diets (Fig. 3n-p)***. The results from these experiments are consistent with our previous *ex vivo* data, which were performed in different animals. We hope this additional data addresses any concerns about variability arising from comparisons across different animals.

7. Extended data figure 5f: for consistency, please replace pre-feeding with primed-feeding trial. Additionally, why is there only one group? It would be interesting to compare the 3week evolution with eYFP group.

Thank you for this suggestion. We have revised **Extended Data Fig. 5f** to use the term "primed-feeding trial" for consistency. Initially, the data presented included both eYFP and ChR2 groups combined; however, we have now separated these groups in the analysis. Our revised analysis confirms that while the effects of time remain significant, there is no difference between the eYFP and ChR2 groups. This separation and additional comparison further highlight that the observed outcomes are consistent across both groups, emphasizing time as the primary factor influencing these trials.

8. Figure 5h: the color legend is missing; I recommend using the same color as 5g for consistency.

We thank the reviewer for picking up this mismatch, which has been corrected in the revised manuscript.

References

- Chaix, A., Zarrinpar, A., Miu, P., Panda, S., 2014. Time-Restricted Feeding Is a Preventative and Therapeutic Intervention against Diverse Nutritional Challenges. *Cell Metab.* 20, 991–1005. <https://doi.org/10.1016/j.cmet.2014.11.001>
- Furlan, A., Corona, A., Boyle, S., Sharma, R., Rubino, R., Habel, J., Gablenz, E.C., Giovanniello, J., Beyaz, S., Janowitz, T., Shea, S.D., Li, B., 2022. Neurotensin neurons in the extended amygdala control dietary choice and energy homeostasis. *Nat. Neurosci.* 25, 1470–1480. <https://doi.org/10.1038/s41593-022-01178-3>
- García-García, I., Jurado, M.A., Garolera, M., Segura, B., Marqués-Iturria, I., Pueyo, R., Vernet-Vernet, M., Sender-Palacios, M.J., Sala-Llonch, R., Ariza, M., Narberhaus, A., Junqué, C., 2013. Functional connectivity in obesity during reward processing. *NeuroImage* 66, 232–239. <https://doi.org/10.1016/j.neuroimage.2012.10.035>

- Hatori, M., Vollmers, C., Zarrinpar, A., DiTacchio, L., Bushong, E.A., Gill, S., Leblanc, M., Chaix, A., Joens, M., Fitzpatrick, J.A.J., Ellisman, M.H., Panda, S., 2012. Time-Restricted Feeding without Reducing Caloric Intake Prevents Metabolic Diseases in Mice Fed a High-Fat Diet. *Cell Metab.* 15, 848–860. <https://doi.org/10.1016/j.cmet.2012.04.019>
- Kohsaka, A., Laposky, A.D., Ramsey, K.M., Estrada, C., Joshu, C., Kobayashi, Y., Turek, F.W., Bass, J., 2007. High-fat diet disrupts behavioral and molecular circadian rhythms in mice. *Cell Metab.* 6, 414–421.
- Rajan, T.M., Menon, V., 2017. Psychiatric disorders and obesity: a review of association studies. *J. Postgrad. Med.* 63, 182–190.
- Small, D.M., Jones-Gotman, M., Dagher, A., 2003. Feeding-induced dopamine release in dorsal striatum correlates with meal pleasantness ratings in healthy human volunteers. *NeuroImage* 19, 1709–1715. [https://doi.org/10.1016/s1053-8119\(03\)00253-2](https://doi.org/10.1016/s1053-8119(03)00253-2)
- Soden, M.E., Yee, J.X., Zweifel, L.S., 2023. Circuit coordination of opposing neuropeptide and neurotransmitter signals. *Nature* 619, 332–337. <https://doi.org/10.1038/s41586-023-06246-7>
- Stice, E., Spoor, S., Bohon, C., Small, D.M., 2008. Relation between obesity and blunted striatal response to food is moderated by Taq1A A1 allele. *Science* 322, 449–452. <https://doi.org/10.1126/science.1161550>
- Volkow, N.D., Wang, G.-J., Fowler, J.S., Telang, F., 2008. Overlapping neuronal circuits in addiction and obesity: evidence of systems pathology. *Philos. Trans. R. Soc. Lond. B. Biol. Sci.* 363, 3191–3200. <https://doi.org/10.1098/rstb.2008.0107>
- Woodworth, H.L., Batchelor, H.M., Beekly, B.G., Bugescu, R., Brown, J.A., Kurt, G., Fuller, P.M., Leininger, G.M., 2017. Neurotensin Receptor-1 Identifies a Subset of Ventral Tegmental Dopamine Neurons that Coordinates Energy Balance. *Cell Rep.* 20, 1881–1892. <https://doi.org/10.1016/j.celrep.2017.08.001>
- Woodworth, H.L., Perez-Bonilla, P.A., Beekly, B.G., Lewis, T.J., Leininger, G.M., 2018. Identification of Neurotensin Receptor Expressing Cells in the Ventral Tegmental Area across the Lifespan. *eneuro* 5, ENEURO.0191-17.2018. <https://doi.org/10.1523/ENEURO.0191-17.2018>
- Yang, H., de Jong, J.W., Tak, Y., Peck, J., Bateup, H.S., Lammel, S., 2018. Nucleus Accumbens Subnuclei Regulate Motivated Behavior via Direct Inhibition and Disinhibition of VTA Dopamine Subpopulations. *Neuron* 97, 434-449.e4. <https://doi.org/10.1016/j.neuron.2017.12.022>

January 22, 2025

Re: Decision on Nature manuscript 2024-01-01245B

RESPONSE TO REVIEWERS

Referee #1:

I commend the authors on a thorough revision that has addressed my concerns. I have no further comments.

Thank you for your kind words and for acknowledging the thoroughness of our revision.

Referee #2:

The authors have successfully addressed my concerns. The manuscript is significantly improved, and I have no further suggestions.

Thank you for your positive feedback and for recognizing the improvements in our revised manuscript.

Referee #3:

I was pleased to see the revision for the manuscript by Gazit Shimoni and colleagues. Overall, I was incredibly impressed with the responses to critiques and I think the added experiments significantly improve the paper, in particular the addition of the inhibition experiments and in vivo application of Nts-light. Although I still think it is a bit of a missing link that Nts release is not shown from physiological behavior and only from opto stim, I am satisfied with the authors explanation of why this was not feasible at this time. The manuscript is now a very comprehensive investigation that provides an important mechanism for linking obesity with the devaluation of reward, although the potential therapeutic benefit from neurotensin agonism on weight loss is already known. However, the finding that this is linked also to devaluation and reverses devaluation provides a previously unknown angle. I have just two easily addressable points to still make.

Thank you for your thoughtful and positive feedback on our revised manuscript. We greatly appreciate your feedback throughout the review process, which has been instrumental in improving our manuscript.

- 1. I still think the title is somewhat misleading. I do not see much in the paper about obesity progression per se, and I believe the title should reflect more about the devaluation of reward, which is actually the main focus.*

In the revised manuscript, we changed the title to: "Changes in neurotensin signaling drive hedonic devaluation in obesity". However, we prefer to use the term "hedonic devaluation"

rather than “reward devaluation” because our study emphasizes changes in the pleasurable aspects of food consumption, particularly the hedonic drive to consume calorie-rich foods. “Hedonic devaluation” captures this specificity better than “reward devaluation.” Moreover, the mechanisms we describe are linked to hedonic feeding behavior rather than broader reward processes (e.g., motivation or reinforcement learning). This aligns more closely with the concept of “hedonic devaluation.”

2. In addition, perhaps the authors plan to upload it before publication, but I did not see any mention of submitting the RNA sequencing data to the GEO database.

The RNAseq datasets generated during this study are now available at NCBI GEO; GSE287548. This statement has been included in the Data Availability section.

Minor points:

1. Pg. 7. States that upregulated genes are Log Fold Change >-1, but I presume they mean 1?

Thank you. Yes, the reviewer is correct. This has been corrected in the revised manuscript.

2. Extended Fig 4J legend seems to be wrong. The schematic states Day 2 has laser but below day 2 is white and on the graph it's the dark one which has the laser

Thank you for catching this. This has been corrected in the revised figure.

3. P11: the last paragraph states that there is no change in metabolic rate because there was no change in body temperature, but they have an increase in locomotion and per se energy expenditure. This would have to be examined more closely to determine no change in metabolic rate.

We have removed the statement regarding the metabolic rate in order to avoid any potential confusion.

4. Please check all references. For example, ref. 35 is not the first demonstration that Nts in the LS controls feeding, but rather is Azevedo et al., 2020.

Thank you for pointing this out. All references were checked and the paper mentioned by the referee has been included.